# Achieving the Pareto Frontier of Regret Minimization and Best Arm Identification in Multi-Armed Bandits

**Zixin Zhong**                                              *zzhong10@ualberta.ca*
*Department of Computing Science,*
*University of Alberta*

**Wang Chi Cheung**                                          *isecwc@nus.edu.sg*
*Department of Industrial Systems and Management,*
*Institute of Operations Research and Analytics,*
*National University of Singapore*

**Vincent Y. F. Tan**                                        *vtan@nus.edu.sg*
*Department of Mathematics,*
*Department of Electrical and Computer Engineering,*
*Institute of Operations Research and Analytics,*
*National University of Singapore*

**Reviewed on OpenReview:** *https://openreview.net/forum?id=XXfEmIMJDm*

## Abstract

We study the Pareto frontier of two archetypal objectives in multi-armed bandits, namely, regret minimization (RM) and best arm identification (BAI) with a fixed horizon. It is folklore that the balance between exploitation and exploration is crucial for both RM and BAI, but exploration is more critical in achieving the optimal performance for the latter objective. To this end, we design and analyze the BoBW-LIL'UCB($\gamma$) algorithm. Complementarily, by establishing lower bounds on the regret achievable by any algorithm with a given BAI failure probability, we show that (i) no algorithm can simultaneously perform optimally for both the RM and BAI objectives, and (ii) BoBW-LIL'UCB($\gamma$) achieves order-wise optimal performance for RM or BAI under different values of $\gamma$. Our work elucidates the trade-off more precisely by showing how the constants in previous works depend on certain hardness parameters. Finally, we show that BoBW-LIL'UCB outperforms a close competitor UCB$_\alpha$ (Degenne et al., 2019) in terms of the time complexity and the regret on diverse datasets such as MovieLens and Published Kinase Inhibitor Set.

## 1 Introduction

Consider a drug company Dandit (Drug Bandit) that wants to design an effective vaccine for a certain virus. It has a certain number of feasible options, say $L = 10$. Because Dandit has a limited budget, it can only test vaccines for a fixed number of times, say $T = 1,000$. Using the limited number of tests, it wants to find the option that will lead to the "best" outcome, e.g., the maximum efficacy of the drug. At the same time, Dandit aims to protect individuals from potentially adverse side effects of the vaccines to be tested. How can Dandit find the optimal drug design and, at the same time, protect the health of participants? We design an algorithm BoBW-LIL'UCB that allows Dandit to balance between these two competing targets. In complement, we also show that it is *impossible* for Dandit to achieve optimal performances for both targets *simultaneously*, and Dandit has to settle for operating on the Pareto frontier of the two objectives.

To solve Dandit's problem, we study the *Cumulative Regret Minimization* (RM) and *Best Arm Identification* (BAI) problems for stochastic bandits with a *fixed time horizon or budget*. While most existing works only study one of these two targets (Auer et al., 2002a; Audibert & Bubeck, 2010), Degenne et al. (2019) designed

the $\mathrm{UCB}_\alpha$ algorithm for both RM and BAI with a *fixed confidence*. Therefore, these studies are not directly applicable to Dandit's problem as Dandit is interested in obtaining the optimal item and minimizing the damage across a *fixed* number of tests. However, our setting dovetails neatly with company Dandit's goals. Dandit can utilize our algorithm to sequentially and adaptively select different design options to test the vaccines and to eventually balance between choosing the optimal vaccine and, in the process, mitigating any physical damage on the participants. We also show that Dandit cannot achieve both targets optimally and simultaneously.

Beyond any specific applications, we believe this problem is of fundamental theoretical importance in the broad context of *multi-armed bandits* (MAB). In order to design an efficient bandit algorithm, a well-known challenge is to balance between *exploitation* and *exploration* (Auer et al., 2002a; Lattimore & Szepesvári, 2020; Kaufmann & Garivier, 2017). Our work **quantifies** the Pareto frontier of RM and BAI, as well as the effects of exploitation and exploration on these two aims.

**Main contributions.** In stochastic bandits, there are $L$ items with different unknown reward distributions. At each time step, a random reward is generated from each item's distribution. Based on the previous observations, a learning agent selects an item and observes its reward. Given the number of time steps $T \in \mathbb{N}$, the agent aims to maximize the cumulative rewards and to identify the optimal item with high probability.

Our first main contribution is the BoBW-lil'UCB($\gamma$) algorithm. BoBW-lil'UCB($\gamma$) is designed for both RM and BAI over a fixed time horizon, which achieves Pareto-optimality of RM and BAI in some regimes.

(i) On one hand, we can shrink the confidence radius of each item by increasing $\gamma$, which encourages BoBW-lil'UCB($\gamma$) to pull items with high empirical mean rewards (exploitation) and generally leads to high rewards (i.e., small regret).
(ii) On the other hand, we can enlarge the confidence radius by decreasing $\gamma$ to encourage the exploration of items that have not been sufficiently pulled in previous time steps (exploration); this will result in a high BAI success probability.

The parameter $\gamma$ in BoBW-lil'UCB($\gamma$) can be tuned such that either its cumulative regret or its failure probability almost matches the corresponding state-of-the-art lower bound (Lai & Robbins, 1985; Carpentier & Locatelli, 2016). The performance of BoBW-lil'UCB($\gamma$) implies that exploitation is more critical in achieving the optimal performance for RM, while exploration is more crucial for BAI in stochastic bandits. We also analyze the Exp3.P algorithm proposed by Auer et al. (2002b) for both RM and BAI, which indicates the similar trade-off between these two aims in adversarial bandits.

Moreover, we evaluate the Pareto frontier of RM and BAI theoretically. In Lattimore & Szepesvári (2020), Note 33.2 and Exercise 33.5 only explore the sub-optimality for BAI of an asymptotically-optimal RM algorithm, and provide asymptotic bounds with constant $\varepsilon$ (See Section 3 for the meaning of $\varepsilon$). Our work goes beyond the asymptotic regimes in that observation, by exploring the Pareto frontier of RM and BAI for any algorithm in Theorems 5.1 and 5.3 in the finite horizon / budget setting. Our **non-asymptotic** bounds **quantify** how the trade-off between regret and BAI probability depends on the hardness quantity $H_2$ and gap parameter $\Delta_{1,i}$'s of an instance (see definitions in Section 2), instead of fixed constants such as $\varepsilon$ (Lattimore & Szepesvári, 2020). Another relevant work is Bubeck et al. (2009), which explores the trade-off between cumulative regret and *simple regret*.[1] Due to the relation between BAI and simple regret, our results precisely **quantify** the values of constants $C$ and $D$ in Bubeck et al. (2009) (see Section 5 for details). While these two works focus on the stochastic bandits, we also analyze the Pareto frontier between RM and BAI in adversarial bandits in Appendix B.

Furthermore, BoBW-lil'UCB($\gamma$) empirically outperforms a close competitor $\mathrm{UCB}_\alpha$ (Degenne et al., 2019) in difficult scenarios in which the differences between the optimal and suboptimal items are small. While both algorithms identify the optimal item with high probability, $\mathrm{UCB}_\alpha$, designed for the fixed-confidence case, requires a longer horizon to do so and also suffers from larger regret. This demonstrates the superiority of BoBW-lil'UCB($\gamma$) under the fixed-budget setting, which it is specifically designed for.

**Novelty.** (i) We are the first to design an algorithm for both RM and BAI with a fixed budget. We can adjust the proposed BoBW-lil'UCB($\gamma$) algorithm to perform (near-)optimally for both RM and BAI with

---

[1]When there is no ambiguity, we abbreviate "cumulative regret" as "regret".

proper choices of $\gamma$. (ii) The performance of BoBW-lil'UCB($\gamma$) implies that exploitation is more crucial to obtain a small regret, while exploration is more critical to shrink the BAI failure probability. (iii) We quantify the Pareto frontier of RM and BAI. We show that it is inevitable for any algorithm to compromise between RM and BAI in a fixed horizon setting. Beyond the stochastic bandits, we also provide a preliminary study on the adversarial bandits.

**Literature review.** Both the RM and BAI problems have been studied extensively for stochastic multi-armed bandits. Firstly, an RM algorithm aims to maximize its cumulative rewards, i.e., to minimize its regret (the gap between the highest cumulative rewards and the obtained rewards). One line of seminal works on RM involve the class of *Upper Confidence Bound* (UCB) algorithms (Auer et al., 2002a; Garivier & Cappé, 2011), while another line of works study *Thompson sampling* (TS) algorithms (Agrawal & Goyal, 2012; Russo & Van Roy, 2014; Agrawal & Goyal, 2017). Lai & Robbins (1985) derived a lower bound on the regret of any online algorithm.

Secondly, there are two complementary settings for BAI: (i) given $T \in \mathbb{N}$, the agent aims to maximize the probability of finding the optimal item in at most $T$ steps (Audibert & Bubeck, 2010; Karnin et al., 2013; Zhong et al., 2021a); (ii) given $\delta > 0$, the agent aims to find the optimal item with the probability of at least $1 - \delta$ in the smallest number of time steps (Bubeck et al., 2013; Kaufmann & Kalyanakrishnan, 2013). These two settings are known as the *fixed-budget* and *fixed-confidence* settings respectively. Moreover, Kaufmann et al. (2016) presented theoretical findings for both settings, including a lower bound for two-armed bandits and a lower bound for multi-armed Gaussian bandits under the fixed-budget setting. Carpentier & Locatelli (2016) established a lower bound on the failure probability of any algorithm in a fixed time horizon.

While most existing works focus solely on RM or BAI, Degenne et al. (2019) explored both goals with a fixed confidence and proposed the UCB$_\alpha$ algorithm. Recently Kim et al. (2023) also focused on the fixed-confidence setting and studied the trade-off between RM and Pareto Front Identification (PFI) in linear bandits; PFI is a generalization of BAI since in this setting each arm has a *vector* reward instead of a scalar one. Simchi-Levi & Wang (2023) provided a novel definition of Pareto Optimality and aimed to solve a corresponding minimax multi-objective optimization problem, which has a different focus from this work. To the best of our knowledge, there is no existing analysis of a single, unified algorithm for both RM and BAI given a fixed horizon. Our work fills in this gap by proposing the BoBW-lil'UCB($\gamma$) algorithm and proving that it achieves Pareto-optimality in some regimes. We also study the Pareto frontier of RM and BAI, which depends on the balance between exploitation and exploration. We show that a single algorithm cannot perform optimally for both RM and BAI simultaneously.

## 2 Problem Setup

For any $n \in \mathbb{N}$, we denote the set $\{1, \ldots, n\}$ as $[n]$. Let there be $L \in \mathbb{N}$ ground items, contained in $[L]$. A random variable $X$ (or its distribution) is $\sigma$-sub-Gaussian ($\sigma$-SG) if $\mathbb{E}[e^{\lambda(X - \mathbb{E}X)}] \leq \exp(\lambda^2 \sigma^2 / 2)$. Each item $i \in [L]$ is associated with a $\sigma$-SG reward distribution $\nu_i$, mean $w_i$, and variance $\sigma_i^2$. The distributions $\{\nu_i\}_{i \in [L]}$, means $\{w_i\}_{i \in [L]}$, and variances $\{\sigma_i^2\}_{i \in [L]}$ are unknown to the agent. We let $\{g_{i,t}\}_{t=1}^T$ be the i.i.d. sequence of rewards associated with item $i$ during the $T$ time steps; each $g_{i,t}$ is an independent sample from $\nu_i$.

We focus on stochastic instances with a *unique* item having the highest mean reward, and assume that $w_1 > w_2 \geq \ldots \geq w_L$, so the unique *optimal* item $i^* = 1$. Note that the items can, in general, be arranged in any order; the ordering that $w_i \geq w_j$ for $i < j$ is employed to ease our discussion. We denote $\Delta_{1,i} := w_1 - w_i$ as the *optimality gap* of item $i$, and assume $\Delta_{1,i} \leq 1$ for all $i \in [L]$; this can be achieved by rescaling the instance if necessary. We define the *minimal optimality gap*

$$\Delta := \min_{i \neq 1} \Delta_{1,i}.$$

Clearly, $\Delta > 0$. We characterize the *hardness* of an instance with the following canonical quantities:

$$H_1 := \sum_{i \neq 1} \frac{1}{\Delta_{1,i}} \quad \text{and} \quad H_2 := \sum_{i \neq 1} \frac{1}{\Delta_{1,i}^2}. \tag{2.1}$$

The hardness quantity $H_1$ is involved in some near-optimal regret bounds (Auer et al., 2002a; Agrawal & Goyal, 2012). The quantity $H_2$ was first introduced in Audibert & Bubeck (2010) and appears in many landmark works on BAI (Jamieson et al., 2014; Karnin et al., 2013; Carpentier & Locatelli, 2016).

The agent uses an *online algorithm* $\pi$ to decide the item $i_t^\pi$ to pull at each time step $t$, and the item $i_{\text{out}}^{\pi,T}$ to output eventually. More formally, an algorithm consists of a tuple $\pi := ((\pi_t)_{t=1}^T, \psi_T^{\pi,T})$, where

- the *sampling rule* $\pi_t$ determines, based on the observation history, the item $i_t^\pi$ to pull at time step $t$. That is, the random variable $i_t^\pi$ is $\mathcal{F}_{t-1}$-measurable, where $\mathcal{F}_t := \sigma(i_1^\pi, g_{i_1^\pi, 1}, \ldots, i_t^\pi, g_{i_t^\pi, t})$;
- the *recommendation rule* $\psi_T^{\pi,T}$ chooses an item $i_{\text{out}}^{\pi,T}$, that is, by definition, $\mathcal{F}_T$-measurable.

Moreover, we define the *pseudo-regret $R_T$* of $\pi$ as

$$R_T(\pi) := \max_{1 \le i \le L} \mathbb{E}\left[\sum_{t=1}^T g_{i,t}\right] - \mathbb{E}\left[\sum_{t=1}^T g_{i_t^\pi, t}\right] = T \cdot w_1 - \mathbb{E}\left[\sum_{t=1}^T w_{i_t^\pi}\right].$$

The algorithm $\pi$ aims to both minimize the pseudo-regret $R_T(\pi)$ and at the same time, to identify the optimal item with high probability, i.e., to minimize the *failure probability* $e_T(\pi) := \Pr(i_{\text{out}}^{\pi,T} \ne 1)$. We omit $T$ and / or $\pi$ in the superscript or subscript when there is no cause of confusion. We write $R_T(\pi)$ as $R_T(\pi, \mathcal{I})$, $e_T(\pi)$ as $e_T(\pi, \mathcal{I})$ when we wish to emphasize their dependence on both the algorithm $\pi$ and the instance $\mathcal{I}$.

## 3 Discussion on Existing Algorithms

Although there is no existing work that analyzes a single algorithm for both RM and BAI in a fixed horizon, it is natural to ask if an algorithm which is originally designed for RM can also perform well for BAI, and vice versa. In Table 3.1, we present the theoretical results from some existing works. We focus on algorithms that are with (potential) theoretical guarantees for both RM and BAI. We define

$$H'_p := \max_{i \ne 1} \frac{i^p}{\Delta_i^2} \quad \text{and} \quad C_p := 2^{-p} + \sum_{r=2}^L r^{-p}$$

for $p > 0$ as in Shahrampour et al. (2017). We abbreviate Sequential Halving as SH, Nonlinear Sequential Elimination with parameter $p$ as NSE($p$), and UCB-E with parameter $a$ as UCB-E($a$). Also see Appendix A for more discussions.

Table 3.1: Comparison among upper bounds for algorithms and lower bounds in stochastic bandits.

| Algorithm/Instance | Pseudo-regret $R_T$ | Failure Probability $e_T$ | References |
|---|---|---|---|
| SH | $\Theta(T)$ | $\approx \exp\left(-\dfrac{T}{8H_2 \log_2 L}\right)$ | Karnin et al. (2013) |
| NSE($p$) | $\Theta(T)$ | $\approx \exp\left(-\dfrac{2(T-L)}{H'_p C_p}\right)$ | Shahrampour et al. (2017) |
| UCB-E($\alpha \log T$) | $6.3\alpha^2 H_1 \log T$ | $2LT^{1-2\alpha/25}$ 

 (when $\alpha \log T \le \frac{25(T-L)}{36H_2}$) | Corollary A.1, Audibert & Bubeck (2010) |
| BoBW-lil'UCB($\gamma$) | $H_1 \log T$ (Prob-dep) | $\approx L\exp\left(-\frac{T-L}{144H_2}\right)$ | Theorems 4.1 and 4.2 |
| | $\sqrt{TL}\log T$ (Prob-indep) | (when $\gamma \ge \gamma_1(\Delta, H_2)$) | |
| Stochastic Bandits (Lower Bound) | $\approx 4H_1 \log T$ | $\dfrac{1}{6}\exp\left(-\dfrac{400T}{H_2 \log L}\right)$ | Lai & Robbins (1985); Carpentier & Locatelli (2016) |

According to the discussions on RM and BAI in Lattimore & Szepesvári (2020), any algorithm with an asymptotically optimal regret would incur a failure probability lower bounded by $\Omega(T^{-1})$; this is much larger

than the state-of-the-art lower bound $\Omega(\exp(-400T/(H_2 \log L)))$ by Carpentier & Locatelli (2016). Therefore, we only include algorithms that were designed for BAI in Table 3.1.

Among the various BAI algorithms, SH and NSE($p$) perform almost the best. However, their bounds on the failure probabilities are incomparable in general. The comparison among more BAI algorithms is provided in Table A.1. Due to the designs of SH and NSE($p$), we surmise their regrets grow linearly with $T$, which is vacuous for the RM task.

Although UCB-E($\alpha \log T$) has upper bounds on both pseudo-regret and failure probability, its bound on the latter, which decays only polynomially fast with $T$ when $\alpha$ is an absolute constant, is clearly suboptimal vis-à-vis the state-of-the-art lower bound by Carpentier & Locatelli (2016). In order to achieve an exponentially decaying upper bound on $e_T$ (i.e., $\exp(-\Theta(T))$), we need to set $\alpha = O(T/\log T)$, and hence the regret bound (see Corollary A.1 in the supplementary) will be $O(T^2/\log T)$, which is vacuous.

The discussion above raises a natural question. Is it possible to provide a non-trivial bound on the regret for an algorithm that performs optimally for BAI over a fixed horizon? This motivates us to design BoBW-lil'UCB, which can be tuned to perform near-optimally for both RM and BAI.

## 4 The BoBW-lil'UCB Algorithm

We design and analyze BoBW-lil'UCB($\gamma$) (Best of Both Worlds-Law of Iterated Logs-UCB), an algorithm for both RM and BAI in a fixed horizon. By choosing parameter $\gamma$ judiciously, the guarantees of BoBW-lil'UCB($\gamma$) match those of the state-of-the-art algorithms for both RM (up to log factors) and BAI (concerning the exponential term).

---

**Algorithm 1** BoBW-lil'UCB($\gamma$)

---

1: **Input:** time budget $T$, size of ground set of items $L$, scale $\sigma > 0$, $\varepsilon \in (0,1)$, $\beta \geq 0$, and $\gamma \in (0,1)$.
2: Sample $i_t = i$ for $t = 1, \ldots, L$ and set $t = L$.
3: For all $i \in [L]$, compute $N_{i,L}$, $\hat{g}_{i,L}$, $C_{i,L,\gamma}$, $U_{i,L,\gamma}$:

$$N_{i,t} = \sum_{u=1}^{t} \mathbf{1}\{i_u = i\}, \ \hat{g}_{i,t} = \frac{\sum_{u=1}^{t} g_{i,t} \cdot \mathbf{1}\{i_u = i\}}{N_{i,t}},$$

$$C_{i,t,\gamma} = 5\sigma(1 + \sqrt{\varepsilon})\sqrt{\frac{2(1+\varepsilon)}{N_{i,t}} \cdot \log\left(\frac{\log(\beta + (1+\varepsilon)N_{i,t})}{\gamma}\right)}, \quad U_{i,t,\gamma} = \hat{g}_{i,t} + C_{i,t,\gamma}.$$

4: **for** $t = L+1, \ldots, T$ **do**
5:     Pull item $i_t = \arg\max_{i \in [L]} U_{i,t-1,\gamma}$.
6:     Update $N_{i_t,t}$, $\hat{g}_{i_t,t}$, $C_{i_t,t,\gamma}$, and $U_{i_t,t,\gamma}$.
7: **end for**
8: Output $i_{\text{out}} = \arg\max_{i \in [L]} \hat{g}_{i,T}$.

---

**Design of algorithm.** We design BoBW-lil'UCB in the spirit of the law of the iterated logarithm (LIL) (Darling & Robbins, 1967; Jamieson et al., 2014). We remark that it is a variation of the lil'UCB algorithm proposed by Jamieson et al. (2014). The three differences are:

(i) to construct the confidence radius $C_{i,t,\gamma}$, we replace $(1+\beta)$ and $\delta$ in lil'UCB by 5 and $\gamma$ in BoBW-lil'UCB($\gamma$) respectively;

(ii) in the design of $C_{i,t,\gamma}$, we also replace $\log((1+\varepsilon)N_{i,t})$ by $\log(\beta + (1+\varepsilon)N_{i,t})$;

(iii) BoBW-lil'UCB($\gamma$), which is designed for both RM and BAI in a fixed horizon, involves no stopping rule since it proceeds for *exactly* $T$ time steps; while lil'UCB is designed for BAI with a fixed confidence.

Although our algorithm depends on the choices of $\varepsilon$, $\beta$, and $\gamma$, we term it as BoBW-lil'UCB($\gamma$) instead of the more verbose BoBW-lil'UCB($\varepsilon, \beta, \gamma$) because we scale the confidence radius by only varying $\gamma$ which adjusts the performance of the algorithm. More precisely, inspired by the LIL (see Theorem C.1), we design item $i$'s confidence radius $C_{i,t,\gamma}$ with $N_{i,t}$ (the number of time steps when item $i$ is pulled up to and including

the $t^{\text{th}}$ time step) and $\hat{g}_{i,t}$ (the empirical mean of item $i$ at time step $t$), and its upper confidence bound $U_{i,t,\gamma}$ accordingly.

The design of BoBW-LIL'UCB$(\gamma)$ allows us to shrink $C_{i,t,\gamma}$, the confidence radius of each item $i$, by increasing $\gamma$; and vice versa. Moreover, with a fixed $\gamma$, if item $i$ is rarely pulled in previous time steps, it has a small $N_{i,t}$ and hence a large $C_{i,t,\gamma}$; and vice versa.

(i) Therefore, when $\gamma$ increases, the dominant term in $U_{i,t,\gamma} = \hat{g}_{i,t} + C_{i,t,\gamma}$ becomes the empirical mean $\hat{g}_{i,t}$. Since BoBW-LIL'UCB pulls the item with the largest $U_{i,t-1,\gamma}$ at time step $t$, the algorithm tends to pull the item with the largest empirical mean in this case. In other words, a large $\gamma$ encourages exploitation.

(ii) When $\gamma$ decreases, the confidence radius $C_{i,t,\gamma}$ dominates $U_{i,t,\gamma}$. Consequently, BoBW-LIL'UCB is likely to pull items with large $C_{i,t,\gamma}$, i.e., the rarely pulled items with small $N_{i,t}$. This indicates that a small $\gamma$ encourages exploration.

Altogether, we can scale $U_{i,t,\gamma}$ by adjusting $\gamma$, which allows us to balance exploitation and exploration and trade-off between the twin objectives — RM and BAI.

**Analysis for RM.** We first derive problem-dependent and problem-independent bounds on the pseudo-regret of BoBW-LIL'UCB$(\gamma)$.

**Theorem 4.1** (Bounds on the pseudo-regret of BoBW-LIL'UCB). *Let $\varepsilon \in (0,1)$, $\beta \geq 0$, and $\gamma \in (0, \min\{\log(\beta + 1 + \varepsilon)/e, 1\})$. For all $T \geq 1$, the pseudo-regret of BoBW-LIL'UCB$(\gamma)$ satisfies*

$$R_T \leq O\left(\sigma^2 \cdot \sum_{i \neq 1} \frac{\log(1/\gamma)}{\Delta_{1,i}} + 2TL\gamma^{1+\varepsilon}\right), \quad R_T \leq O\left(\sigma^2\sqrt{TL}\log\left(\frac{\log(T/L\gamma)}{\gamma}\right) + 2TL\gamma^{1+\varepsilon}\right).$$

*Furthermore, we can set $\gamma = (\log T)/T$ to obtain*

$$R_T \leq O\left(\sigma^2 \cdot \sum_{i \neq 1} \frac{\log T}{\Delta_{1,i}}\right), \quad R_T \leq O\left(\sigma^2\sqrt{TL}\log T\right).$$

We observe that the order of the problem-dependent upper bound on the pseudo-regret of BoBW-LIL'UCB$((\log T)/T)$ almost matches that of the lower bound (Lai & Robbins, 1985). Moreover, the worst-case (problem-independent) upper bound of BoBW-LIL'UCB$((\log T)/T)$ is $\tilde{O}(\sqrt{TL})$, which matches the lower bound $O(\sqrt{TL})$ (Bubeck et al., 2012) up to log factors. This implies that we can tune the parameter $\gamma$ in the BoBW-LIL'UCB$(\gamma)$ algorithm to obtain close-to-optimal performance for RM.

We remark that when the optimal item is not unique, we can also derive analogous upper bounds on the pseudo-regret of BoBW-LIL'UCB$(\gamma)$ using a similar line of analysis (see Proposition D.1).

**Analysis for BAI.** Next, we upper bound the failure probability of BoBW-LIL'UCB$(\gamma)$.

**Theorem 4.2** (Bounds on the failure probability of BoBW-LIL'UCB). *Let $\varepsilon \in (0,1)$, $\beta \geq 0$, and $\gamma \in (0, \min\{\log(\beta + 1 + \varepsilon)/e, 1\})$. Let $\Delta_i = \max\{\Delta, \Delta_{1,i}\}$ for all $i \in [L]$. For all $T \geq 1$, the failure probability of BoBW-LIL'UCB$(\gamma)$ satisfies*

$$e_T \leq \frac{2L(2+\varepsilon)}{\varepsilon}\left(\frac{\gamma}{\log(1+\varepsilon)}\right)^{1+\varepsilon}, \quad if \quad \frac{T-L}{(1+\varepsilon)^3} \geq \sum_{i=1}^{L} \frac{72\sigma^2}{\Delta_i^2} \cdot \log\left(\frac{2.8}{\gamma^2}\log\left(\frac{11\sigma(1+\varepsilon)^2}{\Delta_i} + \beta\right)\right). \quad (4.1)$$

*In particular, the bound on $e_T$ in (4.1) holds when $\gamma \geq \gamma_1(\Delta, H_2)$, where*

$$\gamma_1(\Delta, H_2) = \sqrt{2.8\log\left(\frac{6\sqrt{2.8}\sigma(1+\varepsilon)^2}{\Delta} + \beta\right)} \cdot \exp\left(-\frac{T-L}{144\sigma^2(1+\varepsilon)^3(H_2 + \Delta^{-2})}\right).$$

*For all $T \geq 1$, when $\gamma$ assumes its lower bound $\gamma_1(\Delta, H_2)$, we have*

$$e_T \leq \tilde{O}\left(L\exp\left(-\frac{T-L}{144\sigma^2(1+\varepsilon)^2(H_2 + \Delta^{-2})}\right)\right). \quad (4.2)$$

When $T \gg L$, the gap between our upper bound in (4.2) and $\Omega(\exp(-400T/(H_2 \log L)))$, the state-of-the-art lower bound (Carpentier & Locatelli, 2016), is manifested by the (pre-exponential) term $L$ as well as the constant in the exponent. This indicates that BoBW-LIL'UCB$(\gamma)$ can be adjusted to perform near-optimally for BAI over a fixed horizon.

**Further observation.** As discussed earlier, BoBW-lil'UCB($\gamma$) encourages more exploitation than exploration when $\gamma$ is large (e.g. $\gamma = (\log T)/T$) and it stimulates more exploration when $\gamma$ is small (e.g. $\gamma = \gamma_1(\Delta, H_2)$). Besides, Theorems 4.1 and 4.2 imply that the pseudo-regret of BoBW-lil'UCB($\gamma$) decreases with $\gamma$ while its failure probability increases with $\gamma$. Therefore, to minimize the regret, we should increase $\gamma$ to stimulate exploitation; and we should decrease $\gamma$ to encourage exploration for obtaining a small failure probability. This indicates that an optimal RM algorithm encourages more exploitation compared to an optimal BAI one, and vice versa.

## 5 Pareto Frontier of RM and BAI

Theorems 4.1 and 4.2 together suggest that BoBW-lil'UCB($\gamma$) cannot perform optimally for both RM and BAI simultaneously with a universal (or single) choice of $\gamma$. In this section, we prove that no algorithm can perform optimally for these two objectives simultaneously. Given a certain failure probability of an algorithm, our goal is to establish a non-trivial lower bound on its pseudo-regret.

We first consider bandit instances in which items have bounded rewards. Let $\mathcal{B}_1(\underline{\Delta}, \overline{R})$ denote the set of stochastic instances where (i) the minimal optimality gap $\Delta \geq \underline{\Delta}$; and (ii) there exists $R_0 \in \mathbb{R}$ such that the rewards are bounded in $[R_0, R_0 + \overline{R}]$. Let $\mathcal{B}_2(\underline{\Delta}, \overline{R}, \overline{H}_2)$ denote the set of instances that (i) belong to $\mathcal{B}_1(\underline{\Delta}, \overline{R})$, and (ii) have hardness quantities $H_2 \leq \overline{H}_2$.

**Theorem 5.1.** *Let $\phi_T, \underline{\Delta}, \overline{R}, \overline{H}_2 > 0$. Let $\pi$ be any algorithm with $e_T(\pi, \mathcal{I}) \leq \exp(-\phi_T)/4$ for all $\mathcal{I} \in \mathcal{B}_1(\underline{\Delta}, \overline{R})$. Then*

$$\sup_{\mathcal{I} \in \mathcal{B}_1(\underline{\Delta}, \overline{R})} R_T(\pi, \mathcal{I}) \geq \phi_T \cdot \frac{(L-1)\overline{R}}{8\underline{\Delta}}, \qquad \sup_{\mathcal{I} \in \mathcal{B}_2(\underline{\Delta}, \overline{R}, \overline{H}_2)} R_T(\pi, \mathcal{I}) \geq \phi_T \cdot \frac{\underline{\Delta}\overline{H}_2\overline{R}^3}{8}.$$

In Theorem 5.1, we apply the bounds $R_0$ and $R_0 + \overline{R}$ on items' rewards to classify instances. In general, $\Delta H_2 \leq (L-1)/\Delta$ holds for any instance, and equality holds when $\Delta_{1,i} = \Delta$ for all $i \neq 1$. Therefore, $\mathcal{B}_1(\underline{\Delta}, \overline{R}) = \mathcal{B}_2(\underline{\Delta}, \overline{R}, (L-1)/(\underline{\Delta}^2))$. When $\overline{R} > 1$, the analysis for the set $\mathcal{B}_2(\underline{\Delta}, \overline{R}, (L-1)/(\underline{\Delta}^2))$ provides a better bound (higher lower bound) for the set $\mathcal{B}_1(\underline{\Delta}, \overline{R})$.

Our **non-asymptotic** bounds complete the asymptotic observation on the trade-off between regret and BAI in Lattimore & Szepesvári (2020) and **quantify** how the trade-off depends on the hardness quantity $H_2$ and gap $\Delta_{1,i}$'s of an instance, instead of fixed constants such as $\varepsilon$. Another relevant work is Bubeck et al. (2009). On one hand, Bubeck et al. (2009) explores the trade-off between the cumulative regret $R_T$ and the simple regret $r_T$ and shows that any algorithm with $R_T \leq C\psi(T)$ satisfies $r_T \geq \underline{\Delta} \exp(-D\psi(T))/2$ in some instance. On the other hand, our work studies the Pareto frontier of $R_T$ and the BAI failure probability $e_T$. Since $e_T$ and $r_T$ satisfy that $\underline{\Delta} \cdot e_T \leq r_T \leq e_T$, our Theorem 5.1 indicates that

**Corollary 5.2.** *Let $\phi_T, \underline{\Delta} > 0$. Let $\pi$ be any algorithm satisfying*

$$\sup_{\mathcal{I} \in \mathcal{B}_1(\underline{\Delta}, 1)} R_T(\pi, \mathcal{I}) \leq \phi_T \cdot \frac{L-1}{8\underline{\Delta}},$$

*then $e_T \geq \exp(-\phi_T)/4$ and $r_T \geq \underline{\Delta} \exp(-\phi_T)/4$.*

In view of Corollary 5.2, we have precisely *quantified* that $C = (L-1)/(8\underline{\Delta})$ and $D = 1$ in the work of Bubeck et al. (2009).

Furthermore, we establish a similar analysis for instances in which the variance of each item's reward distribution is bounded. Let $\mathcal{B}'_1(\underline{\Delta}, \overline{V})$ denote the set of instances where (i) the minimal optimality gap $\Delta \geq \underline{\Delta}$; (ii) for each item $i$, the variance $\sigma_i^2 \leq \overline{V}$. Let $\mathcal{B}'_2(\underline{\Delta}, \overline{V}, \overline{H}_2)$ denote the set of instances (i) that belong to $\mathcal{B}'_1(\underline{\Delta}, \overline{V})$, and (ii) have hardness quantities $H_2 \leq \overline{H}_2$. The key difference between the proofs of these two theorems lies in the design of hard instances. We elaborate on the details in Appendix E.

**Theorem 5.3.** *Let $\phi_T, \underline{\Delta}, \overline{V}, \overline{H}_2 > 0$. Let $\pi$ be any algorithm with $e_T(\pi, \mathcal{I}) \leq \exp(-\phi_T)/4$ for all $\mathcal{I} \in \mathcal{B}'_1(\underline{\Delta}, \overline{V})$. Then*

$$\sup_{\mathcal{I} \in \mathcal{B}'_1(\underline{\Delta}, \overline{V})} R_T(\pi, \mathcal{I}) \geq \phi_T \cdot \frac{(L-1)\overline{V}}{2\underline{\Delta}}, \qquad \sup_{\mathcal{I} \in \mathcal{B}'_2(\underline{\Delta}, \overline{V}, \overline{H}_2)} R_T(\pi, \mathcal{I}) \geq \phi_T \cdot \frac{\underline{\Delta}\overline{H}_2\overline{V}}{2}.$$

By characterizing stochastic rewards with different statistics, Theorems 5.1 and 5.3 provide different lower bounds on the pseudo-regret. We observe that when the rewards of items are bounded in $[R_0, R_0 + \overline{R}]$ for some $R_0 \in \mathbb{R}$, the variances of the rewards are bounded by $\overline{R}^2/4$. Therefore,

$$\mathcal{B}_1(\underline{\Delta}, \overline{R}) \subset \mathcal{B}_1'\left(\underline{\Delta}, \frac{\overline{R}^2}{4}\right), \qquad \mathcal{B}_2(\underline{\Delta}, \overline{R}, \overline{H}_2) \subset \mathcal{B}_2'\left(\underline{\Delta}, \frac{\overline{R}^2}{4}, \overline{H}_2\right).$$

Besides, it is clear that

$$\mathcal{B}_1(\underline{\Delta}, \overline{R}), \ \mathcal{B}_2(\underline{\Delta}, \overline{R}, h), \ \mathcal{B}_2'\left(\underline{\Delta}, \frac{\overline{R}^2}{4}, h\right) \subset \mathcal{B}_1'\left(\underline{\Delta}, \frac{\overline{R}^2}{4}\right).$$

Due to the relationship among these four sets of instances, we let $\pi$ be an algorithm with $e_T(\pi, \mathcal{I}) \leq \exp(-\phi_T)/4$ in *any* instance of $\mathcal{B}_1'(\underline{\Delta}, \overline{R}^2/4)$, and compare the derived lower bounds on its pseudo-regret $R_T(\pi, \mathcal{I})$ in Table 5.1. Table 5.1 indicates that

- when the bound for $\mathcal{B}_1(\underline{\Delta}, \overline{R})$ (second column of Table 5.1) holds for $\mathcal{B}_1'(\underline{\Delta}, \overline{R}^2/4)$, the quantities $L$ and $\underline{\Delta}$ are of the same order in the bounds derived for $\mathcal{B}_1(\underline{\Delta}, \overline{R})$ and $\mathcal{B}_1'(\underline{\Delta}, \overline{R}^2/4)$ respectively;

- similarly, when the bound for $\mathcal{B}_2(\underline{\Delta}, \overline{R}, \overline{H}_2)$ (third column) holds for $\mathcal{B}_2'(\underline{\Delta}, \overline{R}^2/4, \overline{H}_2)$, the quantities $L$ and $\underline{\Delta}$ are of the same order in the bounds for $\mathcal{B}_2(\underline{\Delta}, \overline{R}, \overline{H}_2)$ and $\mathcal{B}_2'(\underline{\Delta}, \overline{R}^2/4, \overline{H}_2)$.

Table 5.1: Lower bounds on $R_T$ when $e_T \leq \mathrm{e}^{-\phi_T}/4$.

| Instance Set | $\mathcal{B}_1(\underline{\Delta}, \overline{R})$ | $\mathcal{B}_2(\underline{\Delta}, \overline{R}, \overline{H}_2)$ | $\mathcal{B}_1'(\underline{\Delta}, \overline{R}^2/4)$ | $\mathcal{B}_2'(\underline{\Delta}, \overline{R}^2/4, \overline{H}_2)$ |
|---|---|---|---|---|
| Bound on $R_T$ | $\phi_T \cdot (L-1)\overline{R}/(8\underline{\Delta})$ | $\phi_T \cdot \underline{\Delta}\overline{H}_2\overline{R}^3/8$ | $\phi_T \cdot (L-1)\overline{R}^2/(8\underline{\Delta})$ | $\phi_T \cdot \underline{\Delta}\overline{H}_2\overline{R}^2/8$ |

Moreover, when $\overline{R} > 1$, we can apply the analysis of $\mathcal{B}_2(\underline{\Delta}, \overline{R}, \overline{H}_2)$ to obtain a better bound (higher lower bound) for $\mathcal{B}_2'(\underline{\Delta}, \overline{R}^2/4, \overline{H}_2)$.

In *any* set of instances studied in Theorems 5.1 or 5.3,

- when $\phi_T$ linearly grows with $T$, which is typical in the bounds on $e_T$ (Karnin et al., 2013; Carpentier & Locatelli, 2016), the corresponding bound on $R_T$ grows linearly with $T$ (vacuous);

- when the bound on $R_T$ grows with $\log T$ as in Garivier & Cappé (2011) and Lai & Robbins (1985), $\phi_T$ grows logarithmically with $T$ (i.e., the failure probability only decays polynomially).

Thus, we cannot achieve optimal performances for both RM and BAI using any algorithm with fixed parameters. Alternatively, we can apply BoBW-lil'UCB($\gamma$) to achieve the best of both objectives with proper choices of the single parameter $\gamma$.

When the time horizon $T$ is sufficiently large, an optimal or near-optimal RM algorithm, i.e., an algoirthm that achieves the optimal regret with order $O(\log T/\Delta)$, usually pulls the optimal item $T - o(T)$ times and all the suboptimal items for $O(\log T/(\Delta^2))$ times; a good BAI algorithm usually focuses on the best and the second best items and pulls them for roughly the same number of times to distinguish between them. This implies that the proportion of item pulls in an order-wise optimal RM algorithm, such as UCB1, is vastly different from the optimal proportion of item pulls in an BAI algorithm. Hence, a bandit algorithm that is tailored to the regret minimization task, is unlikely to yield a low failure probability $e_T$, i.e., one that has a hardness parameter in the exponent close to $H_2$.

**Tightness of the upper and lower bounds.** We compare the upper and lower bounds on the pseudo-regret of BoBW-lil'UCB($\gamma$) when the horizon $T \to \infty$.

**Corollary 5.4.** *Define the interval $\mathcal{I}(\nu, T) = [\gamma_1(\underline{\Delta}, \overline{H}_2), \min\{\log(\beta + 1 + \varepsilon)/e, (\log T)/T, 1/L\}]$, which is a function of the instance $\nu$ and the fixed horizon $T$. When $\mathcal{I}(\nu, T) \neq \emptyset$, let $\pi_0$ denote the online algorithm* BoBW-lil'UCB($\gamma$) *with $\gamma$ satisfying the condition that $\gamma \in \mathcal{I}(\nu, T)$. Then*

$$\sup_{\mathcal{I} \in \mathcal{B}_2(\underline{\Delta}, 1, \overline{H}_2)} R_T(\pi_0, \mathcal{I}) \in \Omega\left(\underline{\Delta}\overline{H}_2 \log\left(\frac{1}{\gamma L}\right)\right) \bigcap O\left(\frac{L}{\underline{\Delta}} \log\left(\frac{1}{\gamma}\right)\right).$$

We observe from Corollary 5.4,[2] which combines Theorems 4.1, 4.2, and 5.1, that the gap between the upper and lower bounds depend on the term $\underline{\Delta}\overline{H}_2$ in the lower bound and $L/\underline{\Delta}$ in the upper bound. As $H_2 \leq (L-1)\Delta^{-2}$ for any instance, when $\Delta = \Delta_{1,i}$ for all $i \neq 1$ (all suboptimal items have the same suboptimality gap), equality holds, and hence the bounds match up to a small additive $\log(1/L)$ term. Corollary 5.4 implies that the parameter $\gamma$ in BOBW-LIL-UCB($\gamma$) is essential in tuning the algorithm such that it can perform optimally for either RM or BAI. This implies that in some regimes, BOBW-LIL-UCB($\gamma$) achieves Pareto-optimality up to constant or small additive (e.g., $\log(1/L)$) terms.

Besides, to show that there are cases for which $\mathcal{I}(\nu, T) \neq \emptyset$, we use several examples here. In the instance where $L = 256$, $w_1 = 0.5$, $w_i = 0.45$ for $i \neq 1$ and rewards are drawn from Bernoulli distributions, if we let $\varepsilon = 0.01$, $\beta = e$, $\mathcal{I}(\nu, T)$ is always non-empty when time horizon $T \geq 10^6$ as shown in Table 5.2.

Table 5.2: Intervals $\mathcal{I}(\nu, T)$ under different time horizons $T$

| Time horizon $T$ | $\mathcal{I}(\nu, T)$ |
|---|---|
| $10^6$ | $[1.85 \times 10^{-7}, \quad 1.38 \times 10^{-5}]$ |
| $10^7$ | $[4.33 \times 10^{-73}, \quad 1.61 \times 10^{-6}]$ |
| $10^8$ | $[0, \quad 1.84 \times 10^{-7}]$ |
| $10^9$ | $[0, \quad 2.07 \times 10^{-8}]$ |

Furthermore, Corollary 5.4 suggests that the lower bound in Theorem 5.1 is almost tight, as it is achieved by BOBW-LIL'UCB($\gamma$). Hence, up to terms logarithmic in the parameters such as $L$, we have quantified the Pareto frontier for the trade-off between RM and BAI in stochastic bandits.

# 6 Numerical Experiments

We numerically compare BOBW-LIL'UCB($\gamma$) and UCB$_\alpha$ as they are the only algorithms that can be tuned to perform (near-)optimally for both RM and BAI. Since BOBW-LIL'UCB($\gamma$) is designed for the fixed-budget setting and UCB$_\alpha$ is for the fixed-confidence setting, there cannot be a completely fair comparison between them. However, we attempt to perform fair comparisons as much as possible.

We evaluate the algorithms with both synthetic and real data. For BOBW-LIL'UCB($\gamma$), we fix $\varepsilon = 0.01$, $\beta = e$, and vary $\gamma$. For UCB$_\alpha$, we vary $\alpha$. We run BOBW-LIL'UCB($\gamma$) for $T$ (fixed in a specific instance) time steps, when the horizon (stopping time) of UCB$_\alpha$ depends on its stopping rule and the instance. Due to the difference between the fixed-horizon and fixed-confidence settings, the regrets of each algorithm may be accumulated over different time horizons.

For each choice of algorithm and instance, we run $10^4$ independent trials. Since the empirical failure probability of BOBW-LIL'UCB($\gamma$) is below 1% in each instance (see Table H.1 in Appendix H.1), we set $\delta = 0.01$ for UCB$_\alpha$, which guarantees that the failure probability of UCB$_\alpha$ is also below 1%. We also present the experiment results with empirical failure probabilities below 2% in Appendix H (where we set $\delta = 0.02$ for UCB$_\alpha$). We focus on the comparison on (i) the time horizon each algorithm runs; and (ii) the regret incurred over its corresponding horizon. We present the averages and standard deviations of the time horizons and redthe regrets of each algorithm. More numerical results that reinforce the conclusions herein are presented in Appendix H.

## 6.1 Experiments using synthetic data

We set $w_1 = 0.5$, and $w_i = 0.5 - \Delta$ for all $i \neq 1$. We let Bern($a$) denote the Bernoulli distribution with parameter $a$. We consider Bernoulli bandits, i.e., $\nu_i = $ Bern($w_i$). We display some numerical results in Figure 6.1 ; more results are postponed to Appendix H.3.

---

[2]$\overline{H}_2$ has the same units as $L \cdot \underline{\Delta}^{-2}$; hence, the terms in $\Omega(\cdot)$ and $O(\cdot)$ also have the same units.

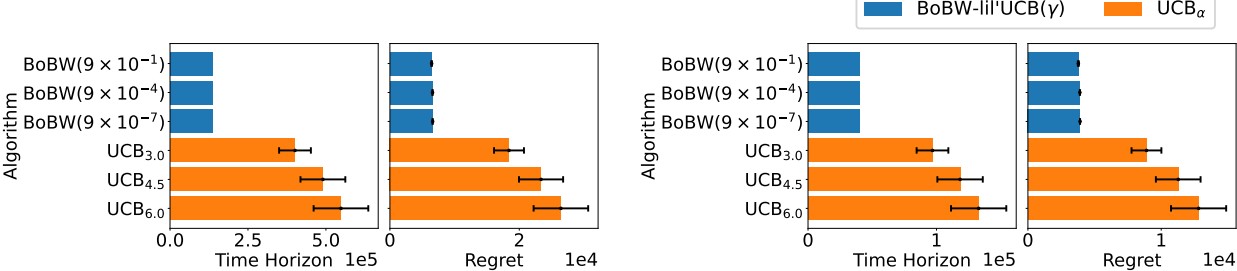

Figure 6.1: Bernoulli instances with $L\!=\!64$, failure probability $\leq 1\%$. Left: $\Delta\!=\!0.05$; right: $\Delta\!=\!0.1$.

Under each instance presented in Figure 6.1, the regret of BoBW-LIL'UCB($\gamma$) is reduced when $\gamma$ grows (see Table H.3 for exact values), which corroborates with Theorem 4.1. Both the regret and the stopping time of UCB$_\alpha$ grow with $\alpha$, which corroborates with Degenne et al. (2019, Theorem 3). Moreover, we observe that the standard deviations of the regrets are larger for UCB$_\alpha$ compared to BoBW-LIL'UCB($\gamma$), which suggests that BoBW-LIL'UCB($\gamma$) is more statistically robust and consistent in terms of the regret. Note that a larger $\Delta$ means that the difference between the optimal and suboptimal items is more pronounced, resulting in an easier instance. Given a fixed horizon $T$, our BoBW-LIL'UCB($\gamma$) algorithm outperforms the UCB$_\alpha$ algorithm with a varying range of parameters $\gamma$ and $\alpha$ in instances with different values of $\Delta$.

## 6.2 Experiments on real datasets

We use two real-world datasets, the *MovieLens 25M* (ML-25M) dataset (Harper & Konstan, 2015) and the *Published Kinase Inhibitor Set 2* (PKIS2) dataset (Drewry et al., 2017), to evaluate the performances of BoBW-LIL'UCB($\gamma$) and UCB$_\alpha$ in two types of practical applications, namely, content recommendation and drug recovery. Similarly as in Zong et al. (2016); Hong et al. (2020); Zhong et al. (2021b); Mason et al. (2020); Mukherjee et al. (2021), we generate data based on the real-world datasets.

**ML-25M dataset.** GroupLens Research provides a collection of datasets online,[3] including the ML-25M dataset. These datasets describe the rating activities from MovieLens, a movie recommendation service, and are widely used to evaluate the performances of bandit algorithms (Zong et al., 2016; Hong et al., 2020; Zhong et al., 2021b). The ML-25M dataset contains about 25 million ratings across about 62 thousand movies. We choose movies with a high number of ratings in our simulations. For each selected movie, we compute the empirical mean rating and generate random ratings according to a standard Gaussian distribution with the corresponding mean. We aim to obtain cumulatively high ratings (RM) and identify the movie with the highest rating (BAI); these are standard objectives in online recommendation systems.

**PKIS2 dataset.** This repository[4] tests 641 small molecule compounds (kinase inhibitor) against 406 protein kinases. This experiment aims to find the most effective inihibitor against a targeted kinase, and is a fundamental study in cancer drug discovery. The entries in PKIS2 indicate the *percentage control* of each inhibitor, which show the effectiveness of inhibitors and follow log-normal distributions (Christmann-Franck et al., 2016). Accordingly, we generate random variables as in Mason et al. (2020); Mukherjee et al. (2021) (see Appendix H.4 for details). We aim to find out the most effective inhibitor with the highest percentage control against one specific kinase MAPKAPK5, and also obtain high percentage controls cumulatively during the online learning process. Our study may aid in understanding how best to design experimental studies that aim to identify the most effective inhibitor in a fixed number of tests (BAI in a fixed horizon), as well as to provide effective inhibitors throughout the course of study (RM).

On the left of Figure 6.2, we report the results of the experiments on the 22 movies with at least $50,000$ ratings from the ML-25M dataset. The other plot in Figure 6.2 considers the effectiveness of 109 inhibitors test against the MAPKAPK5 kinase in the PKIS2 dataset. Both figures suggest that with high probability,

---

[3]https://grouplens.org/datasets/movielens

[4]Table 4 in https://www.biorxiv.org/content/10.1101/104711v1.supplementary-material.

Figure 6.2: Empirical failure probability $\leq 1\%$. Left: ML-25M dataset; right: PKIS2 dataset.

BoBW-LIL'UCB($\gamma$) can identify the most popular movie with the highest rating or the most effective inhibitor against MAPKAPK5 with the highest percentage control within a fixed horizon. UCB$_\alpha$ takes longer to do so, and also suffers from a larger regret. These results from the real-life datasets suggest that given a fixed horizon $T$ and a wide range of parameters, BoBW-LIL'UCB($\gamma$) outperforms UCB$_\alpha$ in these real-life instances, which demonstrates the potential of BoBW-LIL'UCB($\gamma$) in practical settings.

**Acknowledgments**

This work is supported by funding from CIFAR through Amii and NSERC and the Singapore Ministry of Education Academic Research Fund (AcRF) Tier 2 under grant numbers A-8000423-00-00 and A-8000084-01, as well as AcRF Tier 1 under grant numbers A-8000980-00-00 and A-8000189-01-00.

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

# Supplementary Material for
# "Achieving the Pareto Frontier of Regret Minimization and Best Arm Identification in Multi-Armed Bandits"

In Appendix A, we discuss the existing algorithms and relevant theoretical findings in stochastic bandits. In Appendix B, we (i) study the performance of UP-ADV (Audibert & Bubeck, 2010) and EXP3.P (Auer et al., 2002b) for both RM and BAI in adversarial bandits, and (ii) provide a lower bound on the BAI failure probability and the Pareto frontier of RM and BAI in adversarial bandits. In Appendix C, we list the useful facts that are used in the analysis. In Appendices D to G, we present detailed proofs of our theoretical results. In Appendix H, more numerical results are provided.

## A  Detailed discussion on existing algorithms

While most existing works only aim to perform either RM or BAI, Degenne et al. (2019) designed and analyzed an algorithm called $\text{UCB}_\alpha$ for both RM and BAI under the *fixed-confidence* setting. Given any $\delta$, $\text{UCB}_\alpha$ aims to minimize the number of time steps $\tau$ so that $e_\tau \leq \delta$, and, at the same time, the incurred regret $R_\tau$ can also be upper bounded. Therefore, the focus of Degenne et al. (2019) differs from that of our work. We aim to study the pseudo-regret of an algorithm which can identify the best item with high probability in a *fixed horizon $T$* in this work.

To the best of our knowledge, their is no existing work that analyzes a single algorithm for both RM and BAI under the fixed-budget setting. However, it is natural to question if an algorithm which is originally designed for RM can also perform well for BAI, and vice versa. We study some algorithms that are originally designed to achieve optimal performance for either RM or BAI.

**RM.** According to the discussions on RM and BAI in Lattimore & Szepesvári (2020) (see the second point in Note 33.3), for any algorithm with a regret that (nearly) matches the state-of-the-art lower bound (Carpentier & Locatelli, 2016):

$$\liminf_{T \to \infty} \frac{R_T(\pi)}{\log T} \geq \sum_{i \neq 1} \frac{\Delta_{1,i}}{\text{KL}(\nu_i \| \nu_1)},$$

we can construct two instances $\mathcal{I}$ and $\mathcal{I}'$ with

$$w_1^{\mathcal{I}} > w_2^{\mathcal{I}} \geq \ldots \geq w_L^{\mathcal{I}}, \qquad w_i^{\mathcal{I}'} = w_i^{\mathcal{I}} + \varepsilon(w_1^{\mathcal{I}} - w_i^{\mathcal{I}}), \text{ for some } \varepsilon > 0$$

such that

$$e_T(\pi, \mathcal{I}) + e_T(\pi, \mathcal{I}') \geq \Omega(T^{(1+o(1))(1+\varepsilon)^2}). \tag{A.1}$$

This serves as a basic observation on the limitation for BAI of an algorithm that performs (near-)optimally for RM.

**BAI.** Audibert & Bubeck (2010) were the first to explore the BAI problem under the fixed-budget setting. Carpentier & Locatelli (2016) provided a lower bound on the failure probability of any algorithm.

In the spirit of UCB1 (Auer et al., 2002a), Audibert & Bubeck (2010) designed UCB-E for BAI. We let UCB-E($a$) denote the UCB-E algorithm when it is run with parameter $a$. When $T$ is sufficiently large, we can upper bound the pseudo-regret of UCB-E($\alpha \log T$) ($\alpha \geq 2$) with a similar analysis as that for UCB1 (see Proof of Theorem 1 in Auer et al. (2002a)). Besides, we can upper bound its failure probability with Theorem 1 in Audibert & Bubeck (2010).

**Corollary A.1.** *Let $\alpha > 12.5$. Assume that $g_{i,t} \in [0,1]$ for all $i \in [L]$, and $\alpha \log T \le 25(T-L)/(36H_2)$.* UCB-E$(\alpha \log T)$ *satisfies*

$$R_T \le 2\alpha^2 \sum_{i \ne 1} \left( \frac{\log T}{\Delta_{1,i}} \right) + \left( 1 + \frac{\pi^2}{3} \right) \cdot \left( \sum_{i \ne 1} \Delta_{1,i} \right),$$

$$e_T \le 2LT^{(1 - 2\alpha/25)}.$$

When the horizon $T$ grows, Corollary A.1 indicates that the BAI failure probability of UCB-E$(\alpha \log T)$ decays only polynomially fast. In order to achieve the upper bound on $e_T$ as $\exp(-\Theta(T))$, we need to set $\alpha = O(T/\log T)$, and hence the regret bound as shown in Corollary A.1 will be $O(T^2/\log T)$, which is vacuous.

### A.1 Existing results under the fixed-budget setting of BAI

We abbreviate SEQUENTIAL REJECTS as SR, SEQUENTIAL HALVING as SH, NONLINEAR SEQUENTIAL ELIMINATION with parameter $p$ as NSE$(p)$. Besides, we simplify the bounds for algorithms which were initially analyzed for more general problems than identification of the optimal item $i^*$. we define

$$H'_p := \max_{i \ne 1} \frac{i^p}{\Delta_i^2}, \quad C_p := 2^{-p} + \sum_{i=2}^{L} i^{-p}$$

for $p > 0$ as in Shahrampour et al. (2017). We let UGAPEB$(a)$ denote the UGAPEB algorithm when it is run with parameter $a$. In Table A.1, We present existing bounds from some seminal works. The algorithms are listed in chronological order.

Table A.1: Comparison under the fixed-budget setting of BAI: upper bounds for algorithms and lower bounds in stochastic bandits.

| Algorithm/Instance | Reference | Failure probability $e_T$ |
|---|---|---|
| UCB-E$\left( \frac{25(T-L)}{36H_2} \right)$ | Audibert & Bubeck (2010) | $2TL \exp\left( -\frac{T-L}{18H_2} \right)$ |
| SR | Audibert & Bubeck (2010) | $L(L-1) \exp\left( -\frac{T-L}{(1/2 + \sum_{i=2}^{L} 1/i)H_2} \right)$ |
| UGAPEB$\left( \frac{T-L}{16H_2} \right)$ | Gabillon et al. (2012) | $2TL \exp\left( -\frac{T-L}{8H_2} \right)$ |
| SAR | Bubeck et al. (2013) | $2L^2 \exp\left( -\frac{T-L}{8(1/2 + \sum_{i=2}^{L} 1/i)H_2} \right)$ |
| SH | Karnin et al. (2013) | $3 \log_2 L \cdot \exp\left( -\frac{T}{8H_2 \log_2 L} \right)$ |
| NSE$(p)$ | Shahrampour et al. (2017) | $(L-1) \exp\left( -\frac{2(T-L)}{H'_p C_p} \right)$ |
| Stochastic Bandits | Carpentier & Locatelli (2016) | $\frac{1}{6} \exp\left( -\frac{400T}{H_2 \log L} \right)$ (Lower Bound) |

Since SH and NSE pull a number of items "uniformly" in each phase, we surmise the regret grows like $\Theta(T)$. For instance, there are $\log L$ many phases in SH and at least two items are uniformly pulled during each

phase, so at least one non-optimal item $j \neq 1$ is pulled for at least $T/(L \log L)$ times, leading to a regret at least $\Delta_{1,j} \cdot T/(L \log L)$.

As discussed in Shahrampour et al. (2017), $H'_p C_p \leq H_2 \log L$ in some special cases. Therefore, SH is better than NSE($p$) if we disregard the sub-exponential term, while NSE($p$) is better in some cases in its dependence on the exponential term. However, they are incomparable in general.

## B Conclusion and Further Discussion on Adversarial Bandits

In Sections 5 and 6, we explore the Pareto frontier of RM and BAI over a fixed horizon in stochastic bandits. The performance of our BoBW-LIL'UCB algorithm sheds light on the different emphases of RM and BAI. Moreover, we prove that no algorithm can simultaneously perform optimally for both objectives and BoBW-LIL'UCB nearly achieves the Pareto-optimality in some parameter regimes. However, as described in the discussion after Corollary 5.4, although our BoBW-LIL'UCB algorithm nearly achieves the Pareto frontier, we acknowledge that there remains a small gap $\log(1/L)$ which may be closed in the future by developing another more sophisticated algorithm.

In real-life applications, it may be unrealistic to assume i.i.d. stochastic rewards, meaning that the stochastic bandit model may not be appropriate. This brings the study of *adversarial bandits* (Auer et al., 2002b; Abbasi-Yadkori et al., 2018) to the fore. Here, the rewards of each item are not necessarily drawn independently from the same distribution. In adversarial bandits, while there exists a lower bound on the regret of any algorithm (Gerchinovitz & Lattimore, 2016), there is no lower bound on the failure probability for BAI. We fill this gap by proving a lower bound $\Omega(\exp(-150T\underline{\Delta}^2))$ in Theorem B.3, where $\underline{\Delta}$ is the minimal gap between the empirically-optimal items and the other items (see Appendix B.1 for the definitions). This bound is almost tight as it nearly matches the upper bound of UP-ADV (Abbasi-Yadkori et al., 2018).

Furthermore, there is no existing analysis of a *single* algorithm that is applicable to both RM and BAI in adversarial bandits. We fill this gap by studying the performance of EXP3.P($\gamma, \eta$) (Auer et al., 2002b) for both targets. Theorems B.1 and B.2 imply that by adjusting EXP3.P($\gamma, \eta$) with $\gamma$, we can balance between exploitation and exploration, and trade-off between the twin objectives: RM and BAI. Besides, Theorem B.4 implies that no algorithm can simultaneously perform optimally for both objectives in adversarial bandits. However, since the regret bound of EXP3.P($\gamma, \eta$) is problem-independent, we cannot ascertain if EXP3.P($\gamma, \eta$) achieves the Pareto frontier between RM and BAI. The further study of the Pareto frontier in adversarial bandits, especially the stochastically constrained adversarial bandits (Zimmert & Seldin, 2021; Wei & Luo, 2018), may serve as an interesting direction for future work.

**Outline.** In this section, we first formulate the RM and BAI problem in adversarial bandits in Appendix B.1. Next, we study the performance of UP-ADV (Audibert & Bubeck, 2010) and EXP3.P (Auer et al., 2002b) for both RM and BAI in Appendix B.2. Subsequently, we provide a lower bound on the BAI failure probability and the Pareto frontier of RM and BAI in Appendix B.3. We summarize some theoretical findings in Table B.1.

### B.1 Problem setup for adversarial bandits

In an adversarial bandit instance, we let $g_{i,t} \in [0, 1]$ be the reward of item $i$ at time $t$, and let $G_{i,t} := \sum_{u=1}^{t} g_{i,u}$ for all $1 \leq t \leq T$. We define the $\bar{\Delta}_{i,j,T}$, *empirical gap* between item $i$ and $j$ in $[L]$ and the *empirically-optimal* item $\bar{i}_T^*$ as follows:

$$\bar{\Delta}_{i,j,T} := \frac{1}{T} \cdot (G_{i,t} - G_{j,t}), \quad \bar{i}_T^* := \arg\max_{1 \leq i \leq L} G_{i,T}.$$

Moreover, we define the *empirically-minimal optimality gap* as

$$\bar{\Delta}_T := \min_{j \neq \bar{i}_T^*} \bar{\Delta}_{\bar{i}_T^*, j}.$$

We say an instance is *obliviously adversarial*[5], if $\{\boldsymbol{g}_{i,t}\}_{i,t}$ is a sequence of rewards obliviously generated by the instance before online process. We assume the empirically-optimal item $\bar{i}_T^*$ is *unique*, which implies that $\bar{\Delta}_{\min,T} > 0$.

Moreover, we define the *empirical-regret* $\bar{R}_T^\pi$ of an online algorithm $\pi$ (as defined in Section 2) as

$$\bar{R}_T^\pi := \max_{1 \leq i \leq L} \sum_{t=1}^{T} g_{i,t} - \sum_{t=1}^{T} g_{i_t^\pi,t} = G_{\bar{i}_T^*,T}^{\mathcal{I}} - \sum_{t=1}^{T} g_{i_t^\pi,t}.$$

Recall the definition of pseudo-regret $R_T(\pi)$ in Section 2: if an instance is stochastic, $\mathbb{E}\bar{R}_T^\pi = R_T^\pi$; if it is adversarial, $\mathbb{E}\bar{R}_T^\pi \leq R_T^\pi$. The aim of the agent is slightly different in stochastic and adversarial bandits:

- if the instance is stochastic, the algorithm $\pi$ aims to both minimize the pseudo-regret $R_T(\pi)$ and identify the pseudo-optimal item with high probability, i.e., to minimize $e_T(\pi) := \Pr(i_{\text{out}}^{\pi,T} \neq i_T^*)$;
- if the instance is adversarial, the algorithm $\pi$ aims to both minimize the empirical-regret $\bar{R}_T(\pi)$ and identify the empirically-optimal item with high probability, i.e., to minimize $\bar{e}_T(\pi) := \Pr(i_{\text{out}}^{\pi,T} \neq \bar{i}_T^*)$.

We omit $T$ and/or $\pi$ in the superscript or subscript when there is no cause of confusion. We write $\bar{R}_T(\pi)$ as $\bar{R}_T(\pi, \mathcal{I})$, $\bar{e}_T(\pi)$ as $\bar{e}_T(\pi, \mathcal{I})$ when we wish to emphasize their dependence on both the algorithm $\pi$ and the instance $\mathcal{I}$.

### B.2 Adversarial algorithms: UP-ADV and EXP3.P

We discuss the theoretical performances of two basic algorithms in this section.

Hence, we take the performance of this basic algorithm as a benchmark to evaluate any algorithm for this target. Besides, it is clearly that the uniform pull algorithm is the same as Exp3.P algorithm with $\beta = 0$, $\gamma = 1$. We see that UP-ADV satisfies that $\mathbb{E}\bar{R}_T \leq T$, which is consistent with Theorem B.2.

**The UP-ADV algorithm.** First of all, Abbasi-Yadkori et al. (2018) shows that a simple algorithm, which is termed as UP-ADV and chooses an item based on the uniform distribution at each time step $t$, satisfies that

$$\bar{e}_T \leq L \exp\left( - \frac{3T\bar{\Delta}_T^2}{28L} \right). \tag{B.1}$$

Abbasi-Yadkori et al. (2018) claimed that UP-ADV performs near-optimally for BAI in adversarial bandits, which is verified by our Theorem B.3 in the next section. Besides, it is obvious that UP-ADV satisfies $\mathbb{E}\bar{R}_T \leq T$.

---

**Algorithm 2** UNIFORM PULL-ADV (UP-ADV) (Abbasi-Yadkori et al., 2018)

1: **Input:** time budget $T$, size of ground set of items $L$.
2: **for** $t = 1, \ldots, T$ **do**
3:    Choose item $i_t \in [L]$ with probability $1/L$.
4:    Update the estimated cumulative gain $\tilde{G}_{i,t} = \sum_{u=1}^{t} g_{i,u} \cdot \mathbb{I}\{i_u = i\}$.
5: **end for**
6: Output $i_{\text{out}} = \arg\max_{i \in [L]} \tilde{G}_{i,T}$.

---

**The EXP3.P algorithm.** After the EXP3 algorithm and its variations were proposed by Auer et al. (2002b) for RM in adversarial bandits, this class of algorithms has been widely discussed as in Lattimore & Szepesvári (2020); Bubeck et al. (2012). We present EXP3.P$(\gamma, \eta)$ in Algorithm 3.

We first provide the upper bound on the regret of EXP3.P$(\gamma, \eta)$. The proof is similar to that in Bubeck et al. (2012) and is postponed to Appendix F.1

---

[5]When there is no ambiguity, we say an instance is adversarial to indicate that it is obliviously adversarial.

---

**Algorithm 3** EXP3.P($\gamma, \eta$) (Bubeck et al. (2012), Section 3.3, Fig. 3.1)

---

1: **Input:** time budget $T$, size of ground set of items $L$, parameters $\eta > 0$ and $\gamma \in [0, 1]$.
2: Set $p_1$ be the uniform distribution over $[L]$, i.e., $p_{i,1} = 1/L$ $\forall i \in [L]$.
3: **for** $t = 1, \ldots, T$ **do**
4:     Choose item $i_t \in [L]$ with probability $p_{i,1}$.
5:     Compute the estimated gain for each item

$$\tilde{g}_{i,t} = \frac{g_{i,t} \cdot \mathbb{I}\{i_t = i\}}{p_{i,t}}$$

    and update the estimated cumulative gain $\tilde{G}_{i,t} = \sum_{u=1}^{t} \tilde{g}_{i,u}$.
6:     Compute the new probability distribution over the items $p_{t+1} = (p_{1,t+1}, \ldots, p_{L,t+1})$ where

$$p_{i,t+1} = (1 - \gamma) \cdot \frac{\exp(\eta \tilde{G}_{i,t})}{\sum_{\ell=1}^{L} \exp(\eta \tilde{G}_{\ell,t})} + \frac{\gamma}{L}.$$

7: **end for**
8: Output $i_{\text{out}} = \arg\max_{i \in [L]} \tilde{G}_{i,T}$.

---

**Theorem B.1** (Bounds on the regret of EXP3.P($\gamma, \eta$)). *Let $\eta > 0$, $\gamma \in [0, 1/2]$ satisfying that $L\eta \leq \gamma$. Then we can upper bound the regret of* EXP3.P($\gamma, \eta$) *as follows. (i) Fix any given $\delta \in (0, 1)$, with probability at least $1 - \delta$,*

$$\bar{R}_T \leq \gamma T + \eta L T + \ln\left(\frac{L^2 T}{\eta \delta}\right) + \frac{\ln L}{\eta}.$$

*(ii) Moreover,*

$$\mathbb{E}\bar{R}_T \leq \gamma T + \eta L T + \ln\left(\frac{L^2 T}{\eta}\right) + \frac{\ln L}{\eta} + 1.$$

We observe that EXP3.P($1, \eta$) is exactly the same as UP-ADV, and the corresponding bound provided in Theorem B.2 is with the same order as in (B.1) derived by Abbasi-Yadkori et al. (2018). Our upper bound is even slightly smaller regarding the constants since we apply tighter concentration inequalities. Next, we upper bound its failure probability to identify the empirically-optimal item $\bar{i}_T^*$.

**Theorem B.2** (Bound on the failure probability of EXP3.P). *Assume $G_{1,T} \geq G_{2,T} \geq \ldots \geq G_{L,T}$. We see that the optimal item $\bar{i}_T^* = 1$. The failure probability of* EXP3.P($\gamma, \eta$) *satisfies*

$$\bar{e}_T \leq \exp\left(-\frac{\gamma T \bar{\Delta}_{1,2,T}^2}{4L}\right) + \sum_{i=2}^{L} \exp\left(-\frac{3\gamma T (\bar{\Delta}_{1,2,T}/2 + \bar{\Delta}_{2,i,T})^2}{L(3 + \bar{\Delta}_{1,2,T}/2 + \bar{\Delta}_{2,i,T})}\right) \leq L \exp\left(-\frac{\gamma T \bar{\Delta}_T^2}{4L}\right).$$

The key idea among the analysis of Theorem B.2 is to derive high-probability one-sided bounds on $\tilde{G}_{i,T} - G_{i,T}$ for all $i \in [L]$ with Theorems C.2 and C.3. The detailed proof is postponed to Appendix F.2.

Theorems B.1 and B.2 imply that by adjusting EXP3.P($\gamma, \eta$) with $\gamma$, we can balance between exploitation and exploration and trade-off between the twin objectives — RM and BAI. In detail,

- When $\gamma$ increases, the EXP3.P($\gamma, \eta$) algorithm tends to bahave more similarly to UP-ADV, which leads to a larger regret and a smaller failure probability. This indicates that a large $\gamma$ encourages exploitation.

- When $\gamma$ decreases, the EXP3.P($\gamma, \eta$) algorithm tends to emphasize more on the observation from previous time steps and pull the items with high empirically means, which leads to a smaller regret and a larger failure probability. In other words, a small $\gamma$ encourages exploitation.

### B.3 Global performances of adversarial algorithms

In this section, we first lower bound the failure probability to identify the empirically-optimal item in adversarial bandits. Next, given a certain failure probability of an algorithm, we establish a non-trivial lower bound on its empirical-regret. The proofs are in Appendix G.

We consider bandit instances in which items have bounded rewards. Let $\bar{\mathcal{B}}_1(\underline{\Delta}_T, \bar{R})$ denote the set of instances where (i) the empirically-minimal optimality gap $\bar{\Delta}_T \geq \underline{\Delta}_T$ in $T$ time steps; and (ii) there exists $R_0 \in \mathbb{R}$ such the rewards are bounded in $[R_0, R_0 + R]$. We focus on $\bar{\mathcal{B}}_1(\underline{\Delta}_T, 1)$ for brevity; the analysis can be generalized for any $\bar{\mathcal{B}}_1(\underline{\Delta}_T, \bar{R})$.

#### B.3.1 Lower bound on the BAI failure probability in adversarial bandits

**Theorem B.3.** *Let $0 < \underline{\Delta}_T \leq 1$. Then any algorithm $\pi$ satisfies that*

$$\sup_{\bar{\mathcal{B}}_1(\underline{\Delta}_T, 1)} \bar{e}_T(\pi, \mathcal{I}) \geq \frac{1 - \exp(-3T/200)}{4} \cdot \exp\left( - \frac{150 T \underline{\Delta}_T^2}{L} \right).$$

*Furthermore, when $T \geq 10$,*

$$\sup_{\bar{\mathcal{B}}_1(\underline{\Delta}_T, 1)} \bar{e}_T(\pi, \mathcal{I}) \geq \frac{2}{65} \exp\left( - \frac{150 T \underline{\Delta}_T^2}{L} \right).$$

We construct $L$ instances with clipped Gaussian distributions, which are similar to those designed for the analysis of lower bound on regret in Gerchinovitz & Lattimore (2016).

Besides, the gap between our lower bound in Theorem B.3 and the upper bounds of UP-ADV/EXP3.P$(1, \eta)$ in (B.1) and Theorem B.2 is manifested by the (pre-exponential) term $L$ as well as the constant in the exponential term. This indicates that UP-ADV/EXP3.P$(1, \eta)$ perform near-optimally for BAI and our lower bound in Theorem B.3 is almost tight.

#### B.3.2 Trade-off between RM and BAI in adversarial bandits

**Theorem B.4.** *Let $0 < \underline{\Delta}_T \leq 1$ and $T \geq 10$. Let $\pi$ be any algorithm with $\bar{e}_T(\pi, \mathcal{I}) \leq 2 \exp(-\psi_T)/65$ for all $\mathcal{I} \in \bar{\mathcal{B}}_1(\underline{\Delta}_T, 1)$. Then*

$$\sup_{\mathcal{I} \in \bar{\mathcal{B}}_1(\underline{\Delta}_T, 1)} \mathbb{E}\bar{R}_T(\pi, \mathcal{I}) \geq \psi_T \cdot \frac{L - 1}{103 \underline{\Delta}_T}.$$

Theorem B.4 implies that, as shown for the stochastic bandits (see Theorems 5.1 and 5.3), we cannot achieve optimal performances for both RM and BAI using any algorithm with fixed parameters in adversarial bandits. Besides, Theorems B.2 and B.4 indicates that

$$\sup_{\mathcal{I} \in \bar{\mathcal{B}}_1(\underline{\Delta}_T, 1)} \mathbb{E}\bar{R}_T(\text{EXP3.P}(\gamma, \eta), \mathcal{I}) \geq \left( \log\left( \frac{2}{65} \right) + \frac{\gamma T \underline{\Delta}_T^2}{4L} \right) \cdot \frac{L - 1}{103 \underline{\Delta}_T} = \Omega(\gamma T \bar{\Delta}_T).$$

However, since the upper bound on the regret of EXP3.P$(\gamma, \eta)$ in Theorem B.1 is problem-independent, we cannot ascertain if the algorithm achieves the Pareto optimality, which may serve as an interesting direction for future work. Lastly, we summarize some theoretical findings of the adversarial bandits in Table B.1.

## C  Useful facts

### C.1 Concentration

**Theorem C.1** (Non-asymptotic law of the iterated logarithm; Jamieson et al. (2014), Lemma 3). *Let $X_1, X_2, \ldots$ be i.i.d. zero-mean sub-Gaussian random variables with scale $\sigma > 0$; i.e. $\mathbb{E}[e^{\lambda X_i}] \leq \exp(\lambda^2 \sigma^2 / 2)$.*

Table B.1: Comparison among upper bounds for algorithms and lower bounds in adversarial bandits.

| Algorithm/Instance | Expected empirical-regret $\mathbb{E}\bar{R}_T$ | Failure Probability $\bar{e}_T$ |
|---|---|---|
| UP-ADV | $\Theta(T)$ | $L\exp\left(-\dfrac{3T\bar{\Delta}_T^2}{28L}\right)$ |
| | | (Abbasi-Yadkori et al., 2018) |
| EXP3.P$(\gamma,\eta)$ | $\gamma T + \eta LT + \ln\left(\dfrac{L^2 T}{\eta}\right) + \dfrac{\ln L}{\eta} + 1$ | $L\exp\left(-\dfrac{\gamma T\bar{\Delta}_T^2}{4L}\right)$ |
| | (Theorem B.1 ) | (Theorem B.2) |
| Adversarial Bandits | | $\dfrac{2}{65}\exp\left(-\dfrac{150T\bar{\Delta}_T^2}{L}\right)$ |
| | | (Lower Bound, Theorem B.3) |
| Adversarial Bandits | $\psi_T \cdot \dfrac{L-1}{103\underline{\Delta}_T}$ | $\dfrac{2}{65}\exp(-\psi_T)$ |
| | (Lower Bound, Theorem B.4 ) | (Theorem B.4) |

*For all $\varepsilon \in (0,1)$ and $\gamma \in (0, \log(1+\varepsilon)/e)$, we have*

$$\Pr\left(\forall \tau \geq 1, \frac{1}{\tau}\sum_{s=1}^{\tau} X_s \leq \sigma(1+\sqrt{\varepsilon})\sqrt{\frac{2(1+\varepsilon)}{\tau}\cdot \log\left(\frac{\log((1+\varepsilon)\tau)}{\gamma}\right)}\right) \geq 1 - \frac{2+\varepsilon}{\varepsilon}\left(\frac{\gamma}{\log(1+\varepsilon)}\right)^{1+\varepsilon}.$$

**Theorem C.2** (Chung & Lu (2006), Theorem 20). *Let $X_1, \cdots, X_n$ be a martingale adapted to filtration $\mathcal{F} = (\mathcal{F}_i)_i$ satisfying*

1. $\mathrm{Var}(X_i|\mathcal{F}_{i-1}) \leq \sigma_i^2$, *for $1 \leq i \leq n$;*

2. $X_i - X_{i-1} \leq a_i + M$, *for $1 \leq i \leq n$.*

*Then we have*

$$\Pr\left(X_n - \mathbb{E}X_n \geq \lambda\right) \leq \exp\left(-\frac{\lambda^2}{2[\sum_{i=1}^n(\sigma_i^2 + a_i^2) + M\lambda/3]}\right).$$

**Theorem C.3** (Chung & Lu (2006), Theorem 22). *Let $X_1, \cdots, X_n$ be a martingale adapted to filtration $\mathcal{F} = (\mathcal{F}_i)_i$ satisfying*

1. $\mathrm{Var}(X_i|\mathcal{F}_{i-1}) \leq \sigma_i^2$, *for $1 \leq i \leq n$;*

2. $X_{i-1} - X_i \leq a_i + M$, *for $1 \leq i \leq n$.*

*Then we have*

$$\Pr\left(X_n - \mathbb{E}X_n \leq -\lambda\right) \leq \exp\left(-\frac{\lambda^2}{2[\sum_{i=1}^n(\sigma_i^2 + a_i^2) + M\lambda/3]}\right).$$

**Theorem C.4** (Abramowitz & Stegun (1964), Formula 7.1.13; Agrawal & Goyal (2013), Lemma 6; Agrawal & Goyal (2017), Fact 4). *Let $Z \sim \mathcal{N}(\mu, \sigma^2)$. The following inequalities hold:*

$$\frac{1}{2\sqrt{\pi}}\exp\left(-\frac{7z^2}{2}\right) \leq \Pr_Z(|Z-\mu| > z\sigma) \leq \exp\left(-\frac{z^2}{2}\right) \qquad \forall z > 0,$$

$$\frac{1}{2\sqrt{\pi}z}\exp\left(-\frac{z^2}{2}\right) \leq \Pr_Z(|Z-\mu| > z\sigma) \leq \frac{\sqrt{2}}{\sqrt{\pi}z}\exp\left(-\frac{z^2}{2}\right) \qquad \forall z \geq 1.$$

**Theorem C.5** (Standard multiplicative variant of the Chernoff-Hoeffding bound; Dubhashi & Panconesi (2009), Theorem 1.1)**.** *Suppose that $X_1, \ldots, X_T$ are independent $[0,1]$-valued random variables, and let $X = \sum_{t=1}^{T} X_t$. Then for all $\varepsilon \in (0,1)$,*

$$\Pr(X - \mathbb{E}X \geq \varepsilon \mathbb{E}X) \leq \exp\left(-\frac{\varepsilon^2}{3}\mathbb{E}X\right), \ \Pr(X - \mathbb{E}X \leq -\varepsilon \mathbb{E}X) \leq \exp\left(-\frac{\varepsilon^2}{3}\mathbb{E}X\right).$$

## C.2 Change of measure

**Lemma C.6** (Tsybakov (2008), Lemma 2.6)**.** *Let $P$ and $Q$ be two probability distributions on the same measurable space. Then, for every measurable subset $A$ (whose complement we denote by $\bar{A}$),*

$$P(A) + Q(\bar{A}) \geq \frac{1}{2}\exp(-\mathrm{KL}(P \parallel Q)).$$

**Lemma C.7** (Gerchinovitz & Lattimore (2016), Lemma 1)**.** *Consider two instances 1 and 2. We let $N_{i,t}$ denote the number of pulls of item $i$ up to and including time step $t$. Under instance $j$ ($j = 1, 2$),*

- *we let $(g_{i,t}^j)_{t=1}^T$ be the sequence of rewards of item $i$ and $i_t^j$ be the pulled item at time step $t$, and let $P_{j,i}$ denote the distribution of the gain of item $i$;*
- *we assume $\{\boldsymbol{g}_t^j = (g_{1,t}^j, g_{2,t}^j, \ldots, g_{L,t}^j)\}_{t=1}^T$ is an i.i.d. sequence, i.e., $\boldsymbol{g}_{t_1}^j$ and $\boldsymbol{g}_{t_2}^j$ are i.i.d. for $t_1 \neq t_2$ but $\{g_{i,t}^j\}_{i=1}^L$ can be independent.*
- *we let $i_t^j$ be the pulled item at time step $t$, and let $\mathbb{P}_j$ denote the probability law of the process $\{\{i_t^j, g_{i_t^j,t}^j\}\}_{t=1}^T$.*

*Then, we have*

$$\mathrm{KL}(\mathbb{P}_1 \parallel \mathbb{P}_2) = \sum_{i=1}^{L} \mathbb{E}_{\mathbb{P}_1}[N_{i,T}] \cdot \mathrm{KL}(P_{1,i} \parallel P_{2,i}).$$

## C.3 KL divergence

**Theorem C.8** (Pinsker's and reverse Pinsker's inequality; Götze et al. (2019), Lemma 4.1)**.** *Let $P$ and $Q$ be two distributions that are defined in the same finite space $\mathcal{A}$ and have the same support. We have*

$$\delta(P,Q)^2 \leq \frac{1}{2}\mathrm{KL}(P,Q) \leq \frac{1}{\alpha_Q}\delta(P,Q)^2$$

*where $\delta(P,Q) = \sup\{\ |P(A) - Q(A)| \ \big| A \subset \mathcal{A}\} = \frac{1}{2}\sum_{x \in \mathcal{A}}|P(x) - Q(x)|$ is the total variational distance, and $\alpha_Q = \min_{x \in X : Q(x) > 0} Q(x)$.*

**Lemma C.9** (KL divergence between two Gaussian distributions)**.** *Let $P_1 = \mathcal{N}(\mu_1, \sigma_1^2)$, $P_2 = \mathcal{N}(\mu_2, \sigma_2^2)$. Then*

$$\mathrm{KL}(P_1 \| P_2) = \log\left(\frac{\sigma_2}{\sigma_1}\right) + \frac{\sigma_1^2 + (\mu_1 - \mu_2)^2}{2\sigma_2^2} - \frac{1}{2}.$$

**Lemma C.10** (KL divergence between clipped Gaussian distributions; Lemma 7, Gerchinovitz & Lattimore (2016))**.** *Let $Z$ be normally distributed with mean $1/2$ and variance $\sigma^2 > 0$. Let $\mathrm{clip}_{[a,b]}x := \max\{a, \min\{b, x\}\}$ for $a \leq b$. Define $X = \mathrm{clip}_{[0,1]}(Z)$ and $Y = \mathrm{clip}_{[0,1]}(Z - \varepsilon)$ for $\varepsilon \in \mathbb{R}$. Then*

$$\mathrm{KL}(P_X \parallel P_Y) \leq \frac{\varepsilon^2}{2\sigma^2}.$$

# D  Analysis of BoBW-lil'UCB($\gamma$) in stochastic bandits

**Proposition D.1** (Bounds on the pseudo-regret of BoBW-lil'UCB($\gamma$))**.** *Assume the distribution $\nu_i$ is sub-Gaussian with scale $\sigma > 0$ for all $i \in [L]$, and $w_1 \geq w_2 \geq \ldots \geq w_L$. Let $\varepsilon \in (0,1)$, $\beta \geq 0$, and $\gamma \in (0, \min\{\log(\beta + 1 + \varepsilon)/e), 1\})$. The pseudo-regret of BoBW-lil'UCB($\gamma$) satisfies*

$$R_T \leq O\left(\sigma^2(1+\varepsilon)^3 \cdot \sum_{i:\Delta_{1,i}>0} \frac{\log(1/\gamma)}{\Delta_{1,i}}\right), \quad R_T \leq O\left(\sigma^2(1+\varepsilon)^3\sqrt{TL}\log\left(\frac{\log(T/L\gamma)}{\gamma}\right)\right).$$

*Furthermore, we can set $\gamma = 1/\sqrt{T}$ to obtain*

$$R_T \leq O\left(\sigma^2(1+\varepsilon)^3 \cdot \sum_{i:\Delta_{1,i}>0} \frac{\log T}{\Delta_{1,i}}\right), \quad R_T \leq O\left(\sigma^2(1+\varepsilon)^3\sqrt{TL}\log T\right).$$

## D.1  Proof of Theorem 4.1

**Theorem 4.1** (Bounds on the pseudo-regret of BoBW-lil'UCB)**.** *Let $\varepsilon \in (0,1)$, $\beta \geq 0$, and $\gamma \in (0, \min\{\log(\beta + 1 + \varepsilon)/e, 1\})$. For all $T \geq 1$, the pseudo-regret of BoBW-lil'UCB($\gamma$) satisfies*

$$R_T \leq O\left(\sigma^2 \cdot \sum_{i\neq 1} \frac{\log(1/\gamma)}{\Delta_{1,i}} + 2TL\gamma^{1+\varepsilon}\right), \quad R_T \leq O\left(\sigma^2\sqrt{TL}\log\left(\frac{\log(T/L\gamma)}{\gamma}\right) + 2TL\gamma^{1+\varepsilon}\right).$$

*Furthermore, we can set $\gamma = (\log T)/T$ to obtain*

$$R_T \leq O\left(\sigma^2 \cdot \sum_{i\neq 1} \frac{\log T}{\Delta_{1,i}}\right), \quad R_T \leq O\left(\sigma^2\sqrt{TL}\log T\right).$$

*Proof.* Recall that we assume $w_1 > w_2 \geq \ldots \geq w_L$. Therefore, item 1 is optimal and $\Delta_{1,j} > 0$ for all $j \neq 1$.

**Step 1: Concentration.** Let $\mathcal{E}_{i,\gamma} := \{\forall t \geq L, |\hat{g}_{i,t} - w_i| \leq C_{i,t,\gamma}\}$ for all $i \in [L]$. We apply Theorem C.1 to show that $\bigcap_{i=1}^{L} \mathcal{E}_{i,\gamma}$ holds with high probability.

**Lemma D.2** (Concentration of $\hat{g}_{i,t}$)**.** *Fix any $\varepsilon \in (0,1)$ and $\gamma \in (0, \log(\beta + 1 + \varepsilon)/e)$. We have*

$$\Pr\left(\bigcap_{i=1}^{L} \mathcal{E}_{i,\gamma}\right) \geq 1 - \frac{2L(2+\varepsilon)}{\varepsilon}\left(\frac{\gamma}{\log(1+\varepsilon)}\right)^{1+\varepsilon}.$$

**Step 2: Bound on $N_{i,T}$ for $i \neq 1$.** Next, for all $t > L$, when

$$\{\hat{g}_{1,t-1} > w_1 - C_{1,t-1,\gamma}, \quad \hat{g}_{i,t-1} < w_i + C_{i,t-1,\gamma}, \quad \Delta_{1,i} > 2C_{i,t-1,\gamma}, \quad \forall i \neq 1\}$$

holds, we have

$$\{U_{1,t-1,\gamma} = \hat{g}_{1,t-1} + C_{1,t-1,\gamma} > w_1 = w_i + \Delta_{1,i} > w_i + 2C_{i,t-1,\gamma} > \hat{g}_{i,t-1} + C_{i,t-1,\gamma} = U_{i,t-1,\gamma} \quad \forall i \neq 1\},$$

which indicates $i_t = 1$. In other words, when $i_t = i \neq 1$ for $t > L$, one of the following holds:

$$\hat{g}_{1,t-1} \leq w_1 - C_{1,t-1,\gamma}, \quad \hat{g}_{i,t-1} \geq w_i + C_{i,t-1,\gamma}, \quad \Delta_{1,i} \leq 2C_{i,t-1,\gamma},$$

We see that

$$\Delta_{1,i} \leq 2C_{i,t-1,\gamma} = 10\sigma(1+\sqrt{\varepsilon})\sqrt{\frac{2(1+\varepsilon)}{N_{i,t-1}} \cdot \log\left(\frac{\log(\beta + (1+\varepsilon)N_{i,t-1})}{\gamma}\right)}$$

$$\Leftrightarrow N_{i,t-1} \leq \frac{200\sigma^2(1+\sqrt{\varepsilon})^2(1+\varepsilon)}{\Delta_{1,i}^2} \cdot \log\left(\frac{\log(\beta + (1+\varepsilon)N_{i,t-1})}{\gamma}\right).$$

In order to bound $N_{i,t-1}$, we derive the following lemma:

**Lemma D.3.** *For all $\tau > 0$, $1.4ac/\rho + b \geq e$, we have*

$$\tau \leq c \log\left(\frac{\log(a\tau + b)}{\rho}\right) \;\Rightarrow\; \tau \leq c \log\left(\frac{1.4}{\rho}\log\left(\frac{1.4ac}{\rho} + b\right)\right).$$

We apply Lemma D.3 with

$$c = \frac{200\sigma^2(1 + \sqrt{\varepsilon})^2(1 + \varepsilon)}{\Delta_{1,i}^2}, \quad a = 1 + \varepsilon, \text{ and } \rho = \gamma,$$

to obtain

$$N_{i,t-1} \leq \frac{200\sigma^2(1 + \sqrt{\varepsilon})^2(1 + \varepsilon)}{\Delta_{1,i}^2} \cdot \log\left(\frac{a_1}{\gamma}\log\left(\frac{200a_1\sigma^2(1 + \sqrt{\varepsilon})^2(1 + \varepsilon)^2}{\Delta_{1,i}^2\gamma} + \beta\right)\right) := \bar{N}_{i,\gamma}.$$

$1.4ac/\rho + b \geq e$ is satisfied when $\gamma \in (0,1)$. Therefore, when $t > L$, $\bigcap_{i=1}^{L}\mathcal{E}_{i,\gamma}$ holds and $N_{i,t-1} > \bar{N}_{i,\gamma}$ for all $i \neq 1$, we always have $i_t = 1$.

**Step 3: Conclusion.** Consequently,

$$R_T = \mathbb{E}\left[\sum_{t=1}^{T} g_{1,t} - g_{i_t,t}\right] = \mathbb{E}\left[\left(\sum_{t=1}^{T} g_{1,t} - g_{i_t,t}\right)\cdot\mathbf{1}\left(\bigcap_{i=1}^{L}\mathcal{E}_{i,\gamma}\right)\right] + \mathbb{E}\left[\left(\sum_{t=1}^{T} g_{1,t} - g_{i_t,t}\right)\cdot\mathbf{1}\left(\overline{\bigcap_{i=1}^{L}\mathcal{E}_{i,\gamma}}\right)\right]$$

$$\leq \mathbb{E}\left[\left(\sum_{t=1}^{T} g_{1,t} - g_{i_t,t}\right)\cdot\mathbf{1}\left(\bigcap_{i=1}^{L}\mathcal{E}_{i,\gamma}\right)\right] + T \cdot \Pr\left(\overline{\bigcap_{i=1}^{L}\mathcal{E}_{i,\gamma}}\right)$$

$$\leq \sum_{j\neq 1}\mathbb{E}\left[\left(\sum_{t=1}^{T} g_{1,t} - g_{i_t,t}\right)\cdot\mathbf{1}\left(i_t = j, \bigcap_{i=1}^{L}\mathcal{E}_{i,\gamma}\right)\right] + T \cdot \Pr\left(\overline{\bigcap_{i=1}^{L}\mathcal{E}_{i,\gamma}}\right)$$

$$\leq \sum_{j\neq 1}\Delta_{1,j}\cdot\mathbb{E}\left[\sum_{t=1}^{T}\cdot\mathbf{1}(i_t = j)\;\Big|\;\bigcap_{i=1}^{L}\mathcal{E}_{i,\gamma}\right] + T \cdot \Pr\left(\overline{\bigcap_{i=1}^{L}\mathcal{E}_{i,\gamma}}\right)$$

$$= \sum_{j\neq 1}\Delta_{1,j}\cdot\mathbb{E}\left[N_{j,T}\;\Big|\;\bigcap_{i=1}^{L}\mathcal{E}_{i,\gamma}\right] + T \cdot \Pr\left(\overline{\bigcap_{i=1}^{L}\mathcal{E}_{i,\gamma}}\right)$$

$$\leq \sum_{j\neq 1}\Delta_{1,j}\cdot(2 + \bar{N}_{j,\gamma}) + T \cdot \Pr\left(\overline{\bigcap_{i=1}^{L}\mathcal{E}_{i,\gamma}}\right)$$

$$= \sum_{i\neq 1} 2\Delta_{1,i} + \sum_{i\neq 1}\frac{200\sigma^2(1 + \sqrt{\varepsilon})^2(1 + \varepsilon)}{\Delta_{1,i}}\cdot\log\left(\frac{a_1}{\gamma}\log\left(\frac{200a_1\sigma^2(1 + \sqrt{\varepsilon})^2(1 + \varepsilon)^2}{\Delta_{1,i}^2\gamma} + \beta\right)\right)$$

$$+ \frac{2TL(2 + \varepsilon)}{\varepsilon}\left(\frac{\gamma}{\log(1 + \varepsilon)}\right)^{1+\varepsilon}$$

$$= \sum_{i\neq 1} 2\Delta_{1,i} + \sum_{i\neq 1}\frac{200\sigma^2(1 + \sqrt{\varepsilon})^2(1 + \varepsilon)}{\Delta_{1,i}}\cdot\log\left(\frac{2a_1}{\gamma}\log\left(\frac{10\sqrt{2a_1}\cdot\sigma(1 + \sqrt{\varepsilon})(1 + \varepsilon)}{\Delta_{1,i}\sqrt{\gamma}} + \beta\right)\right)$$

$$+ \frac{2TL(2 + \varepsilon)}{\varepsilon}\left(\frac{\gamma}{\log(1 + \varepsilon)}\right)^{1+\varepsilon}.$$

We see that $a_1 = 1.4 \leq 2$. If we divide the ground set into two classes depending on whether $\Delta_{1,i} \geq \sqrt{L/T}$, we have

$$
\begin{aligned}
R_T \leq{}& T \cdot \sqrt{\frac{L}{T}} + 2L + \frac{200L\sigma^2(1+\sqrt{\varepsilon})^2(1+\varepsilon)}{\sqrt{L/T}} \log\left(\frac{4}{\gamma}\log\left(\frac{20\sigma(1+\sqrt{\varepsilon})(1+\varepsilon)}{\sqrt{L/T}\sqrt{\gamma}} + \beta\right)\right) \\
& + \frac{2TL(2+\varepsilon)}{\varepsilon}\left(\frac{\gamma}{\log(1+\varepsilon)}\right)^{1+\varepsilon} \\
={}& \sqrt{TL} \cdot \left[1 + 200\sigma^2(1+\sqrt{\varepsilon})^2(1+\varepsilon)\log\left(\frac{4}{\gamma}\log\left(\frac{20\sigma(1+\sqrt{\varepsilon})(1+\varepsilon)}{\sqrt{\gamma L/T}} + \beta\right)\right)\right] \\
& + 2L + \frac{2TL(2+\varepsilon)}{\varepsilon}\left(\frac{\gamma}{\log(1+\varepsilon)}\right)^{1+\varepsilon}.
\end{aligned}
$$

In short, we have

$$
R_T \leq O\left(\sigma^2 \cdot \sum_{i \neq 1} \frac{\log(1/\gamma)}{\Delta_{1,i}} + 2TL\gamma^{1+\varepsilon}\right),
$$

$$
R_T \leq O\left(\sigma^2\sqrt{TL}\log\left(\frac{\log(T/L\gamma)}{\gamma}\right) + 2TL\gamma^{1+\varepsilon}\right).
$$

Let $\gamma = (\log T)/T$, we have

$$
R_T \leq O\left(\sigma^2 \cdot \sum_{i \neq 1} \frac{\log T}{\Delta_{1,i}}\right), \quad R_T \leq O\left(\sigma^2\sqrt{TL}\log T\right).
$$

$\square$

## D.2 Proof of Theorem 4.2

**Theorem 4.2** (Bounds on the failure probability of BoBW-LIL'UCB). *Let* $\varepsilon \in (0,1)$, $\beta \geq 0$, *and* $\gamma \in (0, \min\{\log(\beta + 1 + \varepsilon)/e, 1\})$. *Let* $\Delta_i = \max\{\Delta, \Delta_{1,i}\}$ *for all* $i \in [L]$. *For all* $T \geq 1$, *the failure probability of* BoBW-LIL'UCB$(\gamma)$ *satisfies*

$$
e_T \leq \frac{2L(2+\varepsilon)}{\varepsilon}\left(\frac{\gamma}{\log(1+\varepsilon)}\right)^{1+\varepsilon}, \quad \text{if} \quad \frac{T-L}{(1+\varepsilon)^3} \geq \sum_{i=1}^{L} \frac{72\sigma^2}{\Delta_i^2} \cdot \log\left(\frac{2.8}{\gamma^2}\log\left(\frac{11\sigma(1+\varepsilon)^2}{\Delta_i} + \beta\right)\right). \quad (4.1)
$$

*In particular, the bound on* $e_T$ *in (4.1) holds when* $\gamma \geq \gamma_1(\Delta, H_2)$, *where*

$$
\gamma_1(\Delta, H_2) = \sqrt{2.8\log\left(\frac{6\sqrt{2.8}\sigma(1+\varepsilon)^2}{\Delta} + \beta\right)} \cdot \exp\left(-\frac{T-L}{144\sigma^2(1+\varepsilon)^3(H_2 + \Delta^{-2})}\right).
$$

*For all* $T \geq 1$, *when* $\gamma$ *assumes its lower bound* $\gamma_1(\Delta, H_2)$, *we have*

$$
e_T \leq \tilde{O}\left(L\exp\left(-\frac{T-L}{144\sigma^2(1+\varepsilon)^2(H_2 + \Delta^{-2})}\right)\right). \quad (4.2)
$$

*Proof.* Recall that we assume $w_1 > w_2 \geq \ldots \geq w_L$. We let $\Delta_1 = w_1 - w_2$ and $\Delta_i = w_1 - w_i$ for all $i \neq 1$. Then $\Delta = \Delta_1$ and $\Delta_{1,i} = \Delta_i$ for $i \neq 1$.

**Step 1: Concentration.** Let $\mathcal{E}'_{i,\gamma} := \{\forall t \geq L, |\hat{g}_{i,t} - w_i| \leq C_{i,t,\gamma}/5\}$ for all $i \in [L]$. Similarly to Lemma D.2, we can apply Theorem C.1 to show that

$$
\Pr\left(\bigcap_{i=1}^{L} \mathcal{E}'_{i,\gamma}\right) \geq 1 - \frac{2L(2+\varepsilon)}{\varepsilon}\left(\frac{\gamma}{\log(1+\varepsilon)}\right)^{1+\varepsilon}.
$$

In the following, we prove that conditioning on the event $\{\bigcap_{i=1}^{L} \mathcal{E}'_{i,\gamma}\}$, we have $i_{\text{out}} = 1$, which concludes the proof.

We assume $\bigcap_{i=1}^{L} \mathcal{E}'_{i,\gamma}$ holds from now on. Since $i_{\text{out}}$ is the item with the largest empirical mean, we have

$$\hat{g}_{i_{\text{out}},T} \geq \hat{g}_{i,t} \quad \forall i \neq i_{\text{out}}, \quad \hat{g}_{i_{\text{out}},T} \geq w_{i_{\text{out}}} - C_{i_{\text{out}},T,\gamma}/5, \quad w_i + C_{i,T,\gamma}/5 \geq \hat{g}_{i,t} \quad \forall i \neq i_{\text{out}}.$$

Consequently, to show $i_{\text{out}} = 1$, it is sufficient to show that

$$\frac{C_{i,T,\gamma}}{5} \leq \frac{\Delta_i}{2} \iff \Delta_i \geq \frac{2C_{i,T,\gamma}}{5} = 2\sigma(1+\sqrt{\varepsilon})\sqrt{\frac{2(1+\varepsilon)}{N_{i,T}} \cdot \log\left(\frac{\log(\beta + (1+\varepsilon)N_{i,T})}{\gamma}\right)}$$

$$\iff N_{i,T} \geq \frac{8\sigma^2(1+\sqrt{\varepsilon})^2(1+\varepsilon)}{\Delta_i^2} \cdot \log\left(\frac{\log(\beta + (1+\varepsilon)N_{i,T})}{\gamma}\right) \quad \forall i \in [L]. \tag{D.1}$$

**Step 2: Upper bound $N_{i,T}$ $(i \neq 1)$.** To begin with, we let $a_1 = 2$ and prove prove by induction that

$$N_{i,t} \leq \frac{72\sigma^2(1+\sqrt{\varepsilon})^2(1+\varepsilon)}{\Delta_i^2} \cdot \log\left(\frac{a_1}{\gamma}\log\left(\frac{72a_1\sigma^2(1+\sqrt{\varepsilon})^2(1+\varepsilon)^2}{\Delta_i^2\gamma} + \beta\right)\right) + 1 \quad \forall i \neq 1. \tag{D.2}$$

Clearly, this inequality holds for all $i \neq 1$ when $1 \leq t \leq L$. Now we assume that the inequality holds for all $i \neq 1$ at time $t - 1 (t > L)$. If $i_t \neq i$, we have $N_{i,t} = N_{i,t-1}$ and the inequality still holds for $i$. Otherwise, we have $i_t = i$ and in particular $U_{i,t-1,\gamma} \geq U_{1,t-1,\gamma}$. Since

$$U_{i,t-1,\gamma} = \hat{g}_{i,t-1} + C_{i,t-1,\gamma} \leq w_i + \frac{6C_{i,t-1,\gamma}}{5}, \quad U_{1,t-1,\gamma} = \hat{g}_{1,t-1} + C_{1,t-1,\gamma} \geq w_1 + \frac{4C_{1,t-1,\gamma}}{5} \geq w_1 = w_i + \Delta_i,$$

we have

$$\frac{6C_{i,t-1,\gamma}}{5} \geq \Delta_i \iff N_{i,t-1} \leq \frac{72\sigma^2(1+\sqrt{\varepsilon})^2(1+\varepsilon)}{\Delta_i^2} \cdot \log\left(\frac{\log(\beta + (1+\varepsilon)N_{i,t-1})}{\gamma}\right)$$

$$\overset{(a)}{\Rightarrow} N_{i,t-1} \leq \frac{72\sigma^2(1+\sqrt{\varepsilon})^2(1+\varepsilon)}{\Delta_i^2} \cdot \log\left(\frac{a_1}{\gamma}\log\left(\frac{72a_1\sigma^2(1+\sqrt{\varepsilon})^2(1+\varepsilon)^2}{\Delta_i^2\gamma} + \beta\right)\right).$$

We obtain (a) using Lemma D.3 with $\gamma \in (0, 1)$:

**Lemma D.3.** *For all $\tau > 0$, $1.4ac/\rho + b \geq e$, we have*

$$\tau \leq c\log\left(\frac{\log(a\tau + b)}{\rho}\right) \Rightarrow \tau \leq c\log\left(\frac{1.4}{\rho}\log\left(\frac{1.4ac}{\rho} + b\right)\right).$$

Subsequently, by using $N_{i,t} = N_{i,t-1} + 1$, we obtain (D.2).

**Step 3: Lower bound $N_{i,T}$ $(i \neq 1)$.** Next, we again prove by induction that

$$N_{i,t} \geq 200\sigma^2(1+\varepsilon)(1+\sqrt{\varepsilon})^2 \cdot \log\left(\frac{\log(\beta + (1+\varepsilon)N_{i,t})}{\gamma}\right) \cdot \min\left\{\frac{1}{25\Delta_i^2}, \frac{1}{36(C_{1,t-1,\gamma})^2}\right\}, \quad \forall i \neq 1. \tag{D.3}$$

Clearly, this inequality holds for all $i \neq 1$ when $1 \leq t \leq L$. Now we assume that these inequalities hold for all $i \neq 1$ at time $t - 1 (t > L)$. If $i_t \neq 1$, we have

$$N_{i,t} \geq N_{i,t-1} \quad \forall i \neq 1, \quad N_{1,t} = N_{1,t-1},$$

which implies that the inequalities still hold for all $i \neq 1$. Otherwise, $i_t = 1$ indicates that $U_{1,t-1,\gamma} \geq U_{i,t-1,\gamma}$ for all $i \neq 1$. Since

$$U_{1,t-1,\gamma} = \hat{g}_{1,t-1} + C_{1,t-1,\gamma} \leq w_1 + \frac{6C_{1,t-1,\gamma}}{5}, \quad U_{i,t-1,\gamma} = \hat{g}_{i,t-1} + C_{i,t-1,\gamma} \geq w_i + \frac{4C_{i,t-1,\gamma}}{5},$$

we have

$$\frac{4C_{i,t-1,\gamma}}{5} \leq \Delta_i + \frac{6C_{1,t-1,\gamma}}{5}$$

$$\Leftrightarrow C_{i,t-1,\gamma} = 5\sigma(1+\sqrt{\varepsilon})\sqrt{\frac{2(1+\varepsilon)}{N_{i,t-1}} \cdot \log\left(\frac{\log(\beta + (1+\varepsilon)N_{i,t-1})}{\gamma}\right)} \leq \frac{5\Delta_i + 6C_{1,t-1,\gamma}}{4}$$

$$\Leftrightarrow \frac{20\sigma(1+\sqrt{\varepsilon})}{5\Delta_i + 6C_{1,t-1,\gamma}} \cdot \sqrt{\log\left(\frac{\log(\beta + (1+\varepsilon)N_{i,t-1})}{\gamma}\right)} \leq \sqrt{\frac{N_{i,t-1}}{2(1+\varepsilon)}}$$

$$\Leftrightarrow \frac{400\sigma^2(1+\sqrt{\varepsilon})^2}{(5\Delta_i + 6C_{1,t-1,\gamma})^2} \cdot \log\left(\frac{\log(\beta + (1+\varepsilon)N_{i,t-1})}{\gamma}\right) \leq \frac{N_{i,t-1}}{2(1+\varepsilon)}$$

$$\Leftrightarrow N_{i,t-1} \geq \frac{800\sigma^2(1+\varepsilon)(1+\sqrt{\varepsilon})^2}{(5\Delta_i + 6C_{1,t-1,\gamma})^2} \cdot \log\left(\frac{\log(\beta + (1+\varepsilon)N_{i,t-1})}{\gamma}\right).$$

We apply $u + v \leq 2\max\{u, v\}$ and $N_{i,t} = N_{i,t-1}$ for all $i \neq 1$ to obtain (D.3).

**Step 4: Lower bound on $N_{1,T}$.** Recall that we want to show (D.1). (i) To show (D.1) holds for all $i \neq 1$, (D.3) indicates that it is sufficiently to show that

$$\frac{200\sigma^2(1+\varepsilon)(1+\sqrt{\varepsilon})^2}{36(C_{1,T-1,\gamma})^2} \cdot \log\left(\frac{\log(\beta + (1+\varepsilon)N_{i,T})}{\gamma}\right) \geq \frac{8\sigma^2(1+\sqrt{\varepsilon})^2(1+\varepsilon)}{\Delta_i^2} \cdot \log\left(\frac{\log(\beta + (1+\varepsilon)N_{i,T})}{\gamma}\right).$$

Moreover, since $\Delta_1 = \min_{i \in [L]} \Delta_i$, it is sufficient to show

$$\frac{25}{36(C_{1,T-1,\gamma})^2} \geq \frac{1}{\Delta_1^2} \Leftrightarrow C_{1,T-1,\gamma} \leq \frac{5\Delta_1}{6}.$$

(ii) In order to show (D.1) holds for all $i \in [L]$, it is sufficient to show that

$$C_{1,T-1,\gamma} \leq \frac{5\Delta_1}{6}.$$

With $a_1 = 2$, this is implied by

$$N_{1,T-1} \geq \frac{72\sigma^2(1+\sqrt{\varepsilon})^2(1+\varepsilon)}{\Delta_i^2} \cdot \log\left(\frac{a_1}{\gamma}\log\left(\frac{72a_1\beta\sigma^2(1+\sqrt{\varepsilon})^2(1+\varepsilon)^2}{\Delta_i^2\gamma}\right)\right)$$

$$\Leftrightarrow N_{1,T} \geq \frac{72\sigma^2(1+\sqrt{\varepsilon})^2(1+\varepsilon)}{\Delta_i^2} \cdot \log\left(\frac{a_1}{\gamma}\log\left(\frac{72a_1\sigma^2(1+\sqrt{\varepsilon})^2(1+\varepsilon)^2}{\Delta_i^2\gamma} + \beta\right)\right) + 1.$$

We obtain the inequality in the above display by applying Lemma D.3. Meanwhile, (D.2) and $t = \sum_{i=1}^{L} N_{i,t}$ implies that

$$N_{1,T} = T - \sum_{i \neq 1} N_{i,T} \geq T - (L-1) - \sum_{i \neq 1} \frac{72\sigma^2(1+\sqrt{\varepsilon})^2(1+\varepsilon)}{\Delta_i^2} \cdot \log\left(\frac{a_1}{\gamma}\log\left(\frac{72a_1\sigma^2(1+\sqrt{\varepsilon})^2(1+\varepsilon)^2}{\Delta_i^2\gamma} + \beta\right)\right).$$

Altogether, we complete the proof with

$$\sum_{i=1}^{L} \frac{72\sigma^2(1+\sqrt{\varepsilon})^2(1+\varepsilon)}{\Delta_i^2} \cdot \log\left(\frac{a_1}{\gamma}\log\left(\frac{72a_1\sigma^2(1+\sqrt{\varepsilon})^2(1+\varepsilon)^2}{\Delta_i^2\gamma} + \beta\right)\right) \leq T - L + 1. \qquad (D.4)$$

**Step 5: Conclusion.** Since

$$\frac{72\sigma^2(1+\sqrt{\varepsilon})^2(1+\varepsilon)}{\Delta_i^2} \cdot \log\left(\frac{a_1}{\gamma}\log\left(\frac{72a_1\sigma^2(1+\sqrt{\varepsilon})^2(1+\varepsilon)^2}{\Delta_i^2\gamma} + \beta\right)\right)$$

$$\leq \frac{72\sigma^2(1+\varepsilon)^3}{\Delta_i^2} \cdot \log\left(\frac{2a_1}{\gamma^2}\log\left(\frac{6\sqrt{2a_1} \cdot \sigma(1+\varepsilon)^2}{\Delta_i} + \beta\right)\right),$$

To show (D.4), it is sufficient to have

$$\sum_{i=1}^{L} \frac{72\sigma^2(1+\varepsilon)^3}{\Delta_i^2} \cdot \log\left(\frac{2a_1}{\gamma^2}\log\left(\frac{6\sqrt{2a_1}\cdot\sigma(1+\varepsilon)^2}{\Delta_i}+\beta\right)\right) \leq T - L + 1$$

$$\Leftrightarrow \sum_{i=1}^{L} \frac{72\sigma^2(1+\varepsilon)^3}{\Delta_i^2} \cdot \log\left(\frac{2a_1}{\gamma^2}\right) \leq T - L + 1 - \sum_{i=1}^{L} \frac{72\sigma^2(1+\varepsilon)^3}{\Delta_i^2} \cdot \log\left(\log\left(\frac{6\sqrt{2a_1}\cdot\sigma(1+\varepsilon)^2}{\Delta_i}+\beta\right)\right)$$

$$\Leftrightarrow \gamma \geq \sqrt{2a_1} \cdot \exp\left(-\frac{T-L+1-\sum_{i=1}^{L}72\sigma^2\Delta_i^{-2}\cdot(1+\varepsilon)^3\log\left(\log\left(\frac{6\sqrt{2a_1}\cdot\sigma(1+\varepsilon)^2}{\Delta_i}+\beta\right)\right)}{\sum_{i=1}^{L}144\sigma^2(1+\varepsilon)^3\Delta_i^{-2}}\right).$$

Recall the definition of $H_2$ in (2.1):

$$H_2 = \sum_{i\neq 1} \frac{1}{\Delta_{1,i}^2}.$$

Furthermore, it suffices to have

$$\gamma \geq \sqrt{2a_1} \cdot \exp\left(-\frac{T-L}{144\sigma^2(1+\varepsilon)^3(H_2+1/\Delta_{1,2}^2)} + \frac{1}{2}\log\left(\log\left(\frac{6\sqrt{2a_1}\cdot\sigma(1+\varepsilon)^2}{\Delta_{1,2}}+\beta\right)\right)\right)$$

$$= \sqrt{2.8\cdot\log\left(\frac{6\sqrt{2.8}\cdot\sigma(1+\varepsilon)^2}{\Delta_{1,2}}+\beta\right)}\exp\left(-\frac{T-L}{144\sigma^2(1+\varepsilon)^3(H_2+1/\Delta_{1,2}^2)}\right) := \gamma_1(\Delta_{1,2}, H_2).$$

Note that $\Delta = \Delta_{1,2}$. When $\gamma = \gamma_1(\Delta, H_2)$,

$$e_T \leq \frac{2L(2+\varepsilon)}{\varepsilon}\left(\frac{\gamma_1(\Delta, H_2)}{\log(1+\varepsilon)}\right)^{1+\varepsilon}$$

$$= \frac{2L(2+\varepsilon)}{\varepsilon[\log(1+\varepsilon)]^{1+\varepsilon}} \cdot \left[2.8\log\left(\frac{6\sqrt{2.8}\cdot\sigma(1+\varepsilon)^2}{\Delta}+\beta\right)\right]^{(1+\varepsilon)/2} \cdot \exp\left(-\frac{T-L}{144\sigma^2(1+\varepsilon)^2(H_2+1/\Delta^2)}\right).$$

$\square$

## D.3 Proof of Lemma D.2

**Lemma D.2** (Concentration of $\hat{g}_{i,t}$). *Fix any $\varepsilon \in (0,1)$ and $\gamma \in (0, \log(\beta+1+\varepsilon)/e)$. We have*

$$\Pr\left(\bigcap_{i=1}^{L}\mathcal{E}_{i,\gamma}\right) \geq 1 - \frac{2L(2+\varepsilon)}{\varepsilon}\left(\frac{\gamma}{\log(1+\varepsilon)}\right)^{1+\varepsilon}.$$

*Proof.* Let

$$\mathcal{E}_{i,\gamma} := \{\forall t \geq L, |\hat{g}_{i,t} - w_i| \leq C_{i,t,\gamma}\}.$$

Then

$$\Pr(\mathcal{E}_{i,\gamma}) = \Pr\left(\forall t \geq L, |\hat{g}_{i,t} - w_i| \leq 5\sigma(1+\sqrt{\varepsilon})\sqrt{\frac{2(1+\varepsilon)}{N_{i,t}}\cdot\log\left(\frac{\log(\beta+(1+\varepsilon)N_{i,t-1})}{\gamma}\right)}\right)$$

$$= \Pr\left(\forall N_{i,t} \geq 1,\right.$$

$$\left.\left|\left(\frac{1}{N_{i,t}}\sum_{u=1}^{t}g_{i,u}\cdot\mathbb{1}\{i_u=i\}\right)-w_i\right| \leq 5\sigma(1+\sqrt{\varepsilon})\sqrt{\frac{2(1+\varepsilon)}{N_{i,t}}\cdot\log\left(\frac{\log(\beta+(1+\varepsilon)N_{i,t})}{\gamma}\right)}\right)$$

When $\varepsilon \in (0,1)$ and $\gamma \in (0, \log(\beta + 1 + \varepsilon)/e)$, Theorem C.1 indicates that

$$\Pr(\mathcal{E}_{i,\gamma}) \geq 1 - \frac{2(2+\varepsilon)}{\varepsilon} \left( \frac{\gamma}{\log(1+\varepsilon)} \right)^{1+\varepsilon}.$$

Furthermore,

$$\Pr\left( \bigcap_{i=1}^{L} \mathcal{E}_{i,\gamma} \right) = 1 - \Pr\left( \overline{\bigcap_{i=1}^{L} \mathcal{E}_{i,\gamma}} \right) = 1 - \Pr\left( \bigcup_{i=1}^{L} \overline{\mathcal{E}_{i,\gamma}} \right)$$

$$\geq 1 - \sum_{i=1}^{L} \Pr\left( \overline{\mathcal{E}_{i,\gamma}} \right) \geq 1 - \frac{2L(2+\varepsilon)}{\varepsilon} \left( \frac{\gamma}{\log(1+\varepsilon)} \right)^{1+\varepsilon}.$$

$\square$

### D.4 Proof of Lemma D.3

**Lemma D.3.** *For all $\tau > 0$, $1.4ac/\rho + b \geq e$, we have*

$$\tau \leq c\log\left( \frac{\log(a\tau + b)}{\rho} \right) \;\Rightarrow\; \tau \leq c\log\left( \frac{1.4}{\rho} \log\left( \frac{1.4ac}{\rho} + b \right) \right).$$

*Proof.* Let

$$f(\tau) = c\log\left( \frac{\log(a\tau + b)}{\rho} \right), \quad \tau_{a_1, a_2} = c\log\left( \frac{a_1}{\rho} \log\left( \frac{a_2 c}{\rho} + b \right) \right).$$

Then

$$\tau_{a_1, a_2} \geq f(\tau_{a_1, a_2}) \;\Leftrightarrow\; c\log\left( \frac{a_1}{\rho} \log\left( \frac{a_2 c}{\rho} + b \right) \right) \geq c\log\left( \frac{1}{\rho} \log\left[ ac\log\left( \frac{a_1}{\rho} \log\left( \frac{a_2 c}{\rho} \right) \right) + b \right] \right)$$

$$\Leftrightarrow\; a_1 \log\left( \frac{a_2 c}{\rho} + b \right) \geq \log\left[ ac\log\left( \frac{a_1}{\rho} \log\left( \frac{a_2 c}{\rho} \right) \right) + b \right].$$

Let $a_1 \geq 1.4$, then $x^{a_1} \geq x\log x$ for all $x \geq 1$. To obtain $\tau_{a_1, a_2} \geq f(\tau_{a_1, a_2})$, it suffices to have

$$\left( \frac{a_2 c}{\rho} + b \right) \cdot \log\left( \frac{a_2 c}{\rho} + b \right) \geq ac\log\left( \frac{a_1}{\rho} \log\left( \frac{a_2 c}{\rho} \right) \right) + b,$$

which is implied by

$$\frac{a_2 c}{\rho} + b \geq e \;\; \text{and} \;\; \frac{a_2 c}{\rho} \cdot \log\left( \frac{a_2 c}{\rho} + b \right) \geq \frac{aca_1}{\rho} \log\left( \frac{a_2 c}{\rho} \right).$$

Conditioned on $a_2 c/\rho + b \geq e$, the last inequality holds when $a_2 \geq a \cdot a_1$. Since $\tau - f(\tau)$ is monotonically increasing in $\tau$, and $\tau_{a_1, a \cdot a_1} \geq f(\tau_{a_1, a \cdot a_1})$, i.e., $\tau_{a_1, a \cdot a_1} - f(\tau_{a_1, a \cdot a_1}) \geq 0$, we have

$$\tau \geq \tau_{1.4, 1.4a} \;\Rightarrow\; \tau - f(\tau) \geq 0.$$

In other words, when $1.4ac/\rho + b \geq e$,

$$\tau \leq f(\tau) \;\Rightarrow\; \tau \leq \tau_{1.4, 2.8} = c\log\left( \frac{1.4}{\rho} \log\left( \frac{1.4ac}{\rho} \right) \right).$$

$\square$

# E  Analysis of the Pareto frontier of RM and BAI in stochastic bandits

## E.1  Proof of Theorem 5.1

**Theorem 5.1.** *Let* $\phi_T, \underline{\Delta}, \overline{R}, \overline{H}_2 > 0$. *Let* $\pi$ *be any algorithm with* $e_T(\pi, \mathcal{I}) \leq \exp(-\phi_T)/4$ *for all* $\mathcal{I} \in \mathcal{B}_1(\underline{\Delta}, \overline{R})$. *Then*

$$\sup_{\mathcal{I} \in \mathcal{B}_1(\underline{\Delta}, \overline{R})} R_T(\pi, \mathcal{I}) \geq \phi_T \cdot \frac{(L-1)\overline{R}}{8\underline{\Delta}}, \qquad \sup_{\mathcal{I} \in \mathcal{B}_2(\underline{\Delta}, \overline{R}, \overline{H}_2)} R_T(\pi, \mathcal{I}) \geq \phi_T \cdot \frac{\underline{\Delta}\,\overline{H}_2\,\overline{R}^3}{8}.$$

*Proof.* **Step 1: Construct instances.** To begin with, we fix $d_\ell \in (0, 1/4]$ for all $2 \leq \ell \leq L$. We let $\mathrm{Bern}(a)$ denote the Bernoulli distribution with parameter $a$. We define the following distributions:

$$\nu_1 := \mathrm{Bern}(1/2), \quad \nu_\ell := \mathrm{Bern}(1/2 - d_\ell) \quad \forall 1 < \ell \leq L;$$
$$\nu_1' := \mathrm{Bern}(1/2), \quad \nu_\ell' := \mathrm{Bern}(1/2 + d_\ell) \quad \forall 1 < \ell \leq L.$$

We construct $L$ instances such that under instance $\ell$ $(1 \leq \ell \leq L)$, the stochastic reward of item $i$ is drawn from distribution

$$\nu_i^\ell := b \cdot (\nu_i \mathbf{1}\{i \neq \ell\} + \nu_i' \mathbf{1}\{i = \ell\}),$$

where $b > 0$. Under instance $\ell$ $(1 \leq \ell \leq L)$, we see item $\ell$ is optimal, and we define several other notations as follows:

(i) We let $g_{i,t}^\ell$ be the random reward of item $i$ at time step $t$. Then $g_{i,t}^\ell \in \{0, b\}$.

(ii) We let $\Delta_{i,j}^\ell := \mathbb{E}[\sum_{t=1}^T g_{i,t}^\ell - g_{k,t}^\ell]/T$ denote the gap between item $i$ and $j$. Then

$$\Delta_{1,j}^1 = b \cdot d_j \quad \forall 2 \leq j \leq L, \quad \Delta_{\ell,1}^\ell = b \cdot d_\ell, \ \Delta_{\ell,j}^\ell = b \cdot d_\ell + b \cdot d_j \quad \forall 2 \leq j, \ell \leq L, j \neq \ell.$$

(iii) We denote the difficulty of the instance with

$$H_2(\ell) := \sum_{j \neq \ell} (\Delta_{\ell,j}^\ell)^{-2}.$$

Then $H_2(1) = \max\limits_{1 \leq \ell \leq L} H_2(\ell) \leq (L-1)b^{-2} \cdot \max\limits_{2 \leq \ell \leq L} d_\ell^{-2}$.

(iv) We let $i_t^\ell$ be the pulled item at time step $t$, and $O_\ell^t = \{i_u^\ell, g_{i_u^\ell, u}\}_{u=1}^t$ be the sequence of pulled items and observed rewards up to and including time step $t$.

(v) We let $\mathbb{P}_\ell^t$ be the measure on $O_\ell^t$, and let $P_{\ell,i}$ be the measure on the rewards of item $i$.

For simplicity, we abbreviate $\mathbb{P}_\ell^T$, $O_\ell^T$ as $\mathbb{P}_\ell$, $O_\ell$ respectively. Moreover, we let $N_{i,t}$ denote the number of pulls of item $i$ up to and including time step $t$.

**Step 2: Change of measure.** First of all, we apply Lemmas C.6 and C.7 to obtain that for all $1 \leq \ell \leq L$,

$$\Pr_{O_1}(i_{\mathrm{out}} \neq 1) + \Pr_{O_\ell}(i_{\mathrm{out}} = 1) \geq \frac{1}{2} \exp(-\mathrm{KL}(\mathbb{P}_1 \parallel \mathbb{P}_\ell)) = \frac{1}{2} \exp(-\mathbb{E}_{\mathbb{P}_1}[N_{\ell,T}] \cdot \mathrm{KL}(P_{1,\ell} \parallel P_{\ell,\ell})).$$

Suppose the pseudo-regret is upper bounded by $\overline{\mathrm{Reg}}$, we have

$$\begin{aligned}
\overline{\mathrm{Reg}} &\geq \mathbb{E}_{\mathbb{P}_1}\left[ \sum_{t=1}^T \mathbf{1}\{i_t^1 \neq 1\} \cdot (g_{1,t}^1 - g_{i_t,t}^1) \right] = \sum_{t=1}^T \mathbb{E}_{\mathbb{P}_1^t}[\mathbf{1}\{i_t^1 \neq 1\} \cdot (g_{1,t}^1 - g_{i_t,t}^1)] \\
&= \sum_{\ell=2}^L \sum_{t=1}^T \mathbb{E}_{\mathbb{P}_1^t}[\mathbf{1}\{i_t^1 = \ell\} \cdot (g_{1,t}^1 - g_{i_t,t}^1)] = \sum_{\ell=2}^L \sum_{t=1}^T \mathbb{E}_{\mathbb{P}_1^t}[(g_{1,t}^1 - g_{i_t,t}^1)|i_t^1 = \ell] \cdot \mathbb{E}_{\mathbb{P}_1^t}[\mathbf{1}\{i_t^1 = \ell\}] \\
&= \sum_{\ell=2}^L \sum_{t=1}^T \Delta_{1,\ell}^1 \cdot \mathbb{E}_{\mathbb{P}_1^t}[\mathbf{1}\{i_t^1 = \ell\}] = \sum_{\ell=2}^L b \cdot d_\ell \cdot \mathbb{E}_{\mathbb{P}_1}[N_{\ell,T}].
\end{aligned}$$

Since $H_2(\ell) = \sum_{j\neq\ell}(\Delta_{\ell,j}^\ell)^{-2}$, we have

$$\frac{\overline{\mathrm{Reg}}}{H_2(1)} = \frac{\sum_{\ell=2}^L b \cdot d_\ell \cdot \mathbb{E}_{\mathbb{P}_1}[N_{\ell,T}]}{\sum_{\ell=2}^L (\Delta_{1,j}^1)^{-2}} = \frac{\sum_{\ell=2}^L b \cdot d_\ell \cdot \mathbb{E}_{\mathbb{P}_1}[N_{\ell,T}]}{b^{-3} \cdot \sum_{\ell=2}^L d_j^{-2}}.$$

Thus, by the pigeonhole principle, there exists $2 \leq \ell_1 \leq L$ such that

$$b^3 d_{\ell_1}^3 \mathbb{E}_{\mathbb{P}_1} \cdot [N_{\ell_1,T}] = \frac{b \cdot d_{\ell_1} \cdot \mathbb{E}_{\mathbb{P}_1}[N_{\ell_1,T}]}{b^{-2} \cdot d_{\ell_1}^{-2}} \leq \frac{\overline{\mathrm{Reg}}}{H_2(1)} \;\Leftrightarrow\; \mathbb{E}_{\mathbb{P}_1}[N_{\ell_1,T}] \leq \frac{\overline{\mathrm{Reg}}}{b^3 d_{\ell_1}^3 H_2(1)}.$$

Since $d_\ell \in (0, 1/4]$ for all $2 \leq \ell \leq L$, we apply Theorem C.8 to obtain

$$\Pr_{O_1}(i_{\mathrm{out}} \neq 1) + \Pr_{O_{\ell_1}}(i_{\mathrm{out}} = 1) \geq \frac{1}{2}\exp(-\mathbb{E}_{\mathbb{P}_1}[N_{\ell_1,T}] \cdot \mathrm{KL}(P_{1,\ell_1} \| P_{\ell_1,\ell_1})) \geq \frac{1}{2}\exp\left(-\frac{\overline{\mathrm{Reg}}}{b^3 d_{\ell_1}^3 H_2(1)} \cdot \frac{(2d_{\ell_1})^2}{1/4}\right).$$

Since $\Pr_{O_{\ell_j}}(i_{\mathrm{out}} \neq \ell_1) \geq \Pr_{O_{\ell_1}}(i_{\mathrm{out}} = 1)$, we have

$$\max_{1 \leq \ell \leq L}\Pr_{O_\ell}(i_{\mathrm{out}} \neq \ell) \geq \frac{1}{4}\exp\left(-\frac{8\overline{\mathrm{Reg}}}{H_2(1)b^3 \cdot \min_{2\leq\ell\leq L} d_\ell}\right).$$

**Step 3: Conclusion.** We define

$$\ell_2 := \arg\max_{1 \leq \ell \leq L}\Pr_{O_\ell}(i_{\mathrm{out}} \neq \ell).$$

Suppose algorithm $\pi$ satisfies that

$$\Pr_{O_{\ell_2}}(i_{\mathrm{out}} \neq \ell_2) \leq \frac{1}{4}\exp(-\phi_T),$$

then we have

$$\overline{\mathrm{Reg}} \geq \phi_T \cdot \frac{H_2(1)b^3 \cdot \min_{2\leq\ell\leq L} d_\ell}{8}.$$

When $d = d_\ell > 0$ for all $2 \leq \ell \leq L$, we have $H_2(1) = (L-1)/(b^2 d^2)$.

**Step 4: Classification of instances.** Suppose algorithm $\pi$ satisfies that $e_T(\pi) \leq \exp(-\phi_T)/4$. Let $\mathcal{B}_1(\underline{\Delta}, \overline{R})$ denote the set of stochastic instances where (i) the minimal optimality gap $\Delta \geq \underline{\Delta}$; and (ii) there exists $R_0 \in \mathbb{R}$ such the rewards are bounded in $[R_0, R_0 + \overline{R}]$. Then

$$\sup_{\mathcal{I} \in \mathcal{B}_1(\underline{\Delta},\overline{R})} R_T(\pi, \mathcal{I}) \geq \phi_T \cdot \frac{(L-1)\overline{R}}{8\underline{\Delta}} \quad \forall \underline{\Delta}, \overline{R} > 0.$$

Let $\mathcal{B}_2(\underline{\Delta}, \overline{R}, \overline{H}_2)$ denote the set of stochastic instances that (i) belong to $\mathcal{B}_1(\underline{\Delta}, \overline{R})$, and (ii) are with hardness parameter $H_2 \leq \overline{H}_2$. Then, we have

$$\sup_{\mathcal{I} \in \mathcal{B}_2(\underline{\Delta},\overline{R},\overline{H}_2)} R_T(\pi, \mathcal{I}) \geq \phi_T \cdot \frac{\underline{\Delta}\overline{H}_2\overline{R}^3}{8} \quad \forall \underline{\Delta}, \overline{R}, \overline{H}_2 > 0.$$

$\square$

### E.2 Proof of Theorem 5.3

**Theorem 5.3.** *Let $\phi_T, \underline{\Delta}, \overline{V}, \overline{H}_2 > 0$. Let $\pi$ be any algorithm with $e_T(\pi, \mathcal{I}) \leq \exp(-\phi_T)/4$ for all $\mathcal{I} \in \mathcal{B}_1'(\underline{\Delta}, \overline{V})$. Then*

$$\sup_{\mathcal{I} \in \mathcal{B}_1'(\underline{\Delta}, \overline{V})} R_T(\pi, \mathcal{I}) \geq \phi_T \cdot \frac{(L-1)\overline{V}}{2\underline{\Delta}}, \qquad \sup_{\mathcal{I} \in \mathcal{B}_2'(\underline{\Delta}, \overline{V}, \overline{H}_2)} R_T(\pi, \mathcal{I}) \geq \phi_T \cdot \frac{\underline{\Delta}\overline{H}_2\overline{V}}{2}.$$

*Proof.* **Step 1: Construct instances.** To begin with, we fix any $\sigma > 0$, $d_\ell > 0$ for all $2 \leq \ell \leq L$. We define the following distributions:

$$\nu_1 := \mathcal{N}(1/2, \sigma^2), \quad \nu_\ell := \mathcal{N}(1/2 - d_\ell, \sigma^2) \quad \forall 1 < \ell \leq L;$$
$$\nu_1' := \mathcal{N}(1/2, \sigma^2), \quad \nu_\ell' := \mathcal{N}(1/2 + d_\ell, \sigma^2) \quad \forall 1 < \ell \leq L.$$

We construct $L$ instances such that under instance $\ell$ $(1 \leq \ell \leq L)$, the stochastic reward of item $i$ is drawn from distribution

$$\nu_i^\ell := \nu_i 1\{i \neq \ell\} + \nu_i' 1\{i = \ell\}.$$

Under instance $\ell$ $(1 \leq \ell \leq L)$, we see item $\ell$ is optimal, and we define several other notations as follows:

(i) We let $g_{i,t}^\ell$ be the random reward of item $i$ at time step $t$.

(ii) We let $\Delta_{i,j}^\ell := \mathbb{E}[\sum_{t=1}^T g_{i,t}^\ell - g_{k,t}^\ell]/T$ denote the gap between item $i$ and $j$. Then

$$\Delta_{1,j}^1 = d_j \quad \forall 2 \leq j \leq L, \quad \Delta_{\ell,1}^\ell = d_\ell, \ \Delta_{\ell,j}^\ell = d_\ell + d_j \quad \forall 2 \leq j, \ell \leq L, j \neq \ell.$$

(iii) We denote the difficulty of the instance with

$$H_2(\ell) := \sum_{j \neq \ell} (\Delta_{\ell,j}^\ell)^{-2}.$$

Then $H_2(1) = \max_{1 \leq \ell \leq L} H_2(\ell) \leq (L-1) \cdot \max_{2 \leq \ell \leq L} d_\ell^{-2}$.

(iv) We let $i_t^\ell$ be the pulled item at time step $t$, and $O_\ell^t = \{i_u^\ell, g_{i_u^\ell, u}\}_{u=1}^t$ be the sequence of pulled items and observed rewards up to and including time step $t$.

(v) We let $\mathbb{P}_\ell^t$ be the measure on $O_\ell^t$, and let $P_{\ell,i}$ be the measure on the rewards of item $i$.

For simplicity, we abbreviate $\mathbb{P}_\ell^T$, $O_\ell^T$ as $\mathbb{P}_\ell$, $O_\ell$ respectively. Moreover, we let $N_{i,t}$ denote the number of pulls of item $i$ up to and including time step $t$.

**Step 2: Change of measure.** First of all, we apply Lemmas C.6 and C.7 to obtain that for all $1 \leq \ell \leq L$,

$$\Pr_{O_1}(i_{\text{out}} \neq 1) + \Pr_{O_\ell}(i_{\text{out}} = 1) \geq \frac{1}{2} \exp(-\text{KL}(\mathbb{P}_1 \parallel \mathbb{P}_\ell)) = \frac{1}{2} \exp(-\mathbb{E}_{\mathbb{P}_1}[N_{\ell,T}] \cdot \text{KL}(P_{1,\ell} \parallel P_{\ell,\ell})).$$

Suppose the pseudo-regret is upper bounded by $\overline{\text{Reg}}$, we have

$$\overline{\text{Reg}} \geq \mathbb{E}_{\mathbb{P}_1}\left[\sum_{t=1}^T 1\{i_t^1 \neq 1\} \cdot (g_{1,t}^1 - g_{i_t,t}^1)\right] = \sum_{t=1}^T \mathbb{E}_{\mathbb{P}_1^t}[1\{i_t^1 \neq 1\} \cdot (g_{1,t}^1 - g_{i_t,t}^1)]$$

$$= \sum_{\ell=2}^L \sum_{t=1}^T \mathbb{E}_{\mathbb{P}_1^t}[1\{i_t^1 = \ell\} \cdot (g_{1,t}^1 - g_{i_t,t}^1)] = \sum_{\ell=2}^L \sum_{t=1}^T \mathbb{E}_{\mathbb{P}_1^t}[(g_{1,t}^1 - g_{i_t,t}^1)|i_t^1 = \ell] \cdot \mathbb{E}_{\mathbb{P}_1^t}[1\{i_t^1 = \ell\}]$$

$$= \sum_{\ell=2}^L \sum_{t=1}^T d_\ell \cdot \mathbb{E}_{\mathbb{P}_1^t}[1\{i_t^1 = \ell\}] = \sum_{\ell=2}^L d_\ell \cdot \mathbb{E}_{\mathbb{P}_1}[N_{\ell,T}].$$

Since $H_2(\ell) = \sum_{j \neq \ell}(\Delta_{\ell,j}^\ell)^{-2}$, we have

$$\frac{\overline{\text{Reg}}}{H_2(1)} = \frac{\sum_{\ell=2}^L d_\ell \cdot \mathbb{E}_{\mathbb{P}_1}[N_{\ell,T}]}{\sum_{\ell=2}^L (\Delta_{1,j}^1)^{-2}} = \frac{\sum_{\ell=2}^L d_\ell \cdot \mathbb{E}_{\mathbb{P}_1}[N_{\ell,T}]}{\sum_{\ell=2}^L d_j^{-2}}.$$

Thus, by the pigeonhole principle, there exists $2 \leq \ell_1 \leq L$ such that

$$d_{\ell_1}^3 \mathbb{E}_{\mathbb{P}_1} \cdot [N_{\ell_1,T}] = \frac{d_{\ell_1} \cdot \mathbb{E}_{\mathbb{P}_1}[N_{\ell_1,T}]}{d_{\ell_1}^{-2}} \leq \frac{\overline{\text{Reg}}}{H_2(1)} \;\Leftrightarrow\; \mathbb{E}_{\mathbb{P}_1}[N_{\ell_1,T}] \leq \frac{\overline{\text{Reg}}}{d_{\ell_1}^3 H_2(1)}.$$

Further, we apply Lemma C.9 to obtain

$$\Pr_{O_1}(i_{\text{out}} \neq 1) + \Pr_{O_{\ell_1}}(i_{\text{out}} = 1) \geq \frac{1}{2}\exp(-\mathbb{E}_{\mathbb{P}_1}[N_{\ell_1,T}] \cdot \text{KL}(P_{1,\ell_1} \parallel P_{\ell_1,\ell_1})) \geq \frac{1}{2}\exp\left(-\frac{\overline{\text{Reg}}}{d_{\ell_1}^3 H_2(1)} \cdot \frac{(2d_{\ell_1})^2}{2\sigma^2}\right).$$

Since $\Pr_{O_{\ell_j}}(i_{\text{out}} \neq \ell_1) \geq \Pr_{O_{\ell_1}}(i_{\text{out}} = 1)$, we have

$$\max_{1 \leq \ell \leq L} \Pr_{O_\ell}(i_{\text{out}} \neq \ell) \geq \frac{1}{4}\exp\left(-\frac{2\overline{\text{Reg}}}{H_2(1)\sigma^2 \cdot \min_{2 \leq \ell \leq L} d_\ell}\right).$$

**Step 3: Conclusion.** We define

$$\ell_2 := \arg\max_{1 \leq \ell \leq L} \Pr_{O_\ell}(i_{\text{out}} \neq \ell).$$

Suppose algorithm $\pi$ satisfies that

$$\Pr_{O_{\ell_2}}(i_{\text{out}} \neq \ell_2) \leq \frac{1}{4}\exp(-\phi_T),$$

then we have

$$\overline{\text{Reg}} \geq \phi_T \cdot \frac{H_2(1)\sigma^2 \cdot \min_{2 \leq \ell \leq L} d_\ell}{2}.$$

When $d = d_\ell > 0$ for all $2 \leq \ell \leq L$, we have $H_2(1) = (L-1)/d^2$.

**Step 4: Classification of instances.** Suppose algorithm $\pi$ satisfies that $e_T(\pi) \leq \exp(-\phi_T)/4$. Let $\mathcal{B}_1'(\underline{\Delta}, \overline{V})$ denote the set of stochastic instances where (i) the minimal optimality gap $\Delta \geq \underline{\Delta}$; (ii) for each item $i$, the variance $\sigma_i^2 \leq \overline{V}$. Then

$$\sup_{\mathcal{I} \in \mathcal{B}_1'(\underline{\Delta}, \overline{V})} R_T(\pi, \mathcal{I}) \geq \phi_T \cdot \frac{(L-1)\overline{V}}{2\underline{\Delta}} \quad \forall \underline{\Delta}, \overline{V} > 0.$$

Let $\mathcal{B}_2'(\underline{\Delta}, \overline{V}, \overline{H}_2)$ denote the set of stochastic instances (i) that belong to $\mathcal{B}_1'(\underline{\Delta}, \overline{V})$, and (ii) are with the hardness $H_2 \leq \overline{H}_2$. We have

$$\sup_{\mathcal{I} \in \mathcal{B}_2'(\underline{\Delta}, \overline{V}, \overline{H}_2)} R_T(\pi, \mathcal{I}) \geq \phi_T \cdot \frac{\underline{\Delta}\overline{H}_2\overline{V}}{2} \quad \forall \underline{\Delta}, \overline{V}, \overline{H}_2 > 0.$$

$\square$

## E.3 Proof of Corollary 5.4

**Corollary 5.4.** *Define the interval* $\mathcal{I}(\nu, T) = [\gamma_1(\underline{\Delta}, \overline{H}_2), \min\{\log(\beta + 1 + \varepsilon)/e, (\log T)/T, 1/L\}]$, *which is a function of the instance* $\nu$ *and the fixed horizon* $T$. *When* $\mathcal{I}(\nu, T) \neq \emptyset$, *let* $\pi_0$ *denote the online algorithm* $\text{BOBW-LIL'UCB}(\gamma)$ *with* $\gamma$ *satisfying the condition that* $\gamma \in \mathcal{I}(\nu, T)$. *Then*

$$\sup_{\mathcal{I} \in \mathcal{B}_2(\underline{\Delta}, 1, \overline{H}_2)} R_T(\pi_0, \mathcal{I}) \in \Omega\left(\underline{\Delta}\overline{H}_2 \log\left(\frac{1}{\gamma L}\right)\right) \bigcap O\left(\frac{L}{\underline{\Delta}}\log\left(\frac{1}{\gamma}\right)\right).$$

*Proof.* We consider the stochastic instances in $\mathcal{B}_2(\underline{\Delta}, 1, \overline{H}_2)$. By the classification of instances in Theorem 5.1, these instances satisfy the conditions

$$g_{i,t} \in [0,1] \quad \forall i, t, \quad \text{and} \quad H_2 \leq \overline{H}_2.$$

Therefore, the distribution $\nu_i$ is sub-Gaussian with scale $\sigma = 1/2$ for all $i \in [L]$. We assume $T$ is sufficiently large such that

$$\frac{\log T}{T} \geq \gamma_1 = \sqrt{2.8 \log \left( \frac{6\sqrt{2.8}\sigma(1+\varepsilon)^2}{\underline{\Delta}} + \beta \right) \cdot \exp \left( - \frac{T - L}{144\sigma^2(1+\varepsilon)^3(\overline{H}_2 + 1/\underline{\Delta}^2)} \right)}$$

$$= \sqrt{2.8 \log \left( \frac{3\sqrt{2.8}(1+\varepsilon)^2}{\underline{\Delta}} + \beta \right) \cdot \exp \left( - \frac{T - L}{36(1+\varepsilon)^3(\overline{H}_2 + \underline{\Delta}^{-2})} \right)}.$$

As a result, for all instance in $\mathcal{B}_2(\underline{\Delta}, 1, \overline{H}_2)$, since $\Delta \geq \underline{\Delta}$ and $H_2 \leq \overline{H}_2$, we have

$$\frac{\log T}{T} \geq \sqrt{2.8 \log \left( \frac{3\sqrt{2.8}(1+\varepsilon)^2}{\Delta} + \beta \right) \cdot \exp \left( - \frac{T - L}{36(1+\varepsilon)^3(H_2 + \Delta^{-2})} \right)}.$$

Fix any $\gamma \in [\gamma_1, (\log T)/L]$. On one hand, for any instance in $\mathcal{B}_2(\underline{\Delta}, 1, \overline{H}_2)$, Theorem 4.1 implies that BoBW-LIL'UCB($\gamma$) satisfies that

$$R_T \leq O\left( \log \left( \frac{1}{\gamma} \right) \cdot H_1 \right).$$

On the other hand, Theorem 4.2 implies that

$$e_T \leq \frac{2L(2+\varepsilon)}{\varepsilon} \left( \frac{\gamma}{\log(1+\varepsilon)} \right)^{1+\varepsilon}.$$

Moreover, we can apply Theorem 5.1 to obtain that

$$\sup_{\mathcal{I} \in \mathcal{B}_2(\underline{\Delta}, 1, \overline{H}_2)} R_T(\text{BoBW-LIL'UCB}(\gamma), \mathcal{I}) \in \Omega\left( \underline{\Delta}\overline{H}_2 \log \left( \frac{1}{\gamma L} \right) \right).$$

Altogether, we have

$$\sup_{\mathcal{I} \in \mathcal{B}_2(\underline{\Delta}, 1, \overline{H}_2)} R_T(\text{BoBW-LIL'UCB}(\gamma), \mathcal{I}) \in \Omega\left( \underline{\Delta}\overline{H}_2 \log \left( \frac{1}{\gamma L} \right) \right) \bigcap O\left( \frac{(L-1)}{\underline{\Delta}} \log \left( \frac{1}{\gamma} \right) \right).$$

$\square$

# F  Analysis of Exp3.P in adversarial bandits

## F.1  Proof of Theorem B.1

**Theorem B.1** (Bounds on the regret of Exp3.P($\gamma, \eta$))**.** *Let $\eta > 0$, $\gamma \in [0, 1/2]$ satisfying that $L\eta \leq \gamma$. Then we can upper bound the regret of Exp3.P($\gamma, \eta$) as follows. (i) Fix any given $\delta \in (0, 1)$, with probability at least $1 - \delta$,*

$$\bar{R}_T \leq \gamma T + \eta LT + \ln \left( \frac{L^2 T}{\eta \delta} \right) + \frac{\ln L}{\eta}.$$

*(ii) Moreover,*

$$\mathbb{E}\bar{R}_T \leq \gamma T + \eta LT + \ln \left( \frac{L^2 T}{\eta} \right) + \frac{\ln L}{\eta} + 1.$$

*Proof.* The analysis is similar to that of Theorem 3.2 in Bubeck et al. (2012) with $\beta = 0$, while the following lemma signifies the key difference.

**Lemma F.1** (Implied by Bubeck et al. (2012), Lemma 3.1). *For any item $i$, with probability at least $1 - \delta$,*

$$\sum_{t=1}^{T} g_{i,t} \leq \sum_{t=1}^{T} \tilde{g}_{i,t} + \ln\left(\frac{LT}{\eta\delta}\right).$$

Replacing Lemma 3.1 in Bubeck et al. (2012) by Lemma F.1, we can adopt the analysis of Theorem 3.2 in Bubeck et al. (2012) and show that with probability $1 - \delta$,

$$\bar{R}_T \leq \gamma T + \eta LT + \ln\left(\frac{L^2 T}{\eta\delta}\right) + \frac{\ln L}{\eta}.$$

Moreover, we derive that

$$W' = \bar{R}_T - \left[\gamma T + \eta LT + \ln\left(\frac{L^2 T}{\eta}\right) + \frac{\ln L}{\eta}\right],$$

$$\mathbb{P}(W' > \ln\frac{1}{\delta}) = \Pr\left(\bar{R}_T - \left[\gamma T + \eta LT + \ln\left(\frac{L^2 T}{\eta\delta}\right) + \frac{\ln L}{\eta}\right] > 0\right) \leq \delta,$$

$$\mathbb{E}\bar{R}_T - \left[\gamma T + \eta LT + \ln\left(\frac{L^2 T}{\eta}\right) + \frac{\ln L}{\eta}\right] \leq 1,$$

$$\mathbb{E}\bar{R}_T \leq \gamma T + \eta LT + \ln\left(\frac{L^2 T}{\eta}\right) + \frac{\ln L}{\eta} + 1.$$

$\square$

## F.2   Proof of Theorem B.2

**Theorem B.2** (Bound on the failure probability of Exp3.P). *Assume $G_{1,T} \geq G_{2,T} \geq \ldots \geq G_{L,T}$. We see that the optimal item $\bar{i}_T^* = 1$. The failure probability of Exp3.P$(\gamma, \eta)$ satisfies*

$$\bar{e}_T \leq \exp\left(-\frac{\gamma T \bar{\Delta}_{1,2,T}^2}{4L}\right) + \sum_{i=2}^{L} \exp\left(-\frac{3\gamma T (\bar{\Delta}_{1,2,T}/2 + \bar{\Delta}_{2,i,T})^2}{L(3 + \bar{\Delta}_{1,2,T}/2 + \bar{\Delta}_{2,i,T})}\right) \leq L \exp\left(-\frac{\gamma T \bar{\Delta}_T^2}{4L}\right).$$

*Proof.* For brevity, we assume $G_{1,T} \geq G_{2,T} \geq \ldots \geq G_{L,T}$ and abbreviate $\bar{\Delta}_{i,j,T}$ as $\Delta_{i,j}$ for any $i, j \in [L]$. Consequently, the optimal item $\bar{i}^* = 1$.

**Step 1: Construction of martingale.** Let

$$y_{i,t} = \tilde{g}_{i,t} - g_{i,t}, \quad X_{i,t} = \tilde{G}_{i,t} - G_{i,t} = \sum_{u=1}^{t}(\tilde{g}_{i,u} - g_{i,u}) = \sum_{u=1}^{t} y_{i,u}.$$

Now we fix arbitrary $i \in [L]$ and abbreviate $y_{i,t}$ as $y_t$, $X_{i,t}$ as $X_t$ for brevity when there is no ambiguity. Then we have

$$X_t - X_{t-1} = y_t = \tilde{g}_{i,t} - g_{i,t} = g_{i,t} \cdot \left(\frac{\mathbb{I}\{i_t = i\}}{p_{i,t}} - 1\right) = g_{i,t} \cdot \left(\frac{\mathbb{I}\{i_t = i\}}{p_{i,t}} - 1\right),$$

$$-1 \leq y_t \leq \frac{1}{p_{i,t}} - 1 \quad \text{since } g_{i,t} \leq 1,$$

$$\mathbb{E}[y_t|\mathcal{F}_{t-1}] = \mathbb{E}\left[g_{i,t} \cdot \left(\frac{\mathbb{E}[\mathbb{I}\{i_t = i\}|\mathcal{F}_{t-1}]}{p_{i,t}} - 1\right)\Big|\mathcal{F}_{t-1}\right] = 0, \quad \mathbb{E}y_t = \mathbb{E}[\mathbb{E}[y_t|\mathcal{F}_{t-1}]] = 0.$$

Since $y_t$ is a martingale, we can apply Theorem C.2 and C.3 for the analysis.

Meanwhile, note that $g_{i,t}, p_{i,t} \in \mathcal{F}_{t-1}$ and again $g_{i,t} \leq 1$. Since $\mathbb{P}(i_t = i | \mathcal{F}_{t-1}) = p_{i,t}$, the variance conditioned on $\mathcal{F}_{t-1}$ is the variance of the Bernoulli random variable with parameter $p_{i,t}$, scaled to the range $[0, g_{i,t}/p_{i,t}]$. Hence, we have

$$\mathrm{Var}(X_t | \mathcal{F}_{t-1}) = \mathrm{Var}(y_t | \mathcal{F}_{t-1}) = \mathrm{Var}\left( \frac{g_{i,t} \cdot (\mathbb{I}\{i_t = i\} - 1)}{p_{i,t}} \,\Big|\, \mathcal{F}_{t-1} \right) = \mathrm{Var}\left( \frac{g_{i,t} \cdot \mathbb{I}\{i_t = i\}}{p_{i,t}} \,\Big|\, \mathcal{F}_{t-1} \right)$$

$$= \mathrm{Var}\left( \frac{g_{i,t} \cdot \mathbb{I}\{i_t = i\}}{p_{i,t}} \,\Big|\, \mathcal{F}_{t-1} \right) = \frac{g_{i,t}^2}{p_{i,t}^2} \cdot \mathrm{Var}(\mathbb{I}\{i_t = i\} | \mathcal{F}_{t-1}) = \frac{p_{i,t}(1 - p_{i,t})g_{i,t}^2}{p_{i,t}^2} = \frac{(1 - p_{i,t})g_{i,t}^2}{p_{i,t}}$$

$$= g_{i,t}^2 \cdot \left( \frac{1}{p_{i,t}} - 1 \right) \leq \frac{1}{p_{i,t}} - 1 := \sigma_t^2 \qquad \text{since } g_{i,t} \leq 1.$$

On one hand, in order to apply Theorem C.2 to upper bound $\tilde{G}_{i,T} - G_{i,T}$, we need to upper bound $\mathrm{Var}(X_t | \mathcal{F}_{t-1})$, $y_t$, and lower bound $p_{i,t}$. These bounds will depend on the lower bound on $p_{i,t}$. On the other hand, to lower bound $\tilde{G}_{i,T} - G_{i,T}$ with Theorem C.3, we need to upper bound $\mathrm{Var}(X_t | \mathcal{F}_{t-1})$, $p_{i,t}$, and lower bound $y_t$. This motivates to derive bounds on $p_{i,t}$. Since $1 - \gamma + \frac{\gamma}{L} - \frac{1}{L} = (1 - \frac{1}{L})(1 - \gamma) > 0$, there are global bounds on $\{p_{i,t}\}_{i,t}$:

$$\frac{\gamma}{L} \leq p_{i,t} \leq 1 - \gamma + \frac{\gamma}{L}.$$

**Step 2: Bound $\tilde{G}_{i,T} - G_{i,T}$ with high probability.**

**Upper bound on $\tilde{G}_{i,T} - G_{i,T}$.** We first derive upper bounds on $\sum_{t=1}^T \sigma^2$, $y_t$ with lower bounds on $p_{i,t}$:

$$\text{(i)} \quad \sum_{t=1}^T \sigma_t^2 = \sum_{t=1}^T \left( \frac{1}{p_{i,t}} - 1 \right) \leq T \cdot \left( \frac{L}{\gamma} - 1 \right)$$

$$\text{(ii)} \quad y_t \leq \frac{1 - \beta}{p_{i,t}} - 1 \leq \frac{L}{\gamma} - 1 := M.$$

Let $a_t = 0$. We apply Theorem C.2. For all $\lambda \in (0, 1)$,

$$\Pr(X_T - \mathbb{E}X_T \geq \lambda) \leq \exp\left( -\frac{\lambda^2}{\sum_{t=1}^T \sigma_t^2 + M\lambda/3} \right), \qquad \mathbb{E}X_T = \sum_{t=1}^T \mathbb{E}y_t = 0.$$

Therefore, for all $i \in [L]$, $\lambda_i \in (0, 1)$,

$$\mathbb{P}(\tilde{G}_{i,T} - G_{i,T} \geq \lambda_i) \leq \exp\left( -\frac{\lambda_i^2}{\sum_{t=1}^T \sigma_t^2 + M\lambda_i/3} \right). \tag{F.1}$$

**Lower bound on $\tilde{G}_{i,T} - G_{i,T}$.** Similarly, we have $X_{t-1} - X_t = -y_t \leq 1 := M'$. We apply Theorem C.3. Therefore, for all $i \in [L]$, $\lambda_i \in (0, 1)$, we have

$$\Pr(\tilde{G}_{i,T} - G_{i,T} \leq -\lambda_i) \leq \exp\left( -\frac{\lambda_i^2}{\sum_{t=1}^T \sigma_t^2 + M'\lambda_i/3} \right). \tag{F.2}$$

**Step 3: Last step.** We decompose the failure probability as follows:

**Lemma F.2.** *For any fixed time budget $T$, we have*

$$\Pr(i_{\mathrm{out}} \neq 1) \leq \Pr\left( \tilde{G}_{1,T} - G_{1,T} \leq -\frac{T\Delta_{1,2}}{2} \right) + \sum_{i=2}^L \Pr\left( \tilde{G}_{i,T} - G_{i,T} \geq \frac{T\Delta_{1,2}}{2} + T \cdot \Delta_{2,i} \right).$$

Combining (F.1), (F.2) and Lemma F.2 , we see that

$$\lambda_1 \leq \frac{T\Delta_{1,2}}{2}, \quad \lambda_i \leq \frac{T\Delta_{1,2}}{2} + T \cdot \Delta_{2,i} \quad \forall i \neq 1,$$

$$\Rightarrow \ \Pr(i_{\text{out}} \neq 1) \leq \exp\left( - \frac{\lambda_1^2}{\sum_{t=1}^T \sigma_t^2 + M'\lambda_1/3} \right) + \sum_{i=2}^L \exp\left( - \frac{\lambda_i^2}{\sum_{t=1}^T \sigma_t^2 + M\lambda_i/3} \right).$$

Let

$$\lambda_1 := \frac{T\Delta_{1,2}}{2} \qquad \lambda_i := \frac{T\Delta_{1,2}}{2} + T \cdot \Delta_{2,i} \quad \forall i \neq 1.$$

We now complete the proof:

$$\Pr(i_{\text{out}} \neq 1) \leq \exp\left( - \frac{T\Delta_{1,2}^2/4}{L/\gamma} \right) + \sum_{i=2}^L \exp\left( - \frac{T(\Delta_{1,2}/2 + \Delta_{2,i})^2}{L(3 + \Delta_{1,2}/2 + \Delta_{2,i})/(3\gamma)} \right)$$

$$= \exp\left( - \frac{\gamma T\Delta_{1,2}^2}{4L} \right) + \sum_{i=2}^L \exp\left( - \frac{3\gamma T(\Delta_{1,2}/2 + \Delta_{2,i})^2}{L(3 + \Delta_{1,2}/2 + \Delta_{2,i})} \right).$$

$\square$

### F.3   Proof of Lemma F.2

**Lemma F.2.** *For any fixed time budget $T$, we have*

$$\Pr(i_{\text{out}} \neq 1) \leq \Pr\left( \tilde{G}_{1,T} - G_{1,T} \leq - \frac{T\Delta_{1,2}}{2} \right) + \sum_{i=2}^L \Pr\left( \tilde{G}_{i,T} - G_{i,T} \geq \frac{T\Delta_{1,2}}{2} + T \cdot \Delta_{2,i} \right).$$

*Proof.* We observe that

$$\Pr(i_{\text{out}} \neq 1) = \Pr(\exists i \neq 1 : \tilde{G}_{i,T} \geq \tilde{G}_{1,T})$$

$$\leq \Pr\left( \exists i \neq 1 : \tilde{G}_{i,T} - G_{i,T} \geq \frac{T\Delta_{1,2}}{2} + T \cdot \Delta_{2,i}, \text{ or } \tilde{G}_{1,T} - G_{1,T} \leq - \frac{T\Delta_{1,2}}{2} \right) \qquad \text{(F.3)}$$

$$\leq \Pr\left( \tilde{G}_{1,T} - G_{1,T} \leq - \frac{T\Delta_{1,2}}{2} \right) + \sum_{i=2}^L \Pr\left( \tilde{G}_{i,T} - G_{i,T} \geq \frac{T\Delta_{1,2}}{2} + T \cdot \Delta_{2,i} \right). \qquad \text{(F.4)}$$

It is trivial to obtain (F.4) with (F.3). Now we complete the proof with the derivation of (F.3). We denote

$$\mathcal{E}_{1,T} := \left\{ \tilde{G}_{1,T} - G_{1,T} \leq - \frac{T\Delta_{1,2}}{2} \right\}, \qquad \mathcal{E}_{i,T} := \left\{ \tilde{G}_{i,T} - G_{i,T} \geq \frac{T\Delta_{1,2}}{2} + T \cdot \Delta_{2,i} \right\} \quad \forall i \neq 1.$$

We can rewrite (F.3) as $\mathbb{P}(\exists i \neq 1 : \tilde{G}_{i,T} \geq \tilde{G}_{1,T}) \leq \mathbb{P}(\bigcup_{i=1}^L \mathcal{E}_{i,T})$. Hence, it is sufficient to show

$$\left\{ \bigcap_{i=1}^L \mathcal{E}_{i,T}^C \right\} \ \Rightarrow \ \{\tilde{G}_{i,T} < G_{i,T} \quad \forall i \neq 1\}$$

as follows. When $\bigcap_{i=1}^L \mathcal{E}_{i,T}^C$ holds, for any $i \neq 1$, we have

$$\tilde{G}_{i,T} < G_{i,T} + \frac{T\Delta_{1,2}}{2} + T \cdot \Delta_{2,i} = G_{1,T} - T \cdot \Delta_{1,T} + \frac{T\Delta_{1,2}}{2} + T \cdot \Delta_{2,i}$$

$$< \tilde{G}_{1,T} + \frac{T\Delta_{1,2}}{2} - T \cdot \Delta_{1,T} + \frac{T\Delta_{1,2}}{2} + T \cdot \Delta_{2,i} = \tilde{G}_{1,T} T \cdot \Delta_{1,2} - (T \cdot \Delta_{1,2} + T \cdot \Delta_{2,T}) + T \cdot \Delta_{2,i}$$

$$= \tilde{G}_{1,T}.$$

$\square$

# G Analysis of global performances of adversarial algorithms

## G.1 Proof of Theorem B.3

**Theorem B.3.** *Let $0 < \underline{\Delta}_T \leq 1$. Then any algorithm $\pi$ satisfies that*

$$\sup_{\bar{\mathcal{B}}_1(\underline{\Delta}_T, 1)} \bar{e}_T(\pi, \mathcal{I}) \geq \frac{1 - \exp(-3T/200)}{4} \cdot \exp\left(-\frac{150T\underline{\Delta}_T^2}{L}\right).$$

*Furthermore, when $T \geq 10$,*

$$\sup_{\bar{\mathcal{B}}_1(\underline{\Delta}_T, 1)} \bar{e}_T(\pi, \mathcal{I}) \geq \frac{2}{65} \exp\left(-\frac{150T\underline{\Delta}_T^2}{L}\right).$$

*Proof.* **Step 1: Construct instances.** To begin, we let $Z_1, Z_2, \ldots, Z_T$ be a sequence of i.i.d. Gaussian random variables with mean $1/2$ and variance $\sigma^2 \in [1, \infty)$. Let $\varepsilon \in (0, 1/2)$ be a constant that will be chosen differently in each proof. Under instance $\ell$ $(1 \leq \ell \leq L)$, Let $g_{i,t}^\ell$ be the random gain of item $i$ at time step $t$, where

$$g_{i,1}^\ell = \begin{cases} 1/2 & \text{if } i = 1 \\ 1/2 + \varepsilon & \text{if } i = \ell \neq 1 \\ 1/2 - \varepsilon & \text{else} \end{cases}, \quad g_{i,t}^\ell = \begin{cases} \text{clip}_{[0,1]}(Z_t) & \text{if } i = 1 \\ \text{clip}_{[0,1]}(Z_t + \varepsilon) & \text{if } i = \ell \neq 1 \\ \text{clip}_{[0,1]}(Z_t - \varepsilon) & \text{else} \end{cases} \quad \forall t > 1.$$

Note that $\text{clip}_{[a,b]} x := \max\{a, \min\{b, x\}\}$ for $a \leq b$. Under instance $\ell$ $(1 \leq \ell \leq L)$, we define notations as follows:

(i) We let $G_{i,t}^\ell = \sum_{u=1}^t g_{i,t}^\ell$ and $T \cdot \bar{\Delta}_{i,j}^\ell = G_{i,T}^\ell - G_{j,T}^\ell$ for all $i, j \in [L]$, which indicates that $\ell = \arg\max_{i \in [L]} G_{i,T}^\ell$ is the optimal item.

(ii) We let $i_t^\ell$ be the pulled item at time step $t$, and $O_\ell^t = \{i_u^\ell, g_{i_u^\ell, \tau}\}_{u=1}^t$ be the sequence of pulled items and observed gains up to and including time step $t$.

(iii) We let $\mathbb{P}_\ell^t$ be the measure on $O_\ell^t$, and let $P_{\ell,i}$ be the measure on the gain of item $i$.

(iv) We define $\bar{\Delta}_{\min}^\ell := \min_{j \neq \ell} \bar{\Delta}_{\ell,j}^\ell$.

For simplicity, we abbreviate $\mathbb{P}_\ell^T$, $O_\ell^T$ as $\mathbb{P}_\ell$, $O_\ell$ respectively. Moreover, we let $N_i(t)$ denote the number of pulls of item $i$ up to and including time step $t$.

**Step 2: Change of measure.** First of all, we apply Lemmas C.6 and C.7 obtain that for all $1 \leq \ell \leq L$,

$$\Pr_{O_1}(i_{\text{out}} \neq 1) + \Pr_{O_\ell}(i_{\text{out}} = 1) \geq \frac{1}{2} \exp(-\text{KL}(\mathbb{P}_1 \| \mathbb{P}_\ell)) = \frac{1}{2} \exp(-\mathbb{E}_{\mathbb{P}_1}[N_\ell(T)] \cdot \text{KL}(P_{1,\ell} \| P_{\ell,\ell})).$$

Now we turn to bound $\mathbb{E}_{\mathbb{P}_1}[N_\ell(T)]$. Since $\sum_{\ell=1}^L \mathbb{E}_{\mathbb{P}_1}[N_\ell(T)] = T$, there exists $2 \leq \ell_2 \leq L$ such that $\mathbb{E}_{\mathbb{P}_1}[N_{\ell_2}(T)] \leq T/L$.

Further, with Lemma C.10, we can see that

$$\Pr_{O_1} i_{\text{out}} \neq 1) + \Pr_{\mathbb{P}_{\ell_2}}(i_{\text{out}} = 1) \geq \frac{1}{2} \exp(-\mathbb{E}_{\mathbb{P}_1}[N_{\ell_2}(T)] \cdot \text{KL}(P_{1,\ell_2} \| P_{\ell_2,\ell_2}))$$

$$\geq \frac{1}{2} \exp\left(-\frac{T}{L} \cdot \frac{(2\varepsilon)^2}{2\sigma^2}\right).$$

Since $\Pr_{O_{\ell_2}}(i_{\text{out}} \neq \ell_2) \geq \Pr_{O_{\ell_2}}(i_{\text{out}} = 1)$, we have

$$\max_{1 \leq \ell \leq L} \Pr_{O_\ell}(i_{\text{out}} \neq \ell) \geq \frac{1}{4} \exp\left(-\frac{2T\varepsilon^2}{\sigma^2 L}\right).$$

**Step 3: Comparison between $\varepsilon$ and $\bar{\Delta}^\ell_{\min}$.** (i) Under instance 1, since $G^1_{1,T} \geq G^1_{i,T} = G^1_{j,T}$ for all $i, j \neq 1$, i.e., item 1 is the optimal item and all other items are suboptimal with identical rewards after $T$ time steps, we have

$$\bar{\Delta}^1_{\min} = \min_{j \neq 1} \bar{\Delta}^1_{1,j} = \bar{\Delta}^1_{1,2} = \frac{G^1_{1,T} - G^1_{2,T}}{T} = \frac{1}{T} \cdot \sum_{t=1}^{T} [\text{clip}_{[0,1]}(Z_t) - \text{clip}_{[0,1]}(Z_t - \varepsilon)].$$

Let $X_t = \text{clip}_{[0,1]}(Z_t) - \text{clip}_{[0,1]}(Z_t - \varepsilon)$ and $X = \sum_{t=1}^{T} X_t$. Then $X = T\bar{\Delta}^1_{\min}$. We have

$$X_t \geq [\text{clip}_{[0,1]}[Z_t + \varepsilon] - \text{clip}_{[0,1]}(Z_t)] \cdot \mathbf{1}\{\varepsilon \leq Z_t \leq 1 - \varepsilon\} = \varepsilon \cdot \mathbf{1}\{\varepsilon \leq Z_t \leq 1 - \varepsilon\}.$$

Let $z\sigma = 1/2 - \varepsilon$. Theorem C.4 implies that

$$\Pr_{Z_t}\left(\left|Z_t - \frac{1}{2}\right| \geq \frac{1}{2} - \varepsilon\right) \leq \exp\left(-\frac{(1/2 - \varepsilon)^2}{2\sigma^2}\right)$$

$$\Rightarrow \Pr_{Z_t}(\varepsilon \leq Z_t \leq 1 - \varepsilon) \geq 1 - \exp\left(-\frac{(1 - 2\varepsilon)^2}{8\sigma^2}\right) := p(\varepsilon, \sigma).$$

Hence, we have $\mathbb{E}X_t \geq \varepsilon \cdot p(\varepsilon, \sigma)$.

Since $X_t \in [0, \varepsilon]$ for all $t$, $X_1/\varepsilon, \ldots, X_T/\varepsilon$ are independent $[0, 1]$-valued random variables, Theorem C.5 indicates that for all $a \in (0, 1)$,

$$\Pr(X - \mathbb{E}X \leq -a\mathbb{E}X) \leq \exp\left(-\frac{a^2 \mathbb{E}X}{3\varepsilon}\right) \Leftrightarrow \Pr(X \geq (1-a)\mathbb{E}X) \geq 1 - \exp\left(-\frac{a^2 \mathbb{E}X}{3\varepsilon}\right). \tag{G.1}$$

Let $b = 1 - a$. Since $\mathbb{E}X_t \geq \varepsilon \cdot p(\varepsilon, \sigma)$ for all $t$,

$$\Pr(X \geq bT\varepsilon \cdot p(\varepsilon, \sigma)) \geq 1 - \exp\left(-\frac{(1-b)^2 T \cdot p(\varepsilon, \sigma)}{3}\right) \quad \forall b \in (0, 1).$$

In order words,

$$\Pr(\bar{\Delta}^1_{\min} \geq b\varepsilon \cdot p(\varepsilon, \sigma)) \geq 1 - \exp\left(-\frac{(1-b)^2 T \cdot p(\varepsilon, \sigma)}{3}\right) \quad \forall b \in (0, 1), \; \varepsilon \in (0, 1/2).$$

(ii) Under instance $\ell$ ($\ell \neq 1$), since $G^\ell_{\ell,T} \geq G^\ell_{1,T} \geq G^\ell_{i,T} = G^\ell_{j,T}$ for all $i, j \notin \{1, \ell\}$, i.e., item $\ell$ is the optimal item, item 1 is the second optimal item, and all other items are with identical smaller rewards after $T$ time steps, we have

$$\bar{\Delta}^\ell_{\min} = \min_{j \neq \ell} \bar{\Delta}^\ell_{\ell,j} = \bar{\Delta}^\ell_{\ell,1} = \frac{G^\ell_{\ell,T} - G^\ell_{1,T}}{T} = \frac{1}{T} \cdot \sum_{t=1}^{T} [\text{clip}_{[0,1]}(Z_t + \Delta) - \text{clip}_{[0,1]}(Z_t)].$$

Let $X'_t = \text{clip}_{[0,1]}(Z_t + \varepsilon) - \text{clip}_{[0,1]}(Z_t)$ and $X' = \sum_{t=1}^{T} X'_t$. Then $X' = T\bar{\Delta}^\ell_{\min}$. We have

$$X'_t \geq [\text{clip}_{[0,1]}(Z_t + \varepsilon) - \text{clip}_{[0,1]}(Z_t)] \cdot \mathbf{1}\{\varepsilon \leq Z_t \leq 1 - \varepsilon\} = \varepsilon \cdot \mathbf{1}\{\varepsilon \leq Z_t \leq 1 - \varepsilon\}.$$

We again apply Theorem C.4 to $\mathbb{E}X'_t \geq \varepsilon \cdot p(\varepsilon, \sigma)$. Moreover, Theorem C.5 implies that

$$\Pr(\bar{\Delta}^\ell_{\min} \geq b\varepsilon \cdot p(\varepsilon, \sigma)) \geq 1 - \exp\left(-\frac{(1-b)^2 T \cdot p(\varepsilon, \sigma)}{3}\right) \quad \forall b \in (0, 1), \; \varepsilon \in (0, 1/2).$$

(iii) Altogether, for all $1 \leq \ell \leq L$, we have

$$\Pr(\bar{\Delta}^\ell_{\min} \geq b\varepsilon \cdot p(\varepsilon, \sigma)) \geq 1 - \exp\left(-\frac{(1-b)^2 T \cdot p(\varepsilon, \sigma)}{3}\right) \quad \forall b \in (0, 1), \; \varepsilon \in (0, 1/2). \tag{G.2}$$

**Step 4: Consider the instance with the largest error probability.** Let $1 \leq \ell_3 \leq L$ satisfy that

$$\Pr_{O_{\ell_3}} (i_{\text{out}} \neq \ell_3) \geq \frac{1}{4} \exp \left( - \frac{2T\varepsilon^2}{\sigma^2 L} \right).$$

Note that $\bar{\Delta}_{\min}^1, \ldots, \bar{\Delta}_{\min}^L$ are all determined by $\{Z_t\}_{t=1}^T$. We let $O' := O_{\ell_3} \cup \{Z_t\}_{t=1}^T$. Then for all $b \in (0,1)$,

$$
\begin{aligned}
\Pr_{O_{\ell_3}} (i_{\text{out}} \neq \ell_3) &\geq \Pr_{O'} \left( i_{\text{out}} \neq \ell_3, \ \bar{\Delta}_{\min}^{\ell_3} \geq b\varepsilon \cdot p(\varepsilon, \sigma) \right) \\
&\geq \Pr_{O'} \left( i_{\text{out}} \neq \ell_3 \mid \bar{\Delta}_{\min}^{\ell_3} \geq b\varepsilon \cdot p(\varepsilon, \sigma) \right) \cdot \Pr_{O'} \left( \bar{\Delta}_{\min}^{\ell_3} \geq b\varepsilon \cdot p(\varepsilon, \sigma) \right) \\
&\geq \frac{1}{4} \exp \left( - \frac{2T\varepsilon^2}{b^2 \sigma^2 L \cdot p^2(\varepsilon, \sigma)} \right) \cdot \left[ 1 - \exp \left( - \frac{(1-b)^2 T \cdot p(\varepsilon, \sigma)}{3} \right) \right].
\end{aligned}
$$

Let $\varepsilon = 1/10$, $\sigma = 1/3$, $b = 7/10$. Then

$$p(\varepsilon, \sigma) = 1 - \exp \left( - \frac{(1-2\varepsilon)^2}{8\sigma^2} \right) = 1 - \exp \left( - \frac{9}{8} \cdot \left( \frac{4}{5} \right)^2 \right) = 1 - \exp \left( - \frac{18}{25} \right) \geq \frac{1}{2}.$$

Moreover,

$$
\begin{aligned}
\Pr_{O_{\ell_3}} (i_{\text{out}} \neq \ell_3) &\geq \frac{1}{4} \exp \left( - \frac{2T(\bar{\Delta}_{\min}^{\ell_3})^2}{(7/10)^2 \cdot (1/3)^2 \cdot L \cdot (1/2)^2} \right) \cdot \left[ 1 - \exp \left( - \frac{(3/10)^2 \cdot T \cdot (1/2)}{3} \right) \right] \\
&= \frac{1}{4} \exp \left( - \frac{7200 T(\bar{\Delta}_{\min}^{\ell_3})^2}{49L} \right) \cdot \left[ 1 - \exp \left( - \frac{3T}{200} \right) \right] \\
&\geq \frac{1}{4} \exp \left( - \frac{150 T(\bar{\Delta}_{\min}^{\ell_3})^2}{L} \right) \cdot \left[ 1 - \exp \left( - \frac{3T}{200} \right) \right].
\end{aligned}
$$

When $T \geq 10$, since $1 - \exp(-3T/200) \geq 8/65$, we have

$$\Pr_{O_{\ell_3}} (i_{\text{out}} \neq \ell_3) \geq \frac{2}{65} \exp \left( - \frac{150 T(\bar{\Delta}_{\min}^{\ell_3})^2}{L} \right).$$

**Step 5: Classification of instances.** Let $\bar{\mathcal{B}}_1(\underline{\Delta}_T, \bar{R})$ denote the set of instances where (i) the empirically-minimal optimality gap $\bar{\Delta}_T \geq \underline{\Delta}_T$; and (ii) there exists $R_0 \in \mathbb{R}$ such the rewards are bounded in $[R_0, R_0 + \bar{R}]$. Then

$$\sup_{\bar{\mathcal{B}}_1(\underline{\Delta}_T, 1)} \Pr(i_{\text{out}} \neq \bar{i}_T^*) \geq \frac{1 - \exp(-3T/200)}{4} \cdot \exp \left( - \frac{150 T \underline{\Delta}_T^2}{L} \right) \quad \forall 0 < \underline{\Delta}_T \leq 1.$$

When $T \geq 10$,

$$\sup_{\bar{\mathcal{B}}_1(\underline{\Delta}_T, 1)} \Pr(i_{\text{out}} \neq \bar{i}_T^*) \geq \frac{2}{65} \exp \left( - \frac{150 T \underline{\Delta}_T^2}{L} \right) \quad \forall 0 < \underline{\Delta}_T \leq 1.$$

$\square$

## G.2 Proof of Theorem B.4

**Theorem B.4.** *Let $0 < \underline{\Delta}_T \leq 1$ and $T \geq 10$. Let $\pi$ be any algorithm with $\bar{e}_T(\pi, \mathcal{I}) \leq 2\exp(-\psi_T)/65$ for all $\mathcal{I} \in \bar{\mathcal{B}}_1(\underline{\Delta}_T, 1)$. Then*

$$\sup_{\mathcal{I} \in \bar{\mathcal{B}}_1(\underline{\Delta}_T, 1)} \mathbb{E}\bar{R}_T(\pi, \mathcal{I}) \geq \psi_T \cdot \frac{L - 1}{103 \underline{\Delta}_T}.$$

*Proof.* The analysis is similar to that of Theorem B.3 (See Appendix G.1). We construct $L$ instances in the same way.

**Step 1: Change of measure.** We apply Lemmas C.7 and C.6 to show that

$$\Pr_{O_1}(i_{\text{out}} \neq 1) + \Pr_{O_\ell}(i_{\text{out}} = 1) \geq \frac{1}{2} \exp(-\text{KL}(\mathbb{P}_1 \parallel \mathbb{P}_\ell)) = \frac{1}{2} \exp(-\mathbb{E}_{\mathbb{P}_1}[N_\ell(T)] \cdot \text{KL}(P_{1,\ell} \parallel P_{\ell,\ell})).$$

In order to upper bound $\mathbb{E}_{\mathbb{P}_1}[N_\ell(T)]$, we lower bound the number of time steps that $g_{1,t}^1 - g_{\ell,t}^1 = \varepsilon$, i.e., $Z_t \in [0, 1-\varepsilon]$. Let $z\sigma = 1/2 - \varepsilon$. We again apply Theorem C.4 to obtain (G.1):

$$\Pr_{Z_t}(\varepsilon \leq Z_t \leq 1-\varepsilon) \geq 1 - \exp\left(-\frac{(1-2\varepsilon)^2}{8\sigma^2}\right) = p(\varepsilon, \sigma).$$

Since the expectation of the empirical-regret is upper bounded by $\overline{\text{Reg}}$, we have

$$\overline{\text{Reg}} \geq \mathbb{E}_{\mathbb{P}_1}\left[\sum_{t=1}^T \mathbf{1}\{i_t^1 \neq 1\} \cdot (g_{1,t}^1 - g_{i_t,t}^1)\right] \geq \mathbb{E}_{\mathbb{P}_1}\left[\sum_{t=1}^T \mathbf{1}\{i_t^1 \neq 1, \Delta \leq Z_t \leq 1-\varepsilon\} \cdot (g_{1,t}^1 - g_{i_t,t}^1)\right]$$

$$= \sum_{t=1}^T \mathbb{E}_{\mathbb{P}_1^t}[\mathbf{1}\{i_t^1 \neq 1\} \cdot [\text{clip}_{[0,1]}(Z_t) - \text{clip}_{[0,1]}(Z_t - \varepsilon)] \mid \varepsilon \leq Z_t \leq 1-\varepsilon] \cdot \Pr_{Z_t}(\varepsilon \leq Z_t \leq 1-\varepsilon)$$

$$= \sum_{t=1}^T \mathbb{E}_{\mathbb{P}_1^t}[\mathbf{1}\{i_t^1 \neq 1\} \cdot [Z_t - (Z_t - \varepsilon)] \mid \varepsilon \leq Z_t \leq 1-\varepsilon] \cdot p(\varepsilon, \sigma)$$

$$= \sum_{t=1}^T \mathbb{E}_{\mathbb{P}_1^t}[\mathbf{1}\{i_t^1 \neq 1\}] \cdot \varepsilon \cdot p(\varepsilon, \sigma) = \varepsilon \cdot p(\varepsilon, \sigma) \cdot \sum_{\ell=2}^L \mathbb{E}_{\mathbb{P}_1}[N_\ell(T)].$$

Hence, there exists $2 \leq \ell_0 \leq L$ such that

$$\mathbb{E}_{\mathbb{P}_1}[N_{\ell_0}(T)] \leq \frac{\overline{\text{Reg}}}{\varepsilon \cdot p(\varepsilon, \sigma) \cdot (L-1)}.$$

Further, we again apply Lemma C.10 to obtain

$$\Pr_{O_1}(i_{\text{out}} \neq 1) + \Pr_{O_{\ell_0}}(i_{\text{out}} = 1) \geq \frac{1}{2} \exp(-\mathbb{E}_{\mathbb{P}_1}[N_{\ell_0}(T)] \cdot \text{KL}(P_{1,\ell_0} \parallel P_{\ell_0,\ell_0}))$$

$$\geq \frac{1}{2} \exp\left(-\frac{\overline{\text{Reg}}}{\varepsilon \cdot p(\varepsilon, \sigma) \cdot (L-1)} \cdot \frac{(2\varepsilon)^2}{2\sigma^2}\right).$$

Since $\Pr_{O_{\ell_0}}(i_{\text{out}} \neq \ell_0) \geq \Pr_{O_{\ell_0}}(i_{\text{out}} = 1)$, we have

$$\max_{1 \leq \ell \leq L} \Pr_{O_\ell}(i_{\text{out}} \neq \ell) \geq \frac{1}{4} \exp\left(-\frac{2\varepsilon \overline{\text{Reg}}}{\sigma^2 \cdot p(\varepsilon, \sigma) \cdot (L-1)}\right)$$

where $p(\varepsilon, \sigma) = 1 - \exp[-(1-2\varepsilon)^2/(8\sigma^2)]$.

**Step 2: Consider the instance with the largest error probability.** Recall (G.2) from the analysis of Theorem B.3 in Appendix G.1. For all $1 \leq \ell \leq L$, we have

$$\Pr(\bar{\Delta}_{\min}^\ell \geq b\varepsilon \cdot p(\varepsilon, \sigma)) \geq 1 - \exp\left(-\frac{(1-b)^2 T \cdot p(\varepsilon, \sigma)}{3}\right) \quad \forall b \in (0, 1), \ \varepsilon \in (0, 1/2).$$

Let $1 \leq \ell_2 \leq L$ satisfy that

$$\Pr_{O_{\ell_2}}(i_{\text{out}} \neq \ell_2) \geq \frac{1}{4} \exp\left(-\frac{2\varepsilon \overline{\text{Reg}}}{\sigma^2 \cdot p(\varepsilon, \sigma) \cdot (L-1)}\right).$$

Note that $\bar{\Delta}^1_{\min}, \ldots, \bar{\Delta}^L_{\min}$ are all determined by $\{Z_t\}_{t=1}^T$. We let $O' := O_{\ell_2} \cup \{Z_t\}_{t=1}^T$. Then for all $b \in (0, 1)$, $1 \le \ell \le L$,

$$
\begin{aligned}
\Pr_{O_{\ell_2}}(i_{\text{out}} \ne \ell_2) &\ge \Pr_{O'}\left( i_{\text{out}} \ne \ell_2, \ \bar{\Delta}^\ell_{\min} \ge b\varepsilon \cdot p(\varepsilon, \sigma) \right) \\
&\ge \Pr_{O'}\left( i_{\text{out}} \ne \ell_2 \mid \bar{\Delta}^\ell_{\min} \ge b\varepsilon \cdot p(\varepsilon, \sigma) \right) \cdot \Pr_{O'}\left( \bar{\Delta}^\ell_{\min} \ge b\varepsilon \cdot p(\varepsilon, \sigma) \right) \\
&\ge \frac{1}{4} \exp\left( -\frac{2\bar{\Delta}^\ell_{\min} \cdot \overline{\text{Reg}}}{\sigma^2 \cdot b \cdot p^2(\varepsilon, \sigma) \cdot (L-1)} \right) \cdot \left[ 1 - \exp\left( -\frac{(1-b)^2 T \cdot p(\varepsilon, \sigma)}{3} \right) \right].
\end{aligned}
$$

Since this inequality holds for all $1 \le \ell \le L$, we let $\bar{\Delta}_{\min} = \min_{1 \le \ell \le L} \bar{\Delta}^\ell_{\min}$, and have

$$
\Pr_{O_{\ell_2}}(i_{\text{out}} \ne \ell_2) \ge \frac{1}{4} \exp\left( -\frac{2\bar{\Delta}_{\min} \cdot \overline{\text{Reg}}}{\sigma^2 \cdot b \cdot p^2(\varepsilon, \sigma) \cdot (L-1)} \right) \cdot \left[ 1 - \exp\left( -\frac{(1-b)^2 T \cdot p(\varepsilon, \sigma)}{3} \right) \right].
$$

We again let $\varepsilon = 1/10$, $\sigma = 1/3$, $b = 7/10$. Then $p(\varepsilon, \sigma) \ge 1/2$, and

$$
\begin{aligned}
\Pr_{O_{\ell_2}}(i_{\text{out}} \ne \ell_2) &\ge \frac{1}{4} \exp\left( -\frac{2\bar{\Delta}_{\min} \cdot \overline{\text{Reg}}}{(1/3)^2 \cdot (7/10) \cdot (1/2)^2 \cdot (L-1)} \right) \cdot \left[ 1 - \exp\left( -\frac{(3/10)^2 \cdot T \cdot (1/2)}{3} \right) \right] \\
&\ge \frac{1}{4} \exp\left( -\frac{720 \bar{\Delta}_{\min} \cdot \overline{\text{Reg}}}{7(L-1)} \right) \cdot \left[ 1 - \exp\left( -\frac{3T}{200} \right) \right] \\
&\ge \frac{1}{4} \exp\left( -\frac{103 \bar{\Delta}_{\min} \cdot \overline{\text{Reg}}}{L-1} \right) \cdot \left[ 1 - \exp\left( -\frac{3T}{200} \right) \right].
\end{aligned}
$$

When $T \ge 10$, since $1 - \exp(-3T/200) \ge 8/65$, we have

$$
\Pr_{O_{\ell_2}}(i_{\text{out}} \ne \ell_2) \ge \frac{2}{65} \exp\left( -\frac{103 \bar{\Delta}_{\min} \cdot \overline{\text{Reg}}}{L-1} \right).
$$

Suppose algorithm $\pi$ satisfies that

$$
\Pr_{O_{\ell_2}}(i_{\text{out}} \ne \ell_2) \le \frac{2}{65} \exp(-\psi_T),
$$

then we have

$$
\overline{\text{Reg}} \ge \frac{L-1}{103 \bar{\Delta}_{\min}} \cdot \psi_T.
$$

**Step 3: Classification of instances.** Suppose algorithm $\pi$ satisfies that $\bar{e}_T(\pi) \le 2 \exp(-\psi_T)/65$. We again consider $\bar{\mathcal{B}}_1(\underline{\Delta}_T, 1)$ as in Theorem B.3. Recall that $\bar{\mathcal{B}}_1(\underline{\Delta}, \bar{R})$ denote the set of instances where (i) the empirically-minimal optimality gap $\bar{\Delta}_{\min, T} \ge \underline{\Delta}_T$ in $T$ time steps; and (ii) there exists $R_0 \in \mathbb{R}$ such the rewards are bounded in $[R_0, R_0 + R]$. When $T \ge 10$,

$$
\sup_{\mathcal{I} \in \bar{\mathcal{B}}_1(\underline{\Delta}_T, 1)} \mathbb{E} \bar{R}_T(\pi, \mathcal{I}) \ge \psi_T \cdot \frac{L-1}{103 \underline{\Delta}_T} \quad \forall 0 < \underline{\Delta}_T \le 1.
$$

$\square$

# H  Additional numerical results

We present the failure probabilities and counts of algorithms in different instances in Appendix H.1. We provide additional numerical results for both synthetic and real datasets in Appendices H.3 and H.4 respectively. We also elaborate more details about the experiment setup of the PKIS2 dataset in Appendix H.4.

### H.1 Empirical failure probability of BoBW-LIL'UCB($\gamma$)

Table H.1: Empirical failure probability below 1%

| | Bernoulli instances | | | | ML-25M | PKIS2 |
|---|---|---|---|---|---|---|
| $L$ | 64 | | 128 | | 22 | 109 |
| $\Delta$ | 0.05 | 0.1 | 0.1 | 0.2 | | |
| BoBW($6 \times 10^{-7}$) | 0.83% | 0.22% | 0.78% | 0.20% | 0.96% | 0.92% |
| BoBW($6 \times 10^{-4}$) | 0.65% | 0.17% | 0.62% | 0.18% | 0.72% | 0.23% |
| BoBW($6 \times 10^{-1}$) | 0.11% | 0.03% | 0.14% | 0.04% | 0.06% | 0 |

Table H.2: Empirical failure probability below 2%

| | Bernoulli instances | | | | ML-25M | PKIS2 |
|---|---|---|---|---|---|---|
| $L$ | 64 | | 128 | | 22 | 109 |
| $\Delta$ | 0.05 | 0.1 | 0.1 | 0.2 | | |
| BoBW($6 \times 10^{-7}$) | 1.78% | 1.60% | 1.06% | 0.31% | 0.96% | 1.68% |
| BoBW($6 \times 10^{-4}$) | 1.21% | 0.95% | 0.61% | 0.16% | 0.72% | 0.42% |
| BoBW($6 \times 10^{-1}$) | 0.28% | 0.23% | 0.06% | 0.03% | 0.06% | 0 |

**H.2 Empirical regret of algorithms with empirical failure probability below** $1\%$

Table H.3: Empirical regret of algorithms using synthetic data

| Algorithm | $L$ | $\Delta$ | Average regret | Standard deviation of regret |
|---|---|---|---|---|
| BoBW($9 \times 10^{-1}$) | 64 | 0.05 | $6.45 \times 10^3$ | $5.10 \times 10^1$ |
| BoBW($9 \times 10^{-4}$) | 64 | 0.05 | $6.59 \times 10^3$ | $1.38 \times 10^1$ |
| BoBW($9 \times 10^{-7}$) | 64 | 0.05 | $6.61 \times 10^3$ | 9.27 |
| UCB$_{3.0}$ | 64 | 0.05 | $1.84 \times 10^4$ | $2.31 \times 10^3$ |
| UCB$_{4.5}$ | 64 | 0.05 | $2.34 \times 10^4$ | $3.41 \times 10^3$ |
| UCB$_{6.0}$ | 64 | 0.05 | $2.64 \times 10^4$ | $4.22 \times 10^3$ |
| BoBW($9 \times 10^{-1}$) | 64 | 0.1 | $3.78 \times 10^3$ | $3.74 \times 10^1$ |
| BoBW($9 \times 10^{-4}$) | 64 | 0.1 | $3.90 \times 10^3$ | 8.53 |
| BoBW($9 \times 10^{-7}$) | 64 | 0.1 | $3.91 \times 10^3$ | 5.64 |
| UCB$_{3.0}$ | 64 | 0.1 | $8.89 \times 10^3$ | $1.12 \times 10^3$ |
| UCB$_{4.5}$ | 64 | 0.1 | $1.13 \times 10^4$ | $1.68 \times 10^3$ |
| UCB$_{6.0}$ | 64 | 0.1 | $1.28 \times 10^4$ | $2.07 \times 10^3$ |
| BoBW($9 \times 10^{-1}$) | 128 | 0.1 | $6.82 \times 10^3$ | $3.26 \times 10^1$ |
| BoBW($9 \times 10^{-4}$) | 128 | 0.1 | $6.91 \times 10^3$ | 7.37 |
| BoBW($9 \times 10^{-7}$) | 128 | 0.1 | $6.92 \times 10^3$ | 4.83 |
| UCB$_{3.0}$ | 128 | 0.1 | $1.84 \times 10^4$ | $2.21 \times 10^3$ |
| UCB$_{4.5}$ | 128 | 0.1 | $2.35 \times 10^4$ | $3.39 \times 10^3$ |
| UCB$_{6.0}$ | 128 | 0.1 | $2.67 \times 10^4$ | $4.19 \times 10^3$ |
| BoBW($9 \times 10^{-1}$) | 128 | 0.2 | $3.87 \times 10^3$ | $2.49 \times 10^1$ |
| BoBW($9 \times 10^{-4}$) | 128 | 0.2 | $3.95 \times 10^3$ | 4.47 |
| BoBW($9 \times 10^{-7}$) | 128 | 0.2 | $3.96 \times 10^3$ | 2.86 |
| UCB$_{3.0}$ | 128 | 0.2 | $8.72 \times 10^3$ | $1.09 \times 10^3$ |
| UCB$_{4.5}$ | 128 | 0.2 | $1.11 \times 10^4$ | $1.62 \times 10^3$ |
| UCB$_{6.0}$ | 128 | 0.2 | $1.26 \times 10^4$ | $2.01 \times 10^3$ |

Table H.4: Empirical regret of algorithms using the ML-25M dataset

| Algorithm | $L$ | Average regret | Standard deviation of regret |
|---|---|---|---|
| BoBW ($9 \times 10^{-1}$) | 22 | $4.05 \times 10^3$ | 113.98 |
| BoBW ($9 \times 10^{-4}$) | 22 | $5.16 \times 10^3$ | 43.93 |
| BoBW ($9 \times 10^{-7}$) | 22 | $5.38 \times 10^3$ | 31.79 |
| UCB $_{3.0}$ | 22 | $7.05 \times 10^3$ | 927.60 |
| UCB $_{4.5}$ | 22 | $11.22 \times 10^3$ | $1.77 \times 10^3$ |
| UCB $_{6.0}$ | 22 | $15.04 \times 10^3$ | $2.67 \times 10^3$ |

Table H.5: Empirical regret of algorithms using the PKIS2 dataset

| Algorithm | $L$ | Average regret | Standard deviation of regret |
|---|---|---|---|
| BoBW($9 \times 10^{-1}$) | 109 | $8.63 \times 10^6$ | $2.31 \times 10^3$ |
| BoBW($9 \times 10^{-4}$) | 109 | $8.73 \times 10^6$ | $1.55 \times 10^3$ |
| BoBW($9 \times 10^{-7}$) | 109 | $8.78 \times 10^6$ | $1.32 \times 10^3$ |
| UCB$_{3.0}$ | 109 | $17.01 \times 10^6$ | $2.33 \times 10^6$ |
| UCB$_{4.5}$ | 109 | $21.50 \times 10^6$ | $2.94 \times 10^6$ |
| UCB$_{6.0}$ | 109 | $26.31 \times 10^6$ | $3.71 \times 10^6$ |

### H.3 Experiments using synthetic data

In this section, we present more numerical results for larger instances with $L = 128$ items. These figures yield the same conclusions as in Section 6.1.

**Experiments with empirical failure probabilities below $1\%$.**

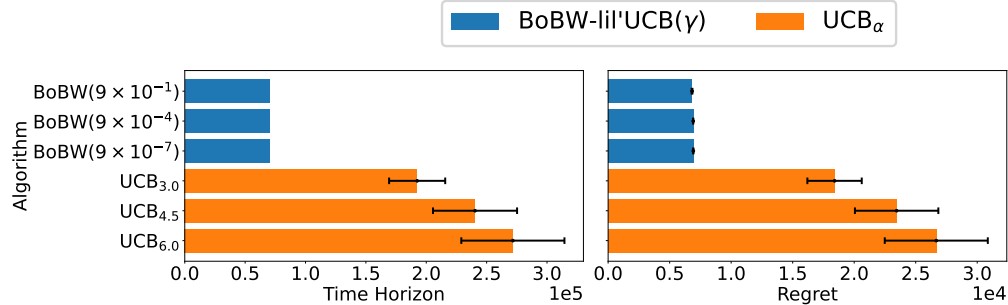

Figure H.1: Empirical failure probability $\leq 1\%$: $L = 128$, $\Delta = 0.1$, $\nu_i = \mathrm{Bern}(w_i)$.

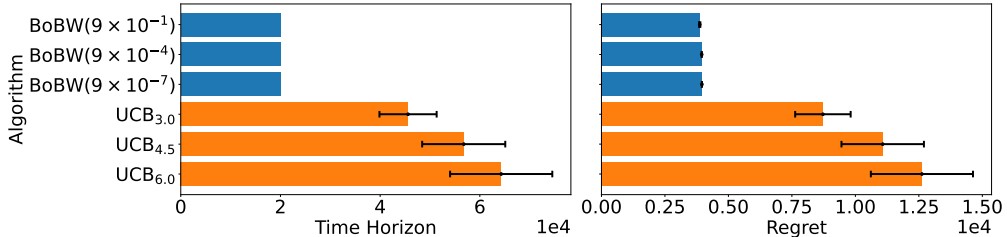

Figure H.2: Empirical failure probability $\leq 1\%$: $L = 128$, $\Delta = 0.2$, $\nu_i = \mathrm{Bern}(w_i)$.

**Experiments with empirical failure probabilities below $2\%$.**

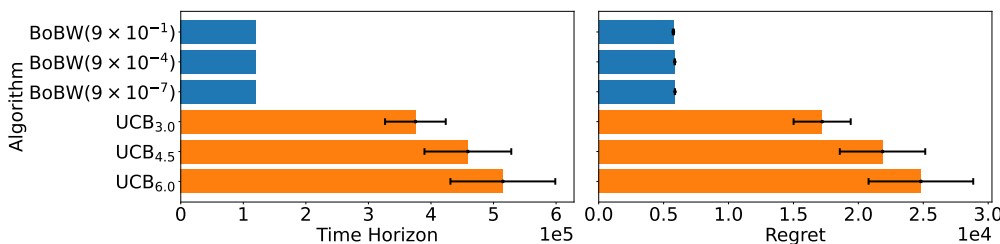

Figure H.3: Empirical failure probability $\leq 2\%$: $L = 64$, $\Delta = 0.05$, $\nu_i = \mathrm{Bern}(w_i)$.

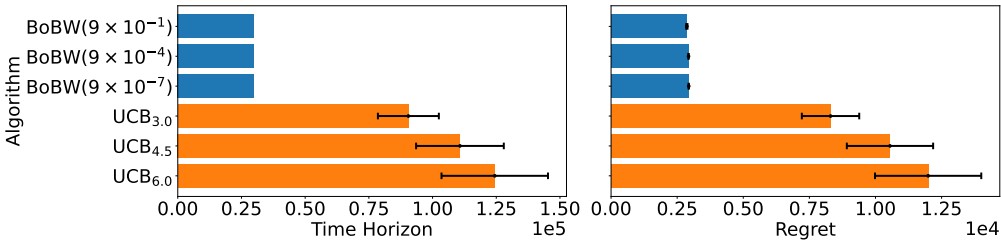

Figure H.4: Empirical failure probability $\leq 2\%$: $L = 64$, $\Delta = 0.1$, $\nu_i = \mathrm{Bern}(w_i)$.

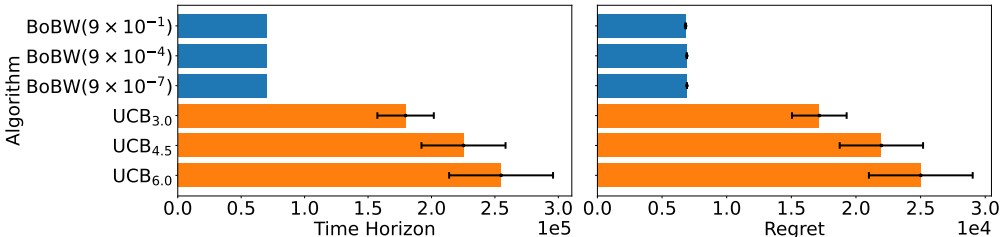

Figure H.5: Empirical failure probability $\leq 2\%$: $L = 128$, $\Delta = 0.1$, $\nu_i = \mathrm{Bern}(w_i)$.

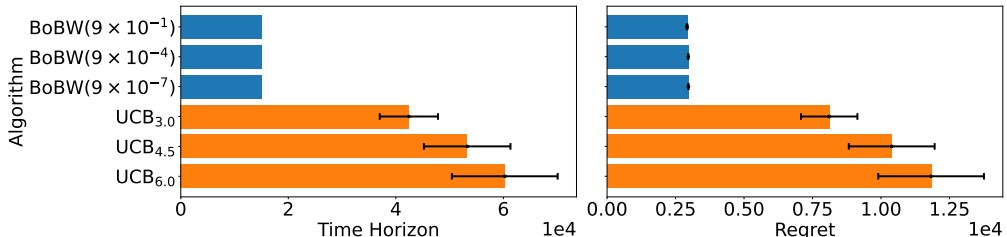

Figure H.6: Empirical failure probability $\leq 2\%$: $L = 128$, $\Delta = 0.2$, $\nu_i = \mathrm{Bern}(w_i)$.

### H.4 Experiments using real data

**PKIS2 dataset.** The repository tests 641 small molecule compounds (kinase inhibitor) against 406 protein kinases. This experiment aims to find the most effective inihibitor against a targeted kinase, and is a fundamental study in cancer drug discovery. PKIS2 presents a 'percentage inhibition' for each inhibitor, which is averaged over several trials. For each entry, we normalize it to be between 0 and 1, and then obtain the *percentage control* by subtracting each of the normalized entries from 1. The percentage control can help understand how effective the inhibitor is against the targeted kinase. Since Christmann-Franck et al. (2016) reported that these values have log-normal distributions with variance less than 1, we sample random variables form a standard normal distribution with the log of the percent control as the mean; the similar setup was used in Mason et al. (2020); Mukherjee et al. (2021). In our experiment, we select the inhibitors tested against one specific kinase MAPKAPK5. We aim to find out the most effective inhibitor with the highest percentage control against MAPKAPK5, and also obtain high percentage controls cumulatively during the process. Our results may benefit the experiments that test inhibitors with genuine cancer patients, which helps to identify the most effective inhibitor with a fixed number of tests and provide effective solutions to the attendants during the tests.

**Experiments with empirical failure probabilities below $2\%$.**

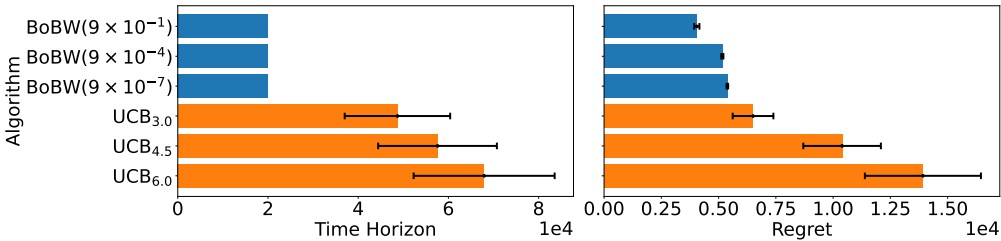

Figure H.7: Empirical failure probability $\leq 2\%$. ML-25M: $L = 22$ movies with at least $50,000$ ratings.

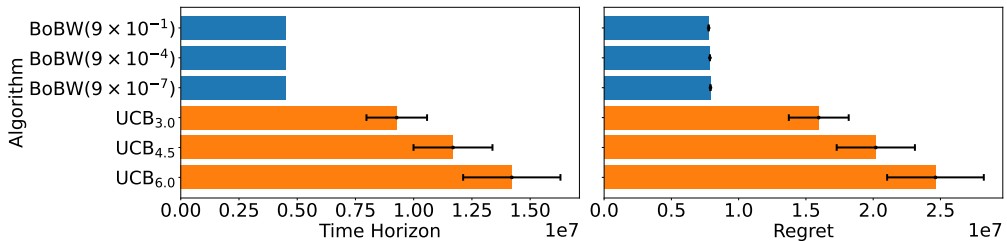

Figure H.8: Empirical failure probability $\leq 2\%$. PKIS2: $L = 109$ inhibitors tested against MAPKAPK5.

