# OpenReview forum: "Achieving the Pareto Frontier of Regret Minimization and Best Arm Identification in Multi-Armed Bandits"
_TMLR — Accepted by TMLR_

### Review · Reviewer_uXLo · 2023-07-06

**Summary Of Contributions:**

The paper studies regret minimization and best arm identification for multi-armed bandits. The main contributions are:
1. The authors developed a new algorithm called BoBW-LIL'UCB. By specifically tuning a parameter $\gamma$ of the algorithm, this new algorithm can either achieve the optimal guarantee for regret minimization of best arm identification (in the fixed budget setting).
2. The authors theoretically quantify the Pareto frontier of regret minimization and best arm identification, and show that cannot hope to simultaneously achieve the optimal guarantees for both regret minimization and best action identification.

**Audience:**

Yes

**Claims And Evidence:**

Yes

**Requested Changes:**

See the weaknesses section.

**Strengths And Weaknesses:**

Strengths: To my knowledge, this is the first paper studies the Pareto frontier of regret minimization and best arm identification in the fixed budget setting. This paper developed both upper and lower bounds for this setting, and I believe such results are important and interesting.

Weaknesses: For the best arm identification setting, it seems that the optimal guarantee is only achieved when some problem-dependent parameter is known to the learner (e.g., the minimum reward gap $\Delta$ in Theorem 4.2). Can authors comment on this issue? E.g., is it possible to achieve the optimal guarantee without knowing such parameters?

While I appreciate the authors analyzing many existing algorithms for the proposed setting, I would encourage the authors to add more comparison algorithms, e.g., the vanilla LIL'UCB algorithm. Or highlighting why the vanilla LIL'UCB cannot achieve the optimal guarantees (after adapting it to the fixed budget setting, which should not be hard).

---

### Review · Reviewer_F4h1 · 2023-07-18

**Summary Of Contributions:**

This work conducts inquiry into the competing objectives of cumulative regret minimization and best arm identification in the stochastic multi-armed bandit problem, and attempts to flesh out a Pareto frontier of achievable performance between the two. This is shown via a lower bound on the cumulative regret of any algorithm that achieves BAI with a given failure rate. The authors also propose a UCB-style algorithm (similar to the one in Jamieson et al. (2014)) that can be fine-tuned to trade off performance between the two objectives.

**Audience:**

Yes

**Broader Impact Concerns:**

Broader impact statement N/A.

**Claims And Evidence:**

Yes

**Requested Changes:**

I am curious if the logarithmic factor in the problem-independent bound can be shaved off?

Also, how would the failure probability $e_T$ look like under classical UCB1 (Auer et al. (2002)) with an exploration coefficient of $\alpha$? This is similar to UCB-E(α log T ) but with $\log t$ instead of $\log T$. Essentially, does knowledge of horizon buy you any gains in the failure rate? Also, the failure probability of UCB-E(α log T ), as it appears in Table 3.1, is misleadingly problem-independent. Please include the condition on $T$ and $H_2$ from Corollary A.1 to avoid ambiguity.

**Strengths And Weaknesses:**

It appears that while the algorithm can be tuned to achieve optimal (or near-optimal) cumulative regret by setting $\gamma = \log T/T$, it cannot be calibrated for BAI at a near-optimal rate as this requires setting $\gamma = \gamma_1\left( \Delta, H_2 \right)$, which requires knowledge of the primitives. Is this true?

---

### Review · Reviewer_2ndQ · 2023-08-10

**Summary Of Contributions:**

This paper proposes a Best of Both worlds algorithm for best arm identification in multi-armed bandits. The algorithm BoBW-LIL'UCB($\gamma$) recovers the Pareto frontier for regret minimization and best arm identification. Additionally they also a series of lower bounds showing the no algorithm can perform optimally for BA and RM objectives simultaneously.

**Audience:**

Yes

**Broader Impact Concerns:**

This work does not raise any major broader impact concerns.

**Claims And Evidence:**

Yes

**Requested Changes:**

As I mentioned in the previous vignette, I didn't find any major issues with this work. I only found a minor 'issue'. In corollary 5.2 there is a boundary condition that is not properly handled. If $\sup_{\mathcal{I} \in \mathcal{B}_1(\underbar \Delta, 1)} R_T(\pi, \mathcal{I}) = \phi_T \frac{(L-1) \bar{R}}{8 \underbar \Delta}$ I am not sure the implication is clear.

**Strengths And Weaknesses:**

Strengths:

1. The paper is very well written. It was a pleasure to read it. The discussion on previous work was substantial, well researched and easy to read.

2.  The algorithmic contribution is solid and matches the objectives of the paper as well as the verbiage used by the authors when explaining their contributions.

3. All facets of the problem are explored in detail including lower bounds.

4. The experimental evaluation us thorough and tries out their methods beyond synthetic problems.

Weaknesses:

I didn't find any major flaws in this work. It is likely it has already gone through several rounds of reviewing. It is in great shape.

---

### Decision · Action_Editors · 2023-09-09

**Recommendation:** Accept as is

**Comment:**

This paper proposes a Best of Both worlds algorithm for best arm identification in multi-armed bandits. The algorithm recovers the Pareto frontier for regret minimization and best arm identification. Further they prove lower bounds showing the no algorithm can perform optimally for BA and RM objectives simultaneously.

The reviewers unanimously liked the paper and found the paper well written and well presented. The results were appreciated by the reviewers. Happy to recommend an accept.

**Audience:**

The paper provides a strong result in the extremely well studied field of regret minimization and best arm identification in multi armed bandits.

**Claims And Evidence:**

Claims made in the submission are supported by accurate, convincing and clear evidence